# A high-resolution physical-biogeochemical model for marine resource applications in the Northern Indian Ocean (MOM6-COBALT-IND12 v1.0)

Enhui Liao[1], Laure Resplandy[2], Fan Yang[2], Yangyang Zhao[2], Sam Ditkovsky[3], Manon Malsang[4], Jenna Pearson[5], Andrew C. Ross[6], Robert Hallberg[6], and Charles Stock[6]

[1]School of Oceanography, Shanghai Jiao Tong University, Shanghai, China
[2]Department of Geosciences and High Meadows Environmental Institute, Princeton University, Princeton, NJ, USA
[3]Program in Atmospheric and Oceanic Sciences, Sayre Hall, Princeton University, Princeton, NJ, USA
[4]Laboratoire d'Océanographie et du Climat Expérimentations et Approches Numériques (LOCEAN), Institut Pierre-Simon Laplace (IPSL), Sorbonne Université, Paris, France
[5]Climatematch Academy, Neuromatch Inc., USA
[6]NOAA OAR Geophysical Fluid Dynamics Laboratory, Princeton, NJ, USA

**Correspondence:** Laure Resplandy (laurer@princeton.edu); Enhui Liao (ehliao@sjtu.edu.cn)

**Abstract.** We introduce and evaluate the regional ocean model MOM6-COBALT-IND12 version 1 coupling the MOM6 ocean dynamics model to the Carbon, Ocean Biogeochemistry and Lower Trophics (COBALT) biogeochemical model at a horizontal resolution of $1/12°$. The model covers the northern Indian Ocean (from $8.6°S$ to the northern continental boundaries), central to the livelihoods and economies of countries that comprise about one-third of the world's population. We demonstrate that

the model effectively captures the key physical and biogeochemical basin-scale features related to seasonal monsoon reversal, interannual Indian Ocean Dipole and multi-decadal variability, as well as intraseasonal and fine-scale variability (e.g., eddies and planetary waves), which are all essential for accurately simulating patterns of coastal upwelling, primary productivity, temperature, salinity, and oxygen levels. Well represented features include the timing and amplitude of the monsoonal blooms triggered by summer coastal upwelling and winter mixing, the strong contrast between the high evaporation / high salinity

Arabian Sea and high precipitation / high runoff / low salinity Bay of Bengal, the seasonality of the Great Whirl gyre and coastal Kelvin upwelling/downwelling waves, as well as the physical and biogeochemical patterns associated with intraseasonal and interannual variability. Quantitatively, the model exhibits relatively small biases, as reflected by root mean square error (RMSE) values in key variables: sea surface temperature ($0.25–0.3$ °C), mixed layer depth ($7–8.09$ m), sea level anomaly ($0.02$ m), sea surface salinity ($0.53-0.71$ psu), vertical chlorophyll ($0.03-0.3$ mg m$^{-3}$), subsurface temperature ($0.33$ °C), and subsurface

salinity ($0.07$ psu). A major model bias ($16$ $\mu$mol kg$^{-1}$ of oxygen) is the larger oxygen minimum zone simulated in the Bay of Bengal, a common challenge of ocean and Earth system models in this region. This bias was partly mitigated by improving the representation of the export and burial of organic detritus to the deep ocean (e.g., sinking speed, riverine lithogenic material inputs that protect organic material and burial fraction), and water-column denitrification (e.g., nitrate-based respiration at higher oxygen levels) using observational constraints. These results indicate that the regional MOM6-COBALT-IND12 v1.0

model is well suited for physical and biogeochemical studies on timescales ranging from weeks to decades, in addition to supporting marine resource applications and management in the northern Indian Ocean.

## 1 Introduction

The northern Indian Ocean (from 8.6°S to the northern continental boundaries and 32°E to 114°E) is central to the livelihood and economy of about one third of the Earth's population which lives in its littoral countries (e.g., India, Indonesia, Pakistan, Bangladesh, Tanzania, Myanmar, Malaysia, Kenya, and Yemen) and provides valuable resources via the "blue economy", such as fishery, aquaculture, and marine tourism (Roy, 2019). A major challenge to understand and anticipate the response of Indian Ocean ecosystems is to account for the full range of spatio-temporal variability and human-driven changes that control the climatic and environmental conditions defining the habitat, success and survival of these ecosystems (Phillips et al., 2021; Pinsky et al., 2013; Deutsch et al., 2015). On seasonal and interannual time-scales, the Indian monsoon and the Indian Ocean Dipole (IOD) control the ocean circulation and regulate temperature (Schott and McCreary, 2001; Saji et al., 1999; Beal et al., 2013), oxygen levels (Resplandy et al., 2012; Vallivattathillam et al., 2017; Pearson et al., 2022; Al Azhar et al., 2017) and primary productivity (Barber et al., 2001; Gauns et al., 2005; Prakash and Ramesh, 2007; Lévy et al., 2007; Kumar et al., 2010; Wiggert et al., 2009; Resplandy et al., 2011; Currie et al., 2013; Sarma and Dalabehera, 2019), with implications for the spatial and temporal distribution of species that are commercially valuable such as tuna, and key to local food security such as small pelagic fish (e.g., Jebri et al., 2020; Wang et al., 2023).

On decadal and multi-decadal timescales, the Indian Ocean has undergone rapid warming, with an increase in sea surface temperature (SST) by about 1°C since the 1950s (Roxy et al., 2020), a decline in primary productivity (Sunanda et al., 2023; Sridevi et al., 2023; Gregg and Rousseaux, 2019; Dalpadado et al., 2021), and a significant loss in oxygen in the Arabian Sea and Bay of Bengal (Banse et al., 2014; Piontkovski and Al-Oufi, 2015; Queste et al., 2018; Rixen et al., 2019a; Naqvi, 2019; Löscher, 2021; Lachkar et al., 2023) as well as in the water masses supplying oxygen to the Indian Ocean (Helm et al., 2011; Ito et al., 2017; Naqvi, 2021; Ditkovsky et al., 2023). Warming, decline in primary productivity, and oxygen loss are projected to continue in the Indian Ocean unless greenhouse gas emissions are rapidly curtailed (Bopp et al., 2013; Kwiatkowski et al., 2017, 2020; Roxy et al., 2020; Lachkar et al., 2018, 2019; Lévy et al., 2022; Ditkovsky et al., 2023; Sharma et al., 2023). Warming is also expected to weaken the monsoon despite a potential increase in extreme rainfall events (e.g., Sooraj et al., 2015; Singh et al., 2019; Roxy et al., 2020). This could modify the supply of freshwater and nutrients to coastal waters, and increase the frequency of extreme positive IOD events (Roxy et al., 2020; Cai et al., 2021), which are known to induce weather extremes (Cai et al., 2021), promote primary productivity in the eastern tropical Indian Ocean (e.g., Wiggert et al., 2009; Currie et al., 2013) and lead to low coastal oxygen levels (coastal hypoxia) in the eastern Bay of Bengal (Pearson et al., 2022). Projections from Coupled Model Intercomparison Project (CMIP) models suggest substantial shifts in net primary production and sharp declines in pH in the coming decades, highlighting the northern Indian Ocean's particular vulnerability to climate change (Sunanda et al., 2021, 2023). Observations indicate that these changes have already impacted ecosystems in the Indian Ocean. For instance, do Rosário Gomes et al. (2008) found that the dominant phytoplankton group during the

winter bloom in the Arabian Sea shifted from diatom to dinoflagellate in recent decades in response to warming and oxygen loss, with potentially large implications for the functioning of this ecosystem. In coastal areas, the effect of natural variability associated with the seasonal monsoon and interannual IOD combines with global warming and anthropogenic activities (waste waters, urbanization, fertilizers etc.) leading to coastal hypoxic events and in extreme cases to massive mortality events with implications for coastal fisheries and aquaculture (low oxygen levels, Naqvi et al., 2009; Naqvi, 2021, 2022; Pearson et al., 2022).

Models are powerful tools for exploring the Indian Ocean's response to climate variability and anthropogenic changes, identifying the processes at play, and assessing the impacts on biogeochemistry and ecosystems (e.g., Sengupta et al., 2001; Rahaman et al., 2014; Lachkar et al., 2018, 2019; Resplandy et al., 2011, 2012; Schmidt et al., 2021; Ditkovsky et al., 2023; Sunanda et al., 2024). Yet, global ocean and Earth system models are plagued by strong biases in the circulation and biogeochemical dynamics in the Indian Ocean (Séférian et al., 2020; Rixen et al., 2020; Li et al., 2016). In particular, global models tend to misrepresent the circulation that regulates the exchanges between the Indian Ocean and the Pacific Ocean (i.e., the Indonesian throughflow), the overflows from marginal seas (Red Sea and Persian Gulf; Lachkar et al., 2019; Schmidt et al., 2021; Ditkovsky et al., 2023), as well as the mesoscale features (eddies and filaments) key to the ocean circulation, biological production, and the supply of nutrients and oxygen in the Indian Ocean (e.g., Wirth et al., 2002; Resplandy et al., 2011, 2012; Nuncio and Kumar, 2012; Vic et al., 2014; Lachkar et al., 2016; Greaser et al., 2020; Vinayachandran et al., 2021). These shortcomings of global models strongly limit our ability to evaluate the biogeochemical and ecosystem response to climate variability and change. It is with these applications in mind that we configured, customised and validated the regional Indian Ocean simulation based on the Modular Ocean Model 6 (MOM6, Adcroft et al., 2019) coupled with the Carbon, Ocean, Biogeochemistry, and Lower Trophics module version 2.0 (COBALTv2, Stock et al., 2014, 2020). The model configuration, called MOM6-COBALT-IND12 version 1 (or MOM6-COBALT-IND12 v1.0), covers the northern Indian Ocean at a horizontal resolution of 1/12° and is designed for physical-biogeochemical studies as well as applications to ecosystems, marine resources and management (Figure 1).

In the following sections, we first present the model physical and biogeochemical configuration (section 2) and the data and metrics used to assess the model (section 3). We then evaluate key monsoon-driven seasonal patterns (section 4), ocean interior ventilation and oxygen minimum zones (OMZs) distribution (section 5), as well as intraseasonal and interannual variability (sections 6 and 7) simulated in the model. Finally, we discuss the main strengths and limitations of the model configuration (section 8).

## 2 Regional Indian Ocean configuration

In this section, we describe the regional model configuration MOM6-COBALT-IND12 v1.0 (called MOM6-COBALT-IND12 in the following), which couples an physical ocean model with a biogeochemical module.

## 2.1 Physical ocean model configuration

The Indian Ocean regional model is based on the Geophysical Fluid Dynamics Laboratory (GFDL) ocean-ice model MOM6 (Adcroft et al., 2019). In the horizontal, the model uses an Arakawa C-grid (Arakawa and Lamb, 1977). The regional configuration MOM6-COBALT-IND12 covers the Arabian Sea and Bay of Bengal and extends to the equatorial Indian Ocean ending south of Java with one open boundary (32°E to 114°E and 8.6°S to 30.3°N; Figure 1). The horizontal resolution is 1/12° ($486 \times 984$ tracer points on the horizontal), with the horizontal grid spacing varying from 9.2 km at the equator to 7.3 km at 30°N. This resolution resolves the first baroclinic radius of deformation with at least 2 grid points and is smaller than the third baroclinic radius of deformation ($R_3 \geq 13$ km) everywhere in the domain except in the Persian Gulf and on the coastal shelf along the eastern Arabian Sea (Chelton et al., 1998; Hallberg, 2013). MOM6-COBALT-IND12 is therefore considered an 'eddy resolving' model for the region with a rectilinear and orthogonal grid (32°E to 114°E and 8.6°S to 30.3°N).

In the vertical, the model includes a 75-layer hybrid z*-isopycnal coordinate system with a z* layers near the surface (about 2 m thick in the upper 20 m in the tropical Indian Ocean) and modified potential density layers below (identical to the hybrid z*-isopycnal coordinate developed in Adcroft et al., 2019, see Figure 2). The model bathymetry was generated using the General Bathymetric Chart of the Oceans version 2020 (GEBCO; Weatherall et al., 2015) by averaging the GEBCO bathymetry (provided at a resolution of 15 arcsec) over each grid cell. The depths of the channel connecting the Red Sea bottom waters and the Arabian Sea (region in 12.5-14.2°N, 42.375-43.375°E) are set to 220 m to allow the outflow. The shallowest bathymetry in the model is 4 m. The model is integrated in time using a split explicit method (Runge-Kutta second-order scheme; Hallberg and Adcroft, 2009). The baroclinic time-step is 600 seconds and the thermodynamic and biogeochemical time-step are 1800 seconds (Table 1). Using an 18-node setup with 40 cores per node, which distributes the 486 x 984 model grid across available processing units, the model can run one year of simulation in about 16 hours of wall clock time (this includes the output of extensive diagnostics).

The configuration of subgrid-scale parameterizations used in MOM6-COBALT-IND12 are based on that of the GFDL Ocean Model version 4 (OM4; Adcroft et al., 2019). We use a background kinematic viscosity and a background diapycnal diffusivity of $1.5 \times 10^{-5}$ m$^2$ s$^{-1}$ (Table 1). As in OM4, viscosity beyond background levels is evaluated as the maximum of a Smagorinsky and resolution-dependent biharmonic viscosity (Griffies and Hallberg, 2000). Additional mixing is represented by planetary boundary layer mixing (Reichl and Hallberg, 2018; Reichl and Li, 2019), shear mixing (Jackson et al., 2008), and mixed-layer restratification due to submesoscale processes (Fox-Kemper et al., 2011). MOM6-COBALT-IND12 also includes bottom boundary layer mixing as in OM4, but the mixing efficiency parameter of this scheme is lowered from 0.2 in OM4 to 0.01 following Ross et al. (2023). The model explicitly resolves barotropic tidal forcing (see next section) and low-mode internal tides are well resolved at 1/12° resolution; however, we parameterize the local dissipation of high-mode internal tides according to topographic roughness data (St. Laurent et al., 2002; Polzin, 2009). See Table 1 for a list of configuration parameters.

**Table 1.** Major parameters and associated values used in the physical ocean (MOM6) component of the model

| Parameter | Value | Reference |
|---|---|---|
| Vertical coordinate | 75 layer hybrid z*-isopycnal | Adcroft et al. (2019) |
| Baroclinic time step | 600 s | |
| Thermodynamic and BGC time step | 1800 s | |
| Planetary boundary layer parameterization | ePBL | Reichl and Hallberg (2018) |
| Submesoscale eddy front length | 500 m | Fox-Kemper et al. (2011) |
| Biharmonic viscosity | Maximum of Smagorinsky and resolution-dependent viscosities | Griffies and Hallberg (2000) |
|    Smagorinsky coefficient | 0.06 | |
|    Resolution-dependent | $0.01\Delta_x^3\,\mathrm{m^4 s^{-1}}$ | Adcroft et al. (2019) |
| Bottom boundary layer mixing efficiency | 0.01 | Legg et al. (2006) |
| Background kinematic viscosity | $1.5 \times 10^{-5}\,\mathrm{m^2 s^{-1}}$ | |
| Background diapycnal diffusivity | $1.5 \times 10^{-5}\,\mathrm{m^2 s^{-1}}$ | |
| Boundary Conditions | | |
|    Sea level and barotropic velocities | Flather scheme | Flather (1976) |
|    Baroclinic velocities | Radiation scheme and nudging (3-day inflow and 360-day outflow) | Orlanski (1976) and Marchesiello et al. (2001) |
|    Temperature and salinity | Reservoirs with 9 km length scale | Ross et al. (2023) |
|    Biogeochemical tracers | Reservoirs with 9 km outflow length scale and 300 km inflow length scale | |
| Tidal SAL coefficient | 0.094 | Irazoqui Apecechea et al. (2017) Stepanov and Hughes (2004); Barton et al. (2022) |
| Opacity Scheme | 3-band with chlorophyll | (Manizza, 2005) |
| Piston velocity for SSS relaxation | $0.1667\,\mathrm{m\,d^{-1}}$ | Adcroft et al. (2019) |

## 2.2 Physical ocean model forcing

### 2.2.1 Initial state, spin-up and atmospheric forcing

The ocean model was initialized using temperature and salinity from annual mean fields from the World Ocean Atlas version 2013 (WOA13; Locarnini et al., 2014; Zweng et al., 2014). Our simulations were run using the atmospheric forcing from the 1/4° horizontal resolution European Center for Medium-range Weather Forecasts reanalysis 5th generation (ERA5) at 1-hour frequency (Hersbach et al., 2020). In the ocean model, air–sea heat fluxes were computed using the bulk algorithm of Large and Yeager (2004), which requires atmospheric input variables referenced at 10 m. As the ERA5 forcing provides near-surface

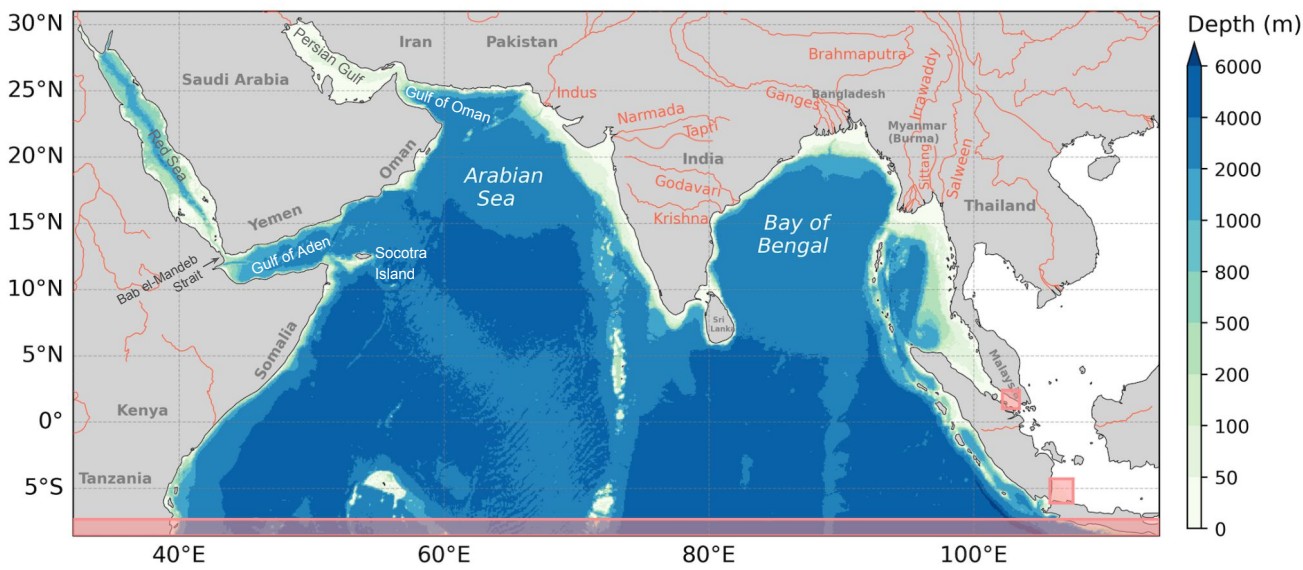

**Figure 1.** Domain and bathymetry of the regional Indian Ocean MOM6-COBALT-IND12. Pink shading indicates the extent of sponge layers (see Section 2.2.2). Major rivers are indicated in red. Socotra Island and the Bab-el-Mandeb Strait are labeled on the map.

temperature and humidity at 2 m, these variables were vertically adjusted to 10 m following the procedure recommended by Large and Yeager (2004), ensuring consistency with the algorithm's assumptions. The sea surface salinity (SSS) was restored to the polar science center hydrographic climatology (PHC2.1), which is based on the World Ocean Atlas 98 with data replenishment in the Arctic Ocean (Steele et al., 2001), with a piston velocity of 0.1667 m d$^{-1}$. We conducted a 32-year spin-up, consisting of four consecutive 8-year loops of the 1980 to 1987 forcing field, and reached a well-equilibrated state with minimal linear trends of physical and biogeochemical variables (e.g., drift in SST, SSS, oxygen, nitrate, primary production and ocean surface partial pressure of carbon dioxide $p$CO$_2$ $< \sim$0.1% for spin-up years 17-32). Using outputs from the end of the spinup simulation as initial conditions, the hindcast simulation was started on 1 January 1980 and was run from 1980 to 2020 for our analysis in this study.

### 2.2.2 Open boundary conditions and tidal forcing

Open boundary conditions (OBC) are set using the Flather formulation for the tidal and sub-tidal sea level and barotropic velocity and the Orlanski formulation for the baroclinic velocity (Flather, 1976; Orlanski, 1976). In addition, we nudge the boundary values towards external forcing with a strong 3-day time-scale for baroclinic normal and tangential velocities en-

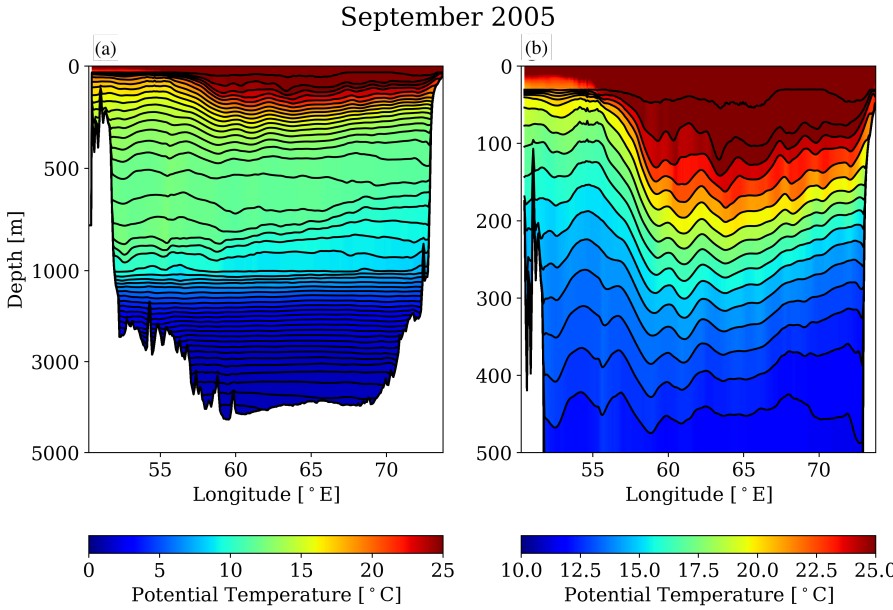

**Figure 2.** West-east cross-section of the Arabian Sea at 15°N showing the structure of the isopycnal vertical coordinate (contours) overlain with potential temperature (colors) in September 2005 (a) over the full column and (b) over the top 500 m. The z* layers in the upper ocean are not shown. The coordinate follows the pattern of the wind-driven upwelling along the coast of Yemen in the west, and the coastal Kelvin wave-driven upwelling along the Indian coast in the east.

tering the model and a weak 360-day time-scale for outgoing velocities (Marchesiello et al., 2001). The boundary value for temperature and salinity are set using a reservoir in which the properties are evolving based on contributions from an inflow (properties outside of the domain set by an boundary forcing file) and outflow (properties simulated inside the model domain) fluxes. Similarly to Ross et al. (2023), the inflow and outflow length scales are set to 9 km (about 1-10 day time-scale for velocities of $10^{-1}$ cm s$^{-1}$) for temperature and salinity (i.e., inflow and outflow have an equal contribution to the boundary reservoir). The model includes a sponge layer over 15 grid points at the southern open boundary, nudging the model to time-varying Ocean Reanalysis System 5 (ORAS5) temperature and salinity with a time-scale increasing from 12 days at the boundary to 174 days at the 15th grid point. The model also includes two sponge layers at the closed boundaries of the Malacca and Sunda Straits with a nudging to the climatological WOA18 data. For the Malacca Strait, temperature and salinity are nudged over 15 grid points with a time-scale increasing from 12 days at the strait outlet to 174 days toward the Indian Ocean. In the Sunda Strait, the nudging is over 21 grid points and the time-scale increases from 12 days at the outlet to 336 days toward the Indian Ocean.

Ten tidal components (i.e., M2, S2, N2, K2, K1, O1, P1, Q1, Mm, and Mf) interpolated from the inversion of TOPEX/-POSEIDON crossover data TPX09 (Egbert and Erofeeva, 2002) are used to generate surface elevation and velocity forcing at the open boundary. Tidal potential forcing from the same ten components is included in the barotropic momentum equations throughout the domain, and the effects of self-attraction and loading are represented using the scalar approximation (Accad

and Pekeris, 1978) with a coefficient of 0.094. Sub-tidal velocities, temperature and salinity at the southern open boundary are from the monthly ORAS5 (Zuo et al., 2019).

### 2.2.3 River freshwater discharge

Freshwater discharge from rivers was prescribed using the gridded daily Global Flood Awareness System (GloFAS) reanalysis version 4.0, as described by Grimaldi et al. (2022) and Harrigan et al. (2023). To map the river discharge data onto the MOM6-COBALT-IND12 grid, we used the GloFAS local drainage direction map to identify outlet points adjacent to the coast, as well as any chains of outlet points connected to these coastal outlets, see details in Burek et al. (2013). The streamflow at these outlet points was introduced at the surface of the nearest model coastal ocean grid cell. To ensure the riverine freshwater flux is mixed into the water column, an extra input of turbulent kinetic energy extending down to a depth of 10 meters was included at the discharge points (Tseng et al., 2016). By comparing GloFAS to published discharge observations (Jian et al., 2009; Siswanto et al., 2023), we found that GloFAS overestimated discharge in the Ganges-Brahmaputra river system, and therefore scaled down the freshwater discharge by 25% to match observations in these two rivers (see Appendix Figure A1; Jian et al., 2009; Siswanto et al., 2023). Additionally, we found that GloFAS underestimated runoff in the Irrawaddy-Sittang river system. To correct for this bias, we applied a linear regression-based correction (see Appendix Figure A1) between the original GloFAS discharge and discharge data from the Global Runoff Data Centre (GRDC; Recknagel et al., 2023) for the Irrawaddy-Sittang regions. Finally, we manually removed discharge in the model sponge layers of the Sunda Strait and Malacca Strait.

### 2.3 Biogeochemical model configuration and changes specific to Indian Ocean

The physical ocean model is coupled to the COBALT v2 (Stock et al., 2014, 2020). COBALTv2 represents 33 tracers including nutrients (nitrate, phosphate, silicate, and iron), three phytoplankton groups (small, large, diazotrophs), three zooplankton groups (small, medium, large), three dissolved organic carbon pools (labile, semi-refractory and refractory), one particulate detritus pool, oxygen, and carbonate system.

Several parameters of the standard COBALTv2 model from Stock et al. (2020) were modified to match observational constraints and characteristics of the Indian Ocean and improve model biases, including a bias in the extent and volume of the OMZ in the Bay of Bengal.

- Detritus sinking velocity was increased from 100 to 120 m $d^{-1}$, based on *in-situ* sediment trap observations indicating sinking speeds up to 160-280 m $d^{-1}$ in the Indian Ocean (Rixen et al., 2019b).

- The burial fraction was increased (the equivalent half-saturation in the denominator of Equation 3 from Dunne et al., 2007, was reduced from 7 to 1 mmolC $m^{-2}$ $d^{-1}$) . This increased the burial of particulate organic carbon from 0.013 PgC $y^{-1}$ to 0.026 PgC $y^{-1}$ in the tropical Indian Ocean, in better agreement with the burial of 0.028 PgC $y^{-1}$ found in the observation-based reconstruction of LaRowe et al. (2020).

- The oxygen half-saturation for nitrification ($k_{nit,O2}$ in Stock et al., 2020) was reduced from 3.9 to 2.0 $\mu$mol $O_2$ $kg^{-1}$, based on recent observations indicating a lower oxygen threshold for ammonium oxidation in the OMZs (Bristow et al., 2016; Peng et al., 2016; Frey et al., 2023).

- The oxygen constraint on water column denitrification was modified from $O_{2,min}/(k_{O2}+O_{2,min})$ when $O_2 < 0.8$ $\mu$mol $kg^{-1}$ (see Appendix A3 in Stock et al., 2020) to $O_2/(k_{O2}+O_2)$ when $O_2 < 4.0$ $\mu$mol $kg^{-1}$, in line with findings that the oxygen threshold below which denitrification starts is typically between 4 and 5 $\mu$mol $kg^{-1}$ (Paulmier and Ruiz-Pino, 2009).

## 2.4 Biogeochemical model forcing

### 2.4.1 Initial state, open boundary conditions and model drift

For the model spin-up, nutrients (nitrate, phosphate, and silicate) and oxygen were initialized using annual means from the World Ocean Atlas 2018 (WOA18; Garcia et al., 2019). Dissolved inorganic carbon (DIC) and alkalinity were initialized using annual means from the Global Ocean Data Analysis Project version 2 (GLODAPv2), which are representative of year 2002 (Olsen et al., 2016). Other biogeochemical tracers were initialized with very low seed values of $10^{-10}$. This initial value has a negligible impact on the solution as most of these remaining tracers have turnover time-scales much shorter than the 32-year spin-up duration (e.g., typically of a few days for phytoplankton), except semi-refractory dissolved organic matter (decay time-scale of 10 years). Atmospheric $CO_2$ forcing was taken from the global carbon budget project (Friedlingstein et al., 2022). Biogeochemical boundary values are prescribed from WOA18 monthly climatologies for nitrate, phosphate, silicate, and oxygen. For DIC and alkalinity boundary values, annual mean fields were estimated using the Empirical Seawater Property Estimation Routines (ESPER) MATLAB code (Carter et al., 2021), based on annual mean temperature and salinity from ORAS5. The OBC for biogeochemical tracers is set using the reservoir scheme (see section 2.2), with an outflow length scale of 9 km but an increased inflow length scale of 300 km, giving more weight to the solution within the model domain. This decoupling between contributions from the inflow and outflow limits the influence of the boundary external forcing on the model domain, specifically when the fields at the boundaries are poorly constrained such as for biogeochemical tracers. Model drift after the 32-year spin-up and over the 41 years of a hindcast simulation with constant forcing is small, with linear trends $< 0.05\%$ for oxygen, nitrate, DIC, alkalinity, semi-refractory dissolved organic nitrogen pools and integrated primary productivity (Appendix Figure A2). The slight drift indicates that the hindcast simulation starts from a well-equilibrated initial state provided by the spin-up simulation.

### 2.4.2 Atmospheric deposition

The model is forced with monthly atmospheric deposition of nitrogen (wet and dry deposition of nitrate and ammonium), iron, phosphorus, and lithogenic dust derived from the archived GFDL Earth system model version 4.1 (ESM4.1) historical

simulation[1] (1980–2014) and Shared Socioeconomic Pathways 5-8.5 (SSP5-8.5) scenario[2] (2014–2020; Stock et al., 2020; Horowitz et al., 2020; Paulot et al., 2020). ESM4.1 includes interactive modules for anthropogenic and natural (e.g., biomass burning, lightning) reactive nitrogen emissions, photochemical reactions, removal of nitrogen by wet and dry deposition, as well as a land-atmosphere-ocean cycling of dust and ocean ammonia outgassing (Paulot et al., 2020; Horowitz et al., 2020). Interannual variability in ESM4.1 is not in phase with observed variability (as for any coupled Earth system model). For dry and wet deposition of oxidized and reduced nitrogen, we therefore used a 15-year moving by month average (e.g., January 2000 is an average of all Januaries between years 1993 and 2007) that retain the seasonality and the decadal anthropogenic increase in deposition but removed the interannual variability (see Appendix Figure A3). For iron, phosphorus and lithogenic material deposition, we used monthly mean climatologies over the 1950-2022 period (ESM4.1 does not include the effects of fossil fuel burning etc. that would yield a significant long term trend in these fields, although it would include the smaller impact of long-term wetting / drying, wind and/or precipitation trends that we ignore here). Iron and dry lithogenic dust depositions are from ESM4.1 outputs. Phosphorus deposition was evaluated using the ESM4.1 climatology in dry lithogenic dust deposition, assuming a phosphorus content of 563 ppm in dust, of which 22% is bioavailable (see Herbert et al., 2018; Ross et al., 2023). See details about the influence of atmospheric deposition in this model in Malsang et al. (2024).

### 2.4.3 River biogeochemical inputs

The riverine fluxes of dissolved and particulate nutrients (nitrogen and phosphorus) are derived from the annual mean loads of inorganic and organic nitrogen and phosphorus from the Global Nutrient Export from WaterSheds2 (GlobalNEWS2), referenced to the year 2000 (Mayorga et al., 2010). We include riverine inputs of dissolved inorganic nitrogen (DIN), dissolved inorganic phosphorus (DIP), dissolved organic nitrogen (DON), dissolved organic phosphorus (DOP), and bio-available particulate organic nitrogen (PON). We do not include bio-available particulate organic phosphorus (POP) as the river input of DIP is already likely too high in GlobalNEWS2 (Jiao et al., 2023). DON and DOP are distributed among different dissolved organic pools, with 30% allocated to the labile pool, 35% to the semi-labile pool, and 35% to the semi-refractory pool (Wiegner et al., 2006). The riverine PON is assumed 100% bio-available.

The riverine input of iron is set at a value of 70 nmol kg$^{-1}$ based on Raiswell and Canfield (2012). In the Bay of Bengal (78°E-103°E) region, the riverine DIN concentration is reduced by 80% based on coastal nitrate data collected by Krishna et al. (2016). This adjustment is supported by Zhou et al. (2022) and Jiao et al. (2023), which compared several global nutrient transport models highlighting that GlobalNEWS2 tended to overestimate total nitrogen riverine inputs. The riverine flux of DIN in the Arabian Sea and the flux of other nutrients in both the Arabian Sea and Bay of Bengal are kept equal to the original values from GlobalNEWS2. The riverine inputs of DIC (0.32 mol m$^{-3}$) and alkalinity (0.42 mol equivalents of alkalinity m$^{-3}$) are assigned constant concentrations, consistent with those used in the GFDL-ESM4.1 Earth system model (Stock et al., 2020).

To reflect spatial differences in sediment supply, we specify riverine lithogenic concentrations based on observational data from Milliman and Farnsworth (2011). The lithogenic input from rivers was adjusted to 200 g m$^{-3}$ for major rivers (i.e., rivers

---

[1]https://www.wdc-climate.de/ui/cmip6?input=CMIP6.CMIP.NOAA-GFDL.GFDL-ESM4.historical

[2]https://www.wdc-climate.de/ui/cmip6?input=CMIP6.ScenarioMIP.NOAA-GFDL.GFDL-ESM4.ssp585

with sediment loads exceeding 10 Mt y$^{-1}$, e.g., Godavari, Krishna, Ganges, Brahmaputra, Irrawaddy, Sittang, Salween, Indus, Tapti and Narmada rivers; see Figure 1 for rivers location) and 20 g m$^{-3}$ for all other rivers, rather than applying a global constant of 13 g m$^{-3}$ used for all rivers as in Stock et al. (2020). These adjustments account for the significantly higher total suspended sediment loads in these rivers (Milliman and Farnsworth, 2011; Rixen et al., 2019b), and are supported by river observations from Milliman and Farnsworth (2011) showing a broad range from 10 g m$^{-3}$ (Muvattupuzha River) to 1,061 g m$^{-3}$ (Ganges River). In the model, this higher lithogenic flux protects more particulate organic matter from remineralization, thereby increasing organic carbon export to the deep ocean and reducing oxygen consumption in the subsurface. This is in line with observations that underscore the significant role of lithogenic matter in reducing organic matter remineralization and accelerating carbon export in the northern Indian Ocean (Rixen et al., 2019b).

These concentrations of nutrients, DIC, alkalinity, lithogenic and organic material (constant in time) are incorporated using the GloFAS freshwater inputs and by assigning them to the nearest neighboring river mouths, with larger rivers given priority over smaller ones. Nutrient loads vary in accordance with changes in river discharges, and the baseline configuration presented in this study does not account for the fluctuations and trends in observed nutrient concentrations during the 1980-2020 model simulation period.

## 3    Methods for assessing model spatial and temporal variability

### 3.1    Physical and biogeochemical datasets

We used satellite and *in-situ* observations to assess modeled physical and biogeochemical basin-scale patterns as well as seasonal, interannual and intraseasonal variability. See Table 2 for a list of all datasets and their references.

For the basin-scale evaluation of physical fields, we used Argo gridded temperature (Roemmich and Gilson, 2009), temperature and salinity from the WOA18 (Garcia et al., 2019), satellite-based SST from the Optimum Interpolation SST (OISST) version 2.1 (Banzon et al., 2016), sea surface height (SSH) and sea level anomaly from AVISO and distributed by the Copernicus Marine and Environment Monitoring Service (CMEMS; http://www.marine.copernicus.eu), the mixed layer depth (MLD) climatology from De Boyer Montégut et al. (2004; updated in November 2008; https://mld.ifremer.fr/Surface_Mixed_Layer_ Depth.php) and ocean surface currents from the OSCAR drifter database (ESR, 2009). In addition, we used data from the Research Moored Array for African-Asian-Australian Monsoon Analysis and Prediction (RAMA), specifically from two moorings capturing the east-west contrast in the basin at 57°E, 4°S and 95°E, 5°S (data downloaded from the Pacific Marine Environmental Laboratory NOAA website; McPhaden et al., 2009), and observations from water mass properties at the Red Sea outflow from Sofianos et al. (2002).

For the basin-scale biogeochemical model evaluation, we used oxygen concentrations from WOA18 (Garcia et al., 2019) and from Bianchi et al. (2012), surface chlorophyll data from the European Space Agency ocean color climate change initiative (OC-CCI version 5.0; Sathyendranath et al., 2019), vertical chlorophyll data from bio-Argo (Wong et al., 2020), and integrated primary productivity from the satellite-based Carbon-based Production Model (CbPM) algorithm, the Carbon, Absorption, and Fluorescence Euphotic-resolving (CAFE) algorithm, the Vertically Generalized Production Model (Standard-

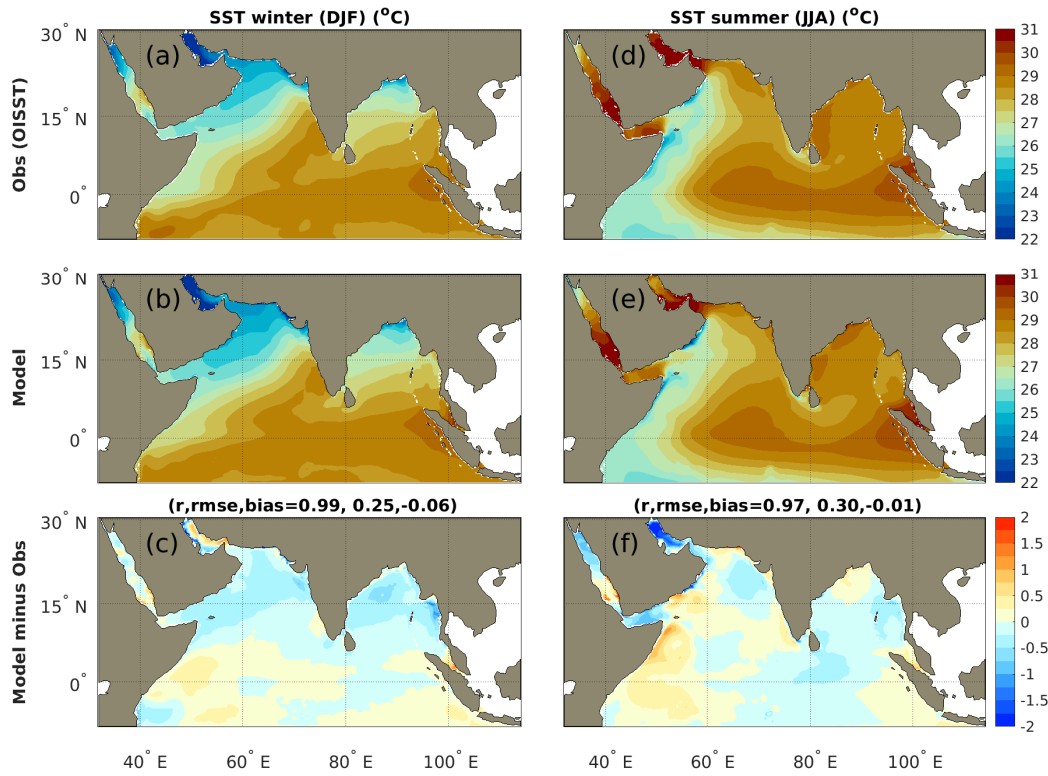

**Figure 3.** Sea surface temperature (SST) during (a-c) winter (December-February) and (d-f) summer (June-August) monsoons. Panels (a,d) show OISST observations, (b,e) show MOM6-COBALT-IND12 model and (c,f) show differences between model and observations. Correlation coefficients r, RMSE and bias between the observed and model seasonal means are indicated. See details on observations in Table 2. Model results are averaged over the 1980-2020 period.

VGPM) algorithm and its alternative formulation (Eppley-VGPM), all accessed via the Ocean Productivity website (http: //sites.science.oregonstate.edu/ocean.productivity/index.php). In addition, we used *in-situ* observations compiled from a literature review including 24 studies and 351 stations (see Table 2 for references). River inputs and particulate organic and lithogenic matter in the model were evaluated using river discharge from the Global Runoff Data Center (GRDC; Recknagel et al., 2023).

## 3.2    Analysis and evaluation metrics

We evaluated the amplitude of intraseasonal variability (ISV) using SSH temporal variability as a proxy for mesoscale eddies and planetary waves (Rossby and Kelvin waves; e.g., Cheng et al., 2013). Observed and simulated SSH were detrended using a linear regression and filtered using a 14-120 days band pass filter to remove the seasonal cycle, interannual variability and long-term trend, and only retain the intraseasonal timescales. The Dipole Mode Index (DMI) used to evaluate IOD phases was

calculated as the SST anomalous gradient between the western equatorial Indian Ocean (50°E-70°E and 10°S-10°N) and the southeastern equatorial Indian Ocean (90°E-110°E and 10°S-0°N, Saji et al., 1999). Finally, we used three metrics throughout the study to compare model results and observations: the Pearson correlation coefficient (r) which measures the correlation between observations and model in time (for time-series) or in space (for maps), the root mean square error (RMSE, i.e., quadratic mean of model minus observations) which measures the model accuracy compared to observations, and the bias (i.e.,

model minus observations) which indicates if the model underestimates or overestimates the observed fields. For the validation of climatological annual and seasonal means, model outputs are averaged over the period 1980–2020. Observation-based data products are treated based on their availability: if the dataset provides climatological means (annual or seasonal), we use the provided values directly. If not, we compute climatological means over the available time span of the observational dataset ( see time span in Table 2).

## 4  Monsoon-driven seasonality

### 4.1  Sea surface temperature as an indicator of seasonal dynamics

Patterns of SST in the northern Indian Ocean follow the well described basin-scale features associated with the monsoon reversal (e.g., Schott and McCreary, 2001). MOM6-COBALT-IND12 captures seasonal SST patterns well, notably the contrast between the vast warm pool (SST >28°C) that extends over most of the basin and the regions with colder SSTs that develop

in response to seasonal variations in atmospheric and oceanic circulation (Figure 3). During the winter monsoon, the model simulates the relatively cold water (SST <26°C) associated with evaporative cooling in the northern Bay of Bengal, and a combination of evaporative cooling and convective mixing (MLD of 40-60 m) in the northern Arabian Sea (Figures 3a-c and 4a-c). During the summer monsoon, the model simulates the colder summer SSTs observed in wind-driven upwelling regions along the western boundary coasts (e.g., Oman, Yemen, Somalia, Kenya and Tanzania where SST <26°C), and in the weaker

upwelling controlled by Kelvin wave propagation along the southwestern Indian coast (SST ∼27°C; Figure 3d-f; see details on wave propagation in section 4.3). At the basin scale, modeled SST patterns shows strong agreement with observed patterns, characterized by a high correlation coefficient (r>0.97), low RMSE (0.25-0.3°C), and small biases (regional mean SST bias of -0.06°C in winter and -0.01°C in summer for the 1980-2020 period). We note that the good agreement between observed and modeled SST is in part attributable to the strong influence of the prescribed observation-driven atmospheric surface boundary

forcing that controls air-sea heat fluxes in the model (e.g., temperature, wind; see section 2). In addition, a comparison between ERA5 and Cross-Calibrated Multi-Platform (CCMP) wind products demonstrates that ERA5 wind forcing effectively captures the seasonal cycle and spatial distribution of the summer and winter monsoons (Appendix Figure A4).

The model captures the seasonal contrast in MLD between the Arabian Sea and the Bay of Bengal, with deeper mixed layers in the Arabian Sea and shallower layers in the Bay of Bengal during both winter and summer (Figure 4). The MLD is generally

deeper in summer than in winter. The spatial patterns, including the locations of local MLD maxima, are broadly consistent with observational data. Quantitatively, the basin-wide correlation values are similar between the two seasons, although the RMSE is larger in summer (8.09 m) than in winter (7.00 m). One possible contributor to the larger summer bias is the enhanced

**Table 2.** Observational products used to evaluate MOM6-COBALT-IND12

| Parameter | Sampling frequency | Reference dataset |
|---|---|---|
| Sea surface temperature | monthly optimum interpolation (1982-2020) | OISSTv2.1 includes satellites, ships, buoys, Argo floats (Banzon et al., 2016) |
| Mixed-layer depth | monthly climatology | De Boyer Montégut et al. (2004) - updated Nov. 2008 |
| Surface currents | 5-day averaged monthly | OSCAR drifter database (ESR, 2009) |
| Sea level anomaly | daily satellite-based | Copernicus (Lopez, 2018) |
| Ocean temperature and salinity | monthly climatologies / in-situ profiles / in-situ profiles | World Ocean Atlas 2018 (WOA18, Garcia et al., 2019) / World Ocean Database 2018 (WOD18, Boyer et al., 2018) / RAMA moorings (McPhaden et al., 2009) |
| Wind speed | monthly satellite / in-situ | CCMP (Mears et al., 2022) / RAMA moorings (McPhaden et al., 2009) |
| Red Sea Outflow properties | in-situ sampling (1995-1996) | Sofianos et al. (2002) |
| Oxygen concentration | monthly climatologies | WOA18 (Garcia et al., 2019) and Bianchi et al. (2012) |
| Surface chlorophyll | monthly climatology | OC-CCI v5.0 (Sathyendranath et al., 2019) |
| River Discharge | daily/annual mean | Global Runoff Data Center (GRDC, Recknagel et al., 2023) Jian et al. (2009), Krishna et al. (2016) |
| Riverine lithogenic flux | in-situ sampling | Milliman and Farnsworth (2011) |
| Marine lithogenic/organic flux | in-situ sampling | Rixen et al. (2019b) |
| Net primary productivity | monthly satellite-based | CbPM (Westberry et al., 2008), CAFE (Silsbe et al., 2016), standard-VGPM, Eppley-VGPM (Behrenfeld and Falkowski, 1997) |
| | in-situ sampling (351 stations) | Saxena et al. (2023); Marra et al. (2021); Sarma et al. (2020) Löscher et al. (2020); Sarma and Dalabehera (2019) Ahmed et al. (2017); Gandhi et al. (2010, 2011) Kumar et al. (2010); Naqvi et al. (2010); Prakash et al. (2008) Prasanna Kumar et al. (2007a, b); Naqvi et al. (2006) Gauns et al. (2005); Kumar et al. (2004) Barber et al. (2001); Watts and Owens (1999); Watts et al. (1999) Savidge and Gilpin (1999); McCarthy et al. (1999) Veldhuis et al. (1997); Devassy et al. (1983) Bhattathiri et al. (1980); Radhakrishna (1978) |

wind forcing during the monsoon season (see Figure A4), which intensifies turbulent mixing and deepens the mixed layer. At the same time, the MOM6 model includes the mixed layer eddy (MLE) parameterization of Fox-Kemper et al. (2011), which

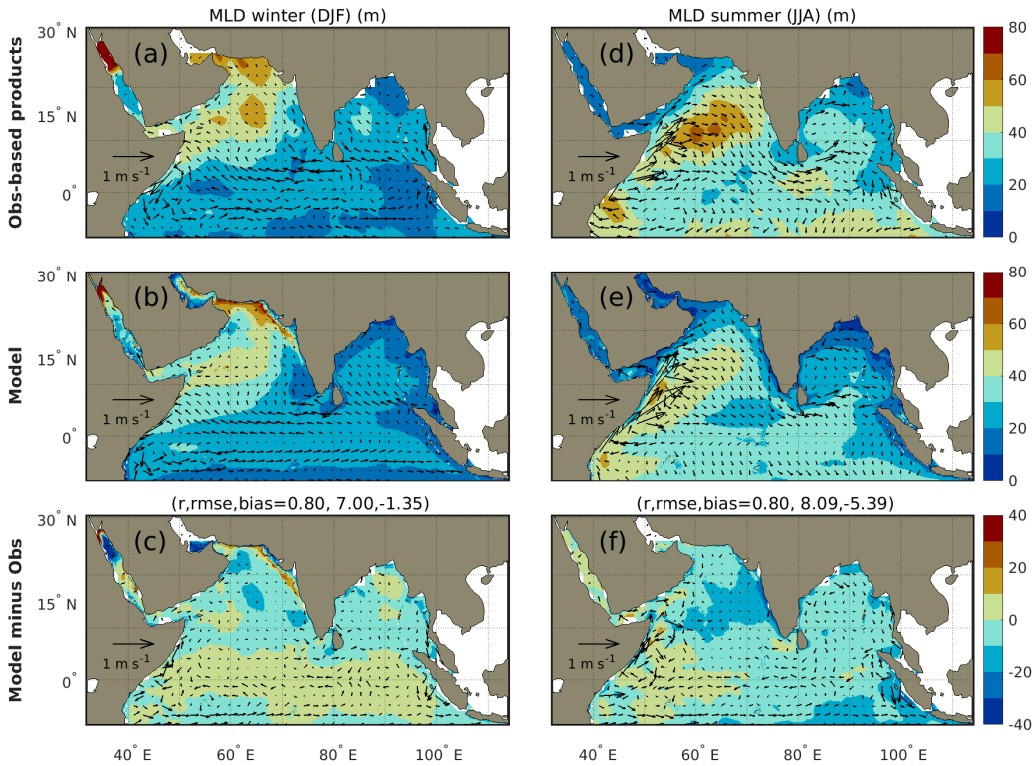

**Figure 4.** Seasonal mean mixed layer depth (MLD) and surface currents during a-c) winter (December-February) and d-f) summer (June-August) monsoons. Panels (a,d) show observations, (b,e) show MOM6-COBALT-IND12 model and (c,f) show differences between model and observations. Correlation coefficients r, RMSE and bias between the observed and model seasonal MLD means are indicated. Observations are an update of De Boyer Montégut et al. (2004) for MLD and the OSCAR drifters database for surface currents (see Table 2). Model results are averaged over the 1980-2020 period.

represents restratification driven by baroclinic eddies within the mixed layer. This restratification process may also be more active in summer, potentially leading to an overcorrection that offsets vertical mixing too strongly. The interaction between intensified wind-driven mixing and enhanced restratification may thus contribute to the larger MLD bias observed in summer compared to winter.

## 4.2 Seasonal reversal of upper ocean circulation

MOM6-COBALT-IND12 reproduces the observed seasonal reversal of the main current systems, as confirmed by comparison with the updated OSCAR drifters database (arrows on Figure 4). In the Equatorial band, these seasonal changes include the shift from an eastward transport by the Northeast Monsoon Current (Equator to 10°N) and westward transport by the South Equatorial Countercurrent (5°S to Equator) in winter, to a mostly westward transport by the the Southwest Monsoon Current

in summer (Equator to 10°N, Figure 4). MOM6-COBALT-IND12 also simulates the summer strengthening and reversal of the western boundary Somali Current system and its extension northward along the Arabian Peninsula (Figure 4). In the following, we compare the simulated and observed seasonal evolution of this western boundary system, with a focus on the characteristics that are most relevant to the biogeochemical response, and refer the reader to prior work for a more in-depth description of its dynamics (e.g., Schott and McCreary, 2001; Wirth et al., 2002; Brandt et al., 2003; Sengupta et al., 2001; Beal and Donohue, 2013; Beal et al., 2013; Vic et al., 2014; Wang et al., 2018).

Figure 5 compares the simulated and observed seasonal evolution of the western boundary system. MOM6-COBALT-IND12 simulates relatively well the observed climatological evolution of the Somali Current. Before the summer monsoon (April), the Somali Current is relatively weak and flows northward along the western boundary, crossing the Equator in both observations and model. At the onset of the summer monsoon (June), the Somali Current intensifies, and separates at around 4°N into a northward alongshore current and an eastward flow that loops back across the equator and feeds the South Equatorial Countercurrent, a feature also known as the Southern Gyre (Beal et al., 2013). Simultaneously, a quasi-stationary anticyclonic mesoscale gyre called the Great Whirl develops at about 10°N (Figure 5). As the southwest monsoon progresses (August), the Great Whirl intensifies, becoming one of the largest and most energetic coherent vortices in the world ocean. A smaller anticyclonic mesoscale eddy, the Socotra Eddy, also develops east of Socotra Island at this time (Figure 5). The structure of the Great Whirl at its peak is relatively similar in the model and shipboard and mooring observations, with an horizontal footprint of ∼500 km, a vertical extent of ∼1000 m, meridional currents of about 1 m s$^{-1}$ at the surface and 0.1 m s$^{-1}$ at 1000 m depth (Figures 5 and 6 and observations reported in Schott and McCreary, 2001; Beal and Donohue, 2013). Finally, during the fall intermonsoon (October), the gyre system decays, and by the winter monsoon (December), the surface signature of the Great Whirl and Socotra Eddy are not visible (Figure 5).

### 4.3 Coastal upwelling and downwelling

Patterns in sea level anomaly can be used as a proxy for coastal seasonal upwelling (negative anomalies) and downwelling (positive anomalies) motions (Figure 7a-d). In summer, the model reproduces the amplitude and patterns of wind-driven upwelling along the western Arabian Sea (e.g., Oman, Yemen and Somalia), and western Bay of Bengal (eastern India) coasts (Figure 7b,d; correlation coefficient r = 0.91; RMSE = 0.02 m). We note that the latter upwelling has little influence on SST in both observations and models (Figure 3) due to the strong near-surface stratification imposed by high freshwater inputs in the Bay of Bengal, and hence the strong atmospheric control on SST in this region (e.g., Shetye et al., 1991; Shenoi et al., 2002). In winter, SLA patterns largely mirror summertime patterns due to the reversal of the winds and ocean circulation, with downwelling motions (positive SLA) that develop along the western Arabian Sea coasts and the western Bay of Bengal (Figure 7a-d). This pattern is also well captured by the model (Figure 7a,c; correlation coefficient r = 0.93; RMSE = 0.02 m).

Wind-driven upwelling and downwelling are strongly modulated by the seasonal propagation of coastal Kelvin waves around the rim of the northern Indian Ocean (e.g., McCreary et al., 1993; Yang et al., 1998; Nienhaus et al., 2012; Vinayachandran et al., 2021). We examine the evolution of these coastal waves following changes in SLA along the Equatorial and coastal wave guides using the review and description provided in Pearson et al. (2022). Modeled coastal SLA patterns remarkably

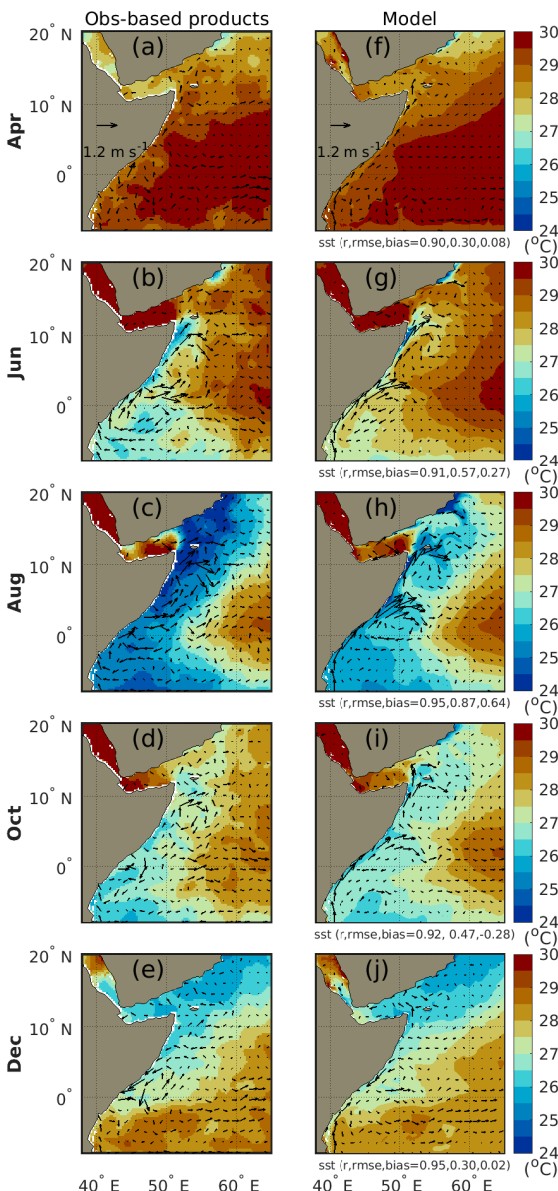

**Figure 5.** Climatological evolution of the western boundary Somali Current system showing SST (colors) and surface currents (vectors) in observation-based products (a-e) and MOM6-COBALT-IND12 (f-j). Observation-based products are from OISSTv2.1 satellite for SST and the OSCAR drifters database for surface currents (see Table 2). Correlation coefficients r, RMSE and bias between the observed and modeled SST means are indicated.

capture the timing and amplitude of the observed patterns, starting with the equatorial upwelling Kelvin waves triggered by wind changes in the summer and winter monsoons (arrows for waves I and II), and the equatorial downwelling Kelvin waves

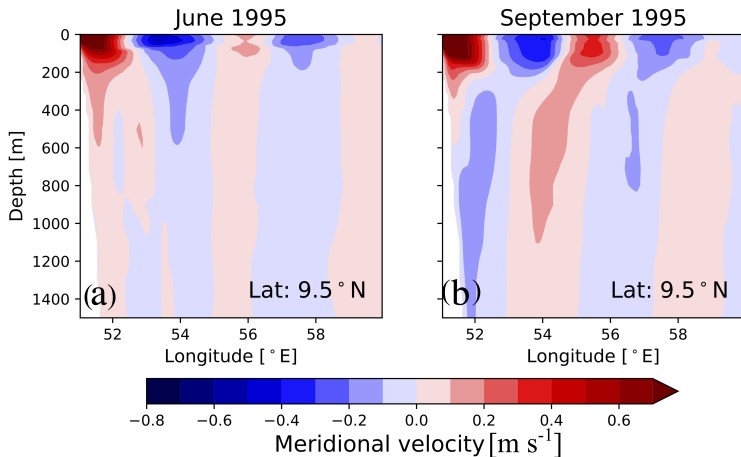

**Figure 6.** West-east depth section of meridional velocities across the Great Whirl in (a) June and (b) September 1995 in the MOM6-COBALT-IND12. These sections are comparable to observations from Beal and Donohue (2013) (see their Figure 2). Positive velocities are northward.

triggered during the spring and fall intermonsoons (arrows for waves III and IV on Figure 7e-f). These successive wave trains travel east and then counter-clockwise around the Bay of Bengal and the Arabian Sea. The model also captures the summer upwelling and winter downwelling waves excited in the northwestern Bay of Bengal (arrow for waves V and VI) and at the tip of India (arrows for waves VII and VIII), reinforcing the wind-driven summer upwelling and winter downwelling (dashed circles) that develop in the western Bay of Bengal and eastern Arabian Sea (Figure 7e-f). See further details in Pearson et al.

(2022) and references herein.

### 4.4 Sea surface salinity and river plumes

The model reproduces the main observed patterns of SSS (Figure 8a-f), including the high SSS (SSS >34 psu) in the Arabian Sea where evaporation exceeds precipitation and riverine runoff, and the much fresher (SSS <34 psu) Bay of Bengal where precipitation and runoff exceed evaporation. Performance metrics indicate that the simulation achieves a strong spatial correlation

(0.95-0.96) and a small regional RMSE (0.53-0.71). It also reproduces the seasonality of SSS associated with the monsoon, in particular the extent of the surface freshwater plumes (SSS <31psu) associated with the river discharge in the Bay of Bengal. Riverine runoff in the Bay of Bengal is lowest during the dry winter monsoon and spring intermonsoon, and peaks during the summer monsoon and early fall intermonsoon, with discharges up to $1.5 \times 10^5$ m$^3$ s$^{-1}$ in the Ganges-Brahmaputra river system and $0.4 \times 10^5$ m$^3$ s$^{-1}$ in the Irrawaddy-Sittang river systems for which we have observed time-series (Figure 8g-h). The

runoff product used to force the model reproduces the seasonality of the Ganges-Brahmaputra and the Irrawaddy-Sittang river systems (GloFAS was modified based on runoff observations in this system; see Section 2.2.3). As a result, the observed and simulated freshwater plumes are confined to the river mouths in late spring when runoff is lowest (April), and extend 200 to 500 km offshore in summer when runoff peaks (August), before being stretched out alongshore in the northern and western

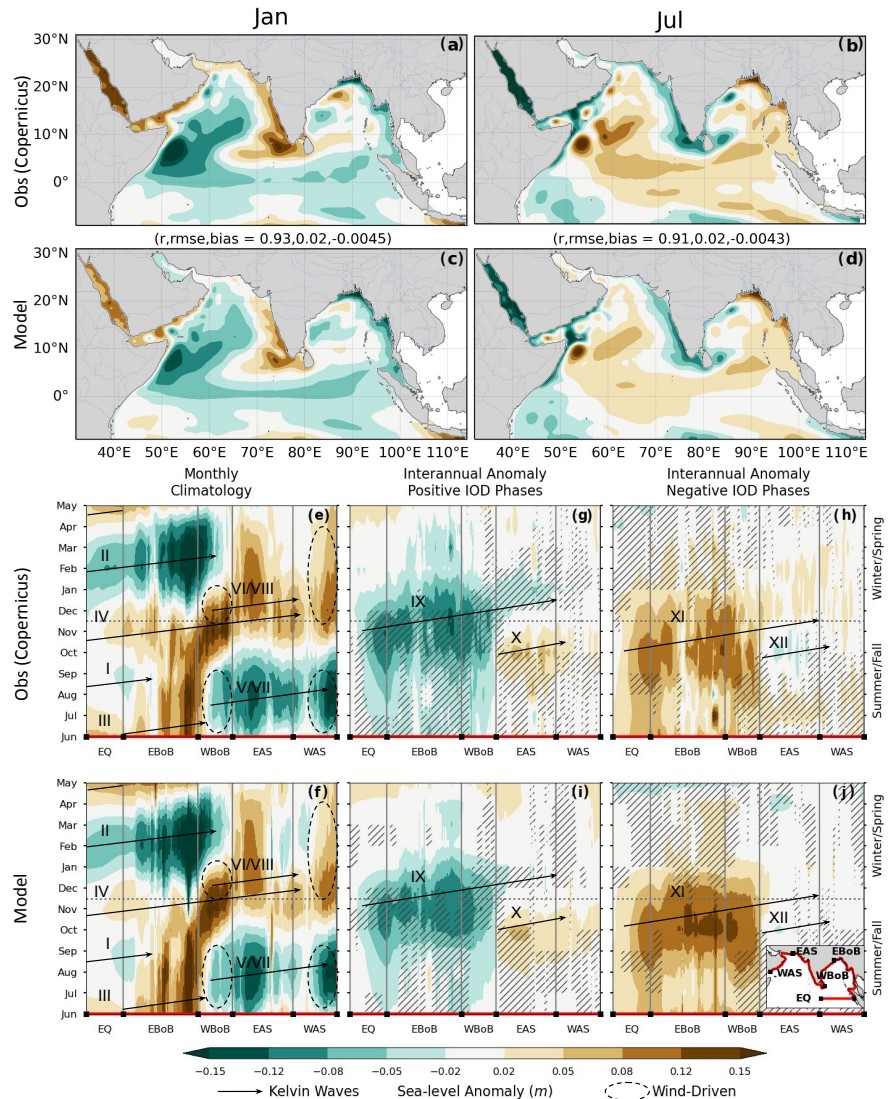

**Figure 7.** Coastal upwelling/downwelling inferred from sea level anomalies (SLA, in m) from satellite observations and MOM6-COBALT-IND12: (a-d) January and July climatological maps (1993–2020 data and model average, (e-f) Hovmüller of seasonal SLA (1993–2020 data and model average, and (g-j) Hovmüller interannual SLA (seasonal cycle removed) for positive IOD (g,i) and negative IOD (h,j) composites. In panels e-j, the x-axis follows the equatorial and coastal wave guides (red line in inset) starting at the equator (EQ), counterclockwise around the eastern and western Bay of Bengal (EBoB/WBoB) and around the eastern and western Arabian Sea (EAS/WAS). Upwelling (negative SLA) and downwelling (positive SLA) are indicated by circles when wind-driven and by arrows when wave-driven (approximate wave speed of 2.4 m/s consistent with theoretical first baroclinic mode Kelvin waves; roman numerals used in text). Unhatched/hatched regions indicate where the IOD anomaly reinforces/opposes the seasonal signal. Satellite SLA is from Copernicus (see Table 2).

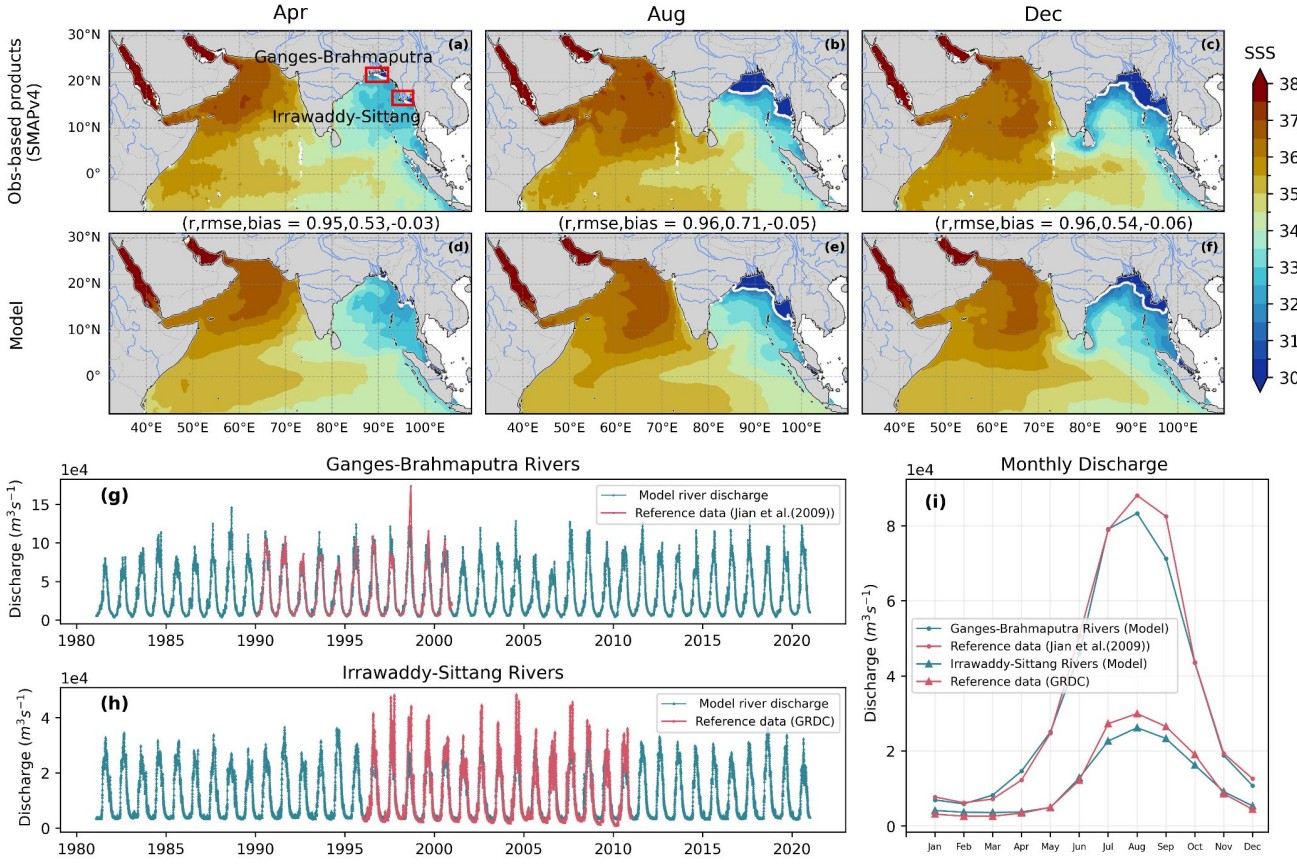

**Figure 8.** Seasonality in sea surface salinity (SSS) and river discharge: (a-f) Climatological SSS in April, August and December in the satellite SMAPv4 product and MOM6-COBALT-IND12 (2015-2019 period for both). White contour delimits waters with SSS < 31 psu. (g-i) Water discharge from observations and the modified GloFAS runoff product used to force MOM6-COBALT-IND12: (g) time-series of the Ganges-Brahmaputra river system, (h) time-series of the Irrawaddy-Sittang river system and (i) seasonal climatology for both systems over the period. See Table 2 for data source. A comparison of modified GloFAS to the raw GloFAS product is presented in Appendix Figure A1.

Bay of Bengal by horizontal transport in fall and winter (December; Figure 8c). The seasonality of SSS and the impact of
river discharge are more limited in the Arabian Sea. The GloFAS runoff product captures the discharge of one of the main river systems for which we have direct observations, i.e., the Narmada-Tapti rivers, with simulated values of 86.95 km$^3$ y$^{-1}$ compared to 75.31 km$^3$ y$^{-1}$ reported by Krishna et al. (2016), and MOM6-COBALT-IND12 reproduces the range of salinity observed on the shelf at the river mouth (Figure 8a-f).

## 4.5 Seasonal plankton bloom dynamics

The northern Indian Ocean is characterized by two blooming seasons associated with the summer and winter monsoons that can be identified from surface chlorophyll (Chl> 0.5 mg m$^{-3}$; Figure 9a-e; e.g., Lévy et al., 2007). In the Arabian Sea, MOM6-COBALT-IND12 simulates the winter bloom (Figure 9a-c), which develops in response to nutrient supply by convective mixing (MLD of 40-80 m; Figure 4) and eddy vertical turbulent transport (Resplandy et al., 2011); it also simulates the summer bloom (Figure 9d-f) associated with the western and eastern Arabian Sea coastal upwelling systems (Oman, Yemen, Somalia, southwest India; see section 4.3) and a combination of horizontal and vertical eddy turbulent transport that supply nutrients to the central Arabian Sea (Resplandy et al., 2011). In the Bay of Bengal, the persistently low surface chlorophyll and its weak seasonality primarily result from strong salinity-driven stratification, which suppresses vertical nutrient supply to the mixed layer year-round (Sarma and Aswanikumar, 1991). Additionally, the presence of a subsurface chlorophyll maximum confines most primary production below the mixed layer, further reducing surface chlorophyll levels and attenuating their seasonal variability (Sarma and Aswanikumar, 1991). The model also simulates the subsurface chlorophyll maximum captured by Argo floats in both the Arabian Sea and Bay of Bengal, with RMSE values over the vertical ranging from 0.03 to 0.3 mg m$^{-3}$ (Appendix Figure A5). This suggests that the model effectively represents the vertical distribution of plankton and associated subsurface biological dynamics. Overall, comparison of our model's mean bias and RMSE with values reported in previous studies suggests that our chlorophyll simulation performance falls within the median range relative to other regional biogeochemical models (Chakraborty et al., 2023; Gutknecht et al., 2016; Sunanda et al., 2024).

The model overestimates surface chlorophyll concentrations by +0.25 to +0.75 mg m$^{-3}$ offshore of the western boundary currents (along Somalia, Kenya, Tanzania, and Oman) and in the southern Bay of Bengal (Figure 9c-f). Such discrepancies might be partly attributable to uncertainties in model chlorophyll estimates arising from photoacclimation, which modulates cellular pigment content under varying light conditions (Stock et al., 2025) and/or biases in satellite-derived chlorophyll, which can differ from in-situ measurements by up to a factor of two and exhibit regional biases, especially in coastal areas (Dierssen, 2010; Sathyendranath et al., 2019; Schofield et al., 2004). Importantly, we find here that this model-satellite discrepancy has limited impact on biogeochemical fluxes. Specifically, the model captures relatively well the observed integrated primary productivity and seasonality obtained from both available *in-situ* sampling (351 stations) and satellite-based products in all regions, in particular those of the CbPM satellite primary productivity product, which is in better agreement with *in-situ* observations than the other satellite products (see Figure 10 and Kalita and Lotliker, 2023, for an evaluation of the different products). The model captures the magnitude of the double bloom productivity in the central and western Arabian Sea (about 1000-1500 mg C m$^{-2}$ d$^{-1}$ in Figure 10a,b,e), as well as the lower productivity observed in the Bay of Bengal (<1000 mg C m$^{-2}$ d$^{-1}$; Figure 10f). The model also captures the timing of the summer bloom peak in productivity in the eastern Arabian Sea (EAS) and Somali upwelling (SOM), although the magnitude of modeled primary productivity might be underestimated in these regions (Figure 10a,c,d). The fact that the model simulates the magnitude of observed primary productivity (in carbon units) but overestimates the surface chlorophyll content suggests that it might overestimate the contribution of large phytoplankton, which is characterized by a higher chlorophyll-to-carbon ratio, compared to small phytoplankton, characterized by a lower

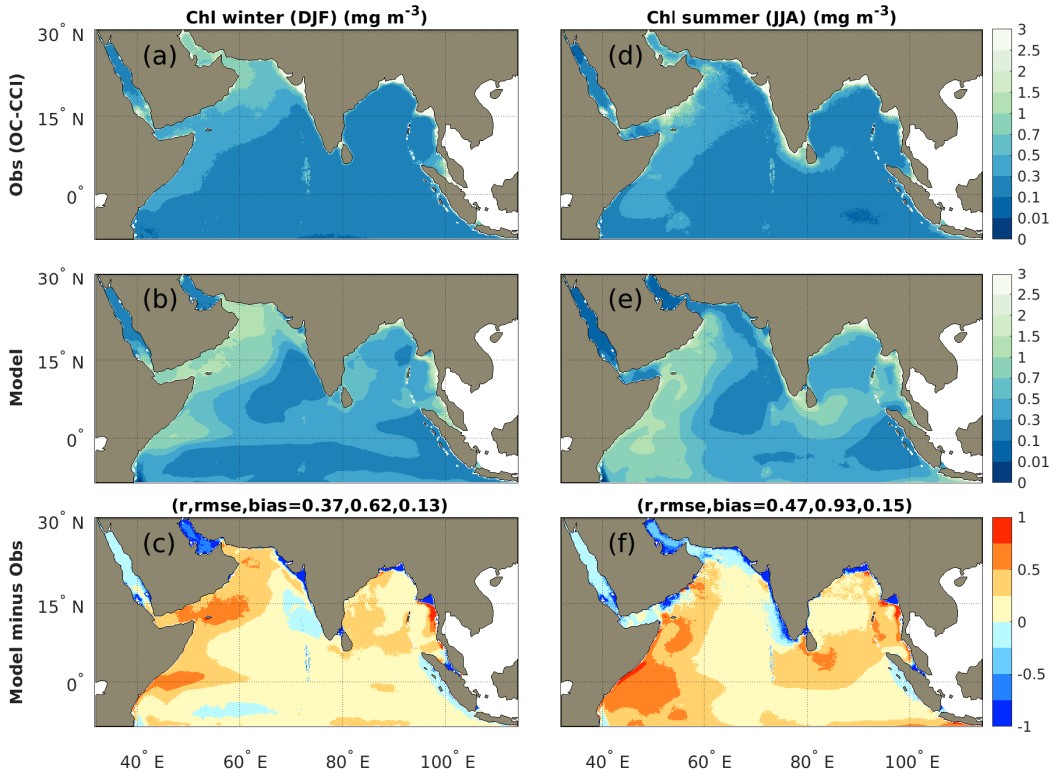

**Figure 9.** Sea surface chlorophyll during (a-c) winter (December-February) and (d-f) summer (June-August) monsoons. Panels (a,d) show satellite observations, (b,e) show MOM6-COBALT-IND12 model and (c,f) show differences between model results and observations. Correlation coefficients r, RMSD and bias between the observed and model annual means are indicated. Chlorophyll observations are from OCI-CC satellite (see details in Table 2). Model results are averaged over the 1980-2020 period.

chlorophyll-to-carbon ratio. This overestimation of the contribution of large phytoplankton to the assemblage would explain the good match in primary productivity and bias in chlorophyll.

In MOM6-COBALT-IND12, the phytoplankton limitation factors vary spatially and seasonally for the three phytoplankton groups included in the model (small, large and diazotroph; Figure 11). In the western Indian Ocean, the model simulates a strong seasonality: nitrogen and phosphorus are the most limiting nutrients in spring and early summer (March to May), but iron limitation becomes more prevalent towards the end of the summer bloom (September) and even persists in certain regions of the northern Arabian Sea until early winter (December) before it gets replenished by winter mixing (Figure 11). This shift

to iron limitation at the end of the summer monsoon is consistent with *in-situ* observations revealing a high-nutrient, low chlorophyll regime where phytoplankton growth is limited by iron in the Arabian Sea (Measures and Vink, 1999; Naqvi et al., 2010; Moffett et al., 2015; Moffett and Landry, 2020). We note, however, that during these periods of iron limitation, growth is weakly limited by nutrients (see total nutrient limitation values >0.5 in western and nothern Arabian Sea in September and

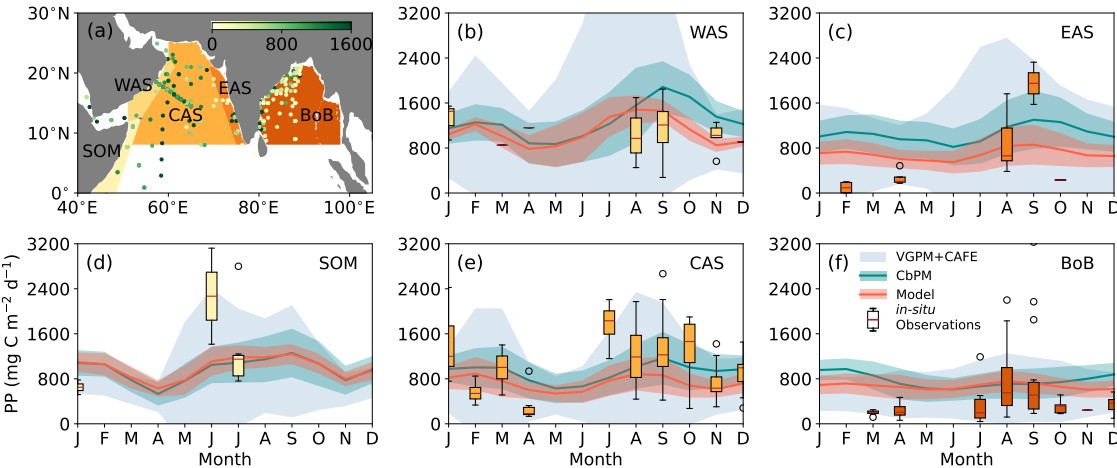

**Figure 10.** Seasonality in integrated primary productivity (PP). (a) maps of regions used to compare observation-based and modeled PP, showing *in-situ* observation sites with PP values in green (in mg C m$^{-2}$ d$^{-1}$), (b-f) monthly climatology of PP in MOM6-COBALT-IND12 model (regional mean $\pm$ 1-sigma in dark orange), from available in-situ observations in each region (boxplots showing median, interquartile, range and outliers defined as outside of 1.5 times the interquartile range), in the CbPM satellite product (regional mean $\pm$ 1-sigma in cyan) which performs best in this region (Kalita and Lotliker, 2023), and three additional satellite products ($\pm$ 1-sigma range of Standard-VGPM, Eppley-VGPM and CAFE in light blue). Regions are the western Arabian Sea (WAS), eastern Arabian Sea (EAS), Somalia coast (SOM), central Arabian Sea (CAS) and Bay of Bengal (BoB). Satellite and *in-situ* sampling observations are detailed in Table 2. Model and satellite-based climatologies are for the available observation period of 2003-2020.

December in the Appendix Figure A6). In the eastern Indian Ocean, the seasonality is weaker and phytoplankton are generally
limited by macronutrients (nitrogen and/or phosphorus), except in the northern Bay of Bengal where iron limitation becomes more important near river mouths that supply macronutrients in excess compared to iron (Figure 11). We note that the strong iron limitation near river mouths might be partly attributed to the way iron limitation is formulated in COBALTv2. Indeed, iron limitation depends on a cell quota (rather than the ambient nutrient concentration used for macro-nutrient limitations), which requires time to establish near the river mouths. Yet, we note that the overall pattern of limitation simulated in the Bay
of Bengal is consistent with incubation experiments showing a strong limitation by macronutrients in the southeastern Indian Ocean and co-limitations between macronutrients and iron in the Bay of Bengal (Twining et al., 2019).

## 5    Ocean interior, ventilation pathways and oxygen minimum zones

### 5.1    Ocean vertical structure and thermocline ventilation pathways

Observed subsurface temperature and salinity (300 m to 700 m average) reveal the signature of the main water masses that
ventilate the thermocline in the Indian Ocean (Figure 12a,d). The Red Sea and Persian Gulf overflows contribute warm and

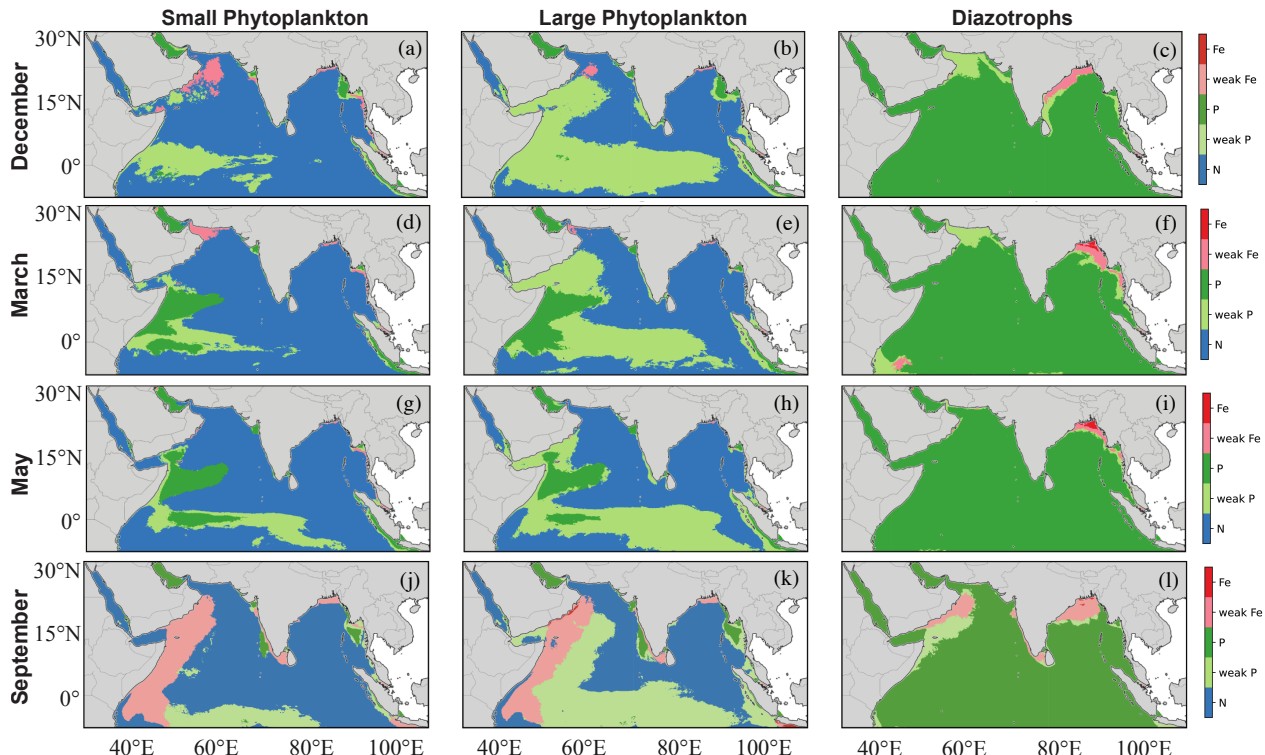

**Figure 11.** Climatological surface nutrient limitation (nitrogen N, phosphorus P and iron Fe) in MOM6-COBALT-IND12 for small phytoplankton, large phytoplankton and diazotrophs in December, March, May and September. Weak P (weak Fe) limitations indicate where P (Fe) is limiting but by a small amount relative to N or Fe (P) are near co-limiting (i.e., near co-limitation with difference between limitation factors < 0.25). Model climatology is based on the 1980-2020 period.

salty waters ($>13°$C and $>35.6$ psu) to the Gulf of Aden and Gulf of Oman in the Arabian Sea, respectively (You and Tomczak, 1993). In contrast, the Indonesian Throughflow (ITF) and the water masses formed in the southern subtropical and subpolar regions (e.g., mode waters and central waters) contribute relatively cold and fresh subsurface waters ($<8°$C and $<35$ psu) in the south of the domain, before being mixed and transported westward by the Southern Equatorial Current system and flowing northward and crossing the Equator along the African continent (You, 1997; Schott et al., 2004; Sprintall et al., 2009; McCreary et al., 2013; Nagura and McPhaden, 2018). Finally, intermediate temperature and salinity in the Bay of Bengal (about $10°$C and 35 psu) arise from the relatively weak thermocline ventilation, mostly maintained by the eastward transport from the Arabian Sea and Equatorial region.

MOM6-COBALT-IND12 reproduces the observed patterns in subsurface temperature and salinity in most of the basin (correlation coefficient r$>0.99$ and RMSE of $0.33°$C and $0.07$ psu). Specifically, the model simulates the contrast between the warm and salty waters in the northeastern Arabian Sea, the cold and freshwaters along the model southern boundary, and the waters with intermediate temperature and salinity in the Bay of Bengal (Figure 12). The largest departures are found in the

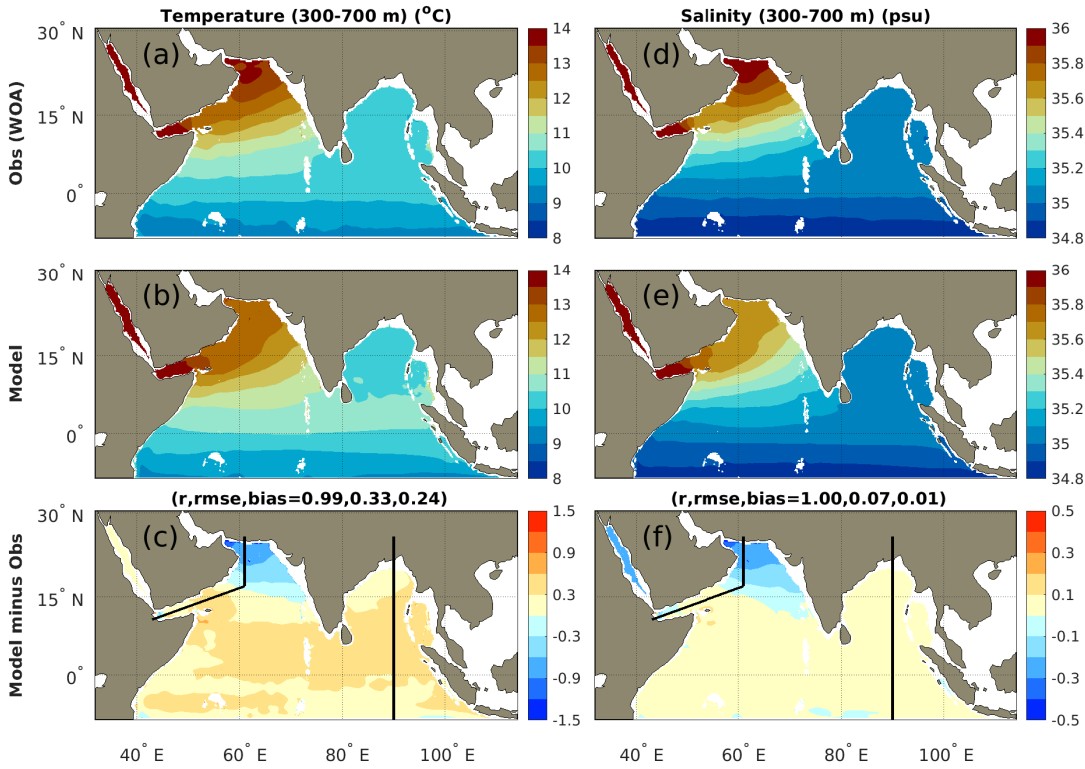

**Figure 12.** Annual mean subsurface (300-700 m depth) temperature and salinity in (a,d) observations, (b,e) MOM6-COBALT-IND12 model and (c,f) differences between model results and observations. Correlation coefficients r, RMSE and bias between the observed and model annual means are indicated. Temperature and salinity observations are from WOA18. Model results are averaged over the 1980-2020 period. Black lines indicate depth sections shown in Figures 13 and 14.

northern Arabian Sea where the model is biased cold and fresh (local bias between -0.8 and -0.3 °C and -0.4 to -0.1 psu; Figure 12), suggesting that the Persian Gulf overflow is not as well simulated as other pathways.

We further examine ventilation pathways using vertical sections in the eastern Indian Ocean, the Gulf of Oman and the Arabian Sea (Figures 13 and 14). In the eastern Indian Ocean (at 90°E), the model reproduces the observed vertical structure, including the intermediate salinity found in the subsurface Bay of Bengal and the influence of fresher ITF waters in the southern part of the domain (at ∼1000 m depth and latitudes <5°S; Figure 13). We note that the model only extends to 8°S, and therefore does not fully resolve the ITF centered at 5-10°S nor the Southern Equatorial Current at 10-20°S, but receives contributions

from ITF waters and southern waters through the open boundary. The model presents, a slight bias in the vertical structure of the Bay of Bengal, with slightly colder and fresher near-surface waters and slightly warmer and saltier subsurface waters with a small influence on the stratification in the region (Figure 13c,f).

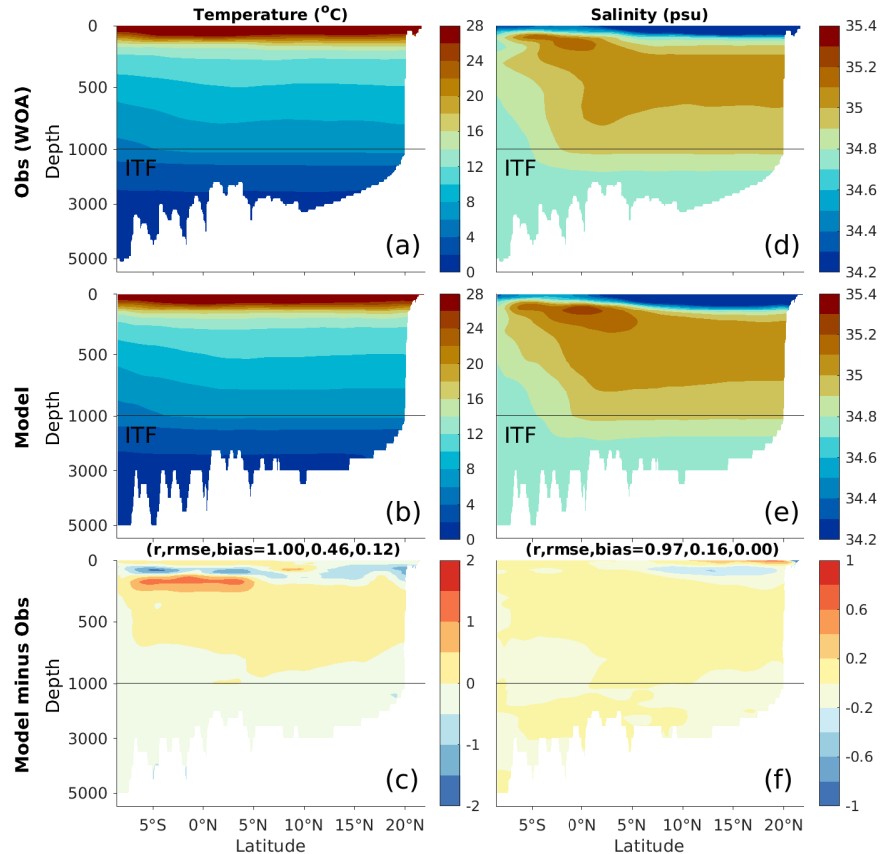

**Figure 13.** Depth section of annual mean temperature and salinity at 90°E in (a,d) observations, (b,e) MOM6-COBALT-IND12 model and (c,f) differences between model results and observations. Correlation coefficients r, RMSE and bias between the observed and model annual means are indicated. See location of section on Figure 12. Temperature and salinity observations are from the WOA18 (Table 2). Model results are averaged over the 1980-2020 period. Indonesian Throughflow waters (ITF) are indicated.

In the Gulf of Oman, observations show the plume of salty and warm Red Sea overflow waters (RSOW) that flow into the Gulf of Aden at a depth of 400-1000 m (>12°C and >36 psu; Figure 14a,d). The model simulates the depth range of the plume but the lower part of the plume is biased salty and warm (local bias of 1 to 2°C and 0.2 to 0.8 psu; Figure 14). This bias in the RSOW plume could come from biases in the source waters that overflow at the Bab-El-Mandeb Strait upstream, or from the misrepresentation of the plume mixing along the pathway. At the Bab-El-Mandeb Strait, we find that the model simulates remarkably well the volume transport of the three water masses flowing in and out of the Red Sea (Figure 15a). Specifically, the model simulates the observed outflow of RSOW that peaks in winter and drastically slows down in summer, the reversal of surface waters flowing into the Red Sea in winter and out of the Red Sea in summer, as well as the inflow of Gulf of Aden intermediate waters (GAIW) that only takes place in summer. The model, however, shows a bias in the density of these water masses, particularly in summer when simulated RSOW are lighter and surface waters (and to some extent GAIW, although

observations are sparse) are denser than observed (Figure 15b). This suggests there is insufficient mixing between the RSOW plume waters and the lighter (colder/fresher) waters above. This hypothesis is also supported by the structure of the temperature and salinity biases along the depth section showing a dipole of too salty / too warm waters in the lower part of the plume (800-1000 m depth) and slightly too fresh / too cold waters in the upper part of the plume (400-800 m depth; Figure 14c,f). This bias is, however, confined to the plume in the Gulf of Aden, and seems to have a relatively small influence on the vertical structure further downstream, explaining the good agreement in subsurface temperature and salinity in the southwestern Arabian Sea (Figure 12).

In the northern Arabian Sea, observations show that Persian Gulf waters (PGW) flow into the Arabian Sea at about 200-400 m depth (Figure 14). In MOM6-COBALT-IND12, however, PGW are too warm, too light and therefore enter the northern Arabian Sea at a too shallow depths of 100-200 m, leading to a cold/fresh bias at 200-400 m depth where PGW are located in observations and a warm-salty bias above (Figure 14). This trapping of the PGW close to the surface significantly changes the vertical structure of the northern Arabian Sea by reducing the stratification in the upper 200 m in the northern Arabian Sea.

## 5.2 Subsurface oxygen and oxygen minimum zones

Observed subsurface oxygen concentrations show the extent of the two OMZs located in the Arabian Sea and Bay of Bengal (Figure 16). In the Arabian Sea, averaged subsurface oxygen concentrations (300-700 m) are lower than 10 $\mu$mol kg$^{-1}$ in most of the region and reach suboxic values (<5 $\mu$mol kg$^{-1}$) around 15-20°N. In the Bay of Bengal, the OMZ is less intense with averaged subsurface concentrations of 10-20 $\mu$mol kg$^{-1}$ and no suboxia. The equatorial subsurface is better oxygenated, but still characterized by relatively low averaged oxygen subsurface concentrations of 50-100 $\mu$mol kg$^{-1}$ (Figure 16). Highest concentrations are found in the southwestern part, where the western boundary current supplies oxygen originating from the Southern Gyre and ITF (transported via the South Equatorial Current).

The MOM6-COBALT-IND12 model reproduces the observed large scale patterns of subsurface oxygen (basin-scale correlation coefficient r = 0.94 and RMSE = 16 $\mu$mol kg$^{-1}$; Figure 16a). The largest biases are found in the eastern (down to -30 $\mu$mol kg$^{-1}$) and western (up to +40 $\mu$mol kg$^{-1}$) north equatorial band where the gradients in oxygen are strong. In this region, the model shows a high oxygen bias near the base of the thermocline (500–1000 m), coinciding with a low nitrate bias (not shown). This pattern points to either a misrepresentation of biological remineralization at depth or an inaccurate representation of the relative contribution of the water masses forming the Central Waters supplying oxygen to this region. These waters originate from a blend of the ITF waters and southern-sourced Mode Waters. Previous studies have demonstrated that oxygen distribution in this region is highly sensitive to the relative contribution between these two sources (Ditkovsky et al., 2023). However, the scarcity of direct observations in this region limits our ability to conclusively attribute the model bias to either mechanism.

Yet, the most biogeochemically relevant bias is probably the overestimation of the extent and intensity of suboxic conditions in the northern Bay of Bengal, where the local difference in modeled versus observed oxygen concentration ranges from -20 to -10 $\mu$mol kg$^{-1}$, reflecting a much larger extent of suboxia in the model than in observations (Figure 16a). We evaluate the model ability to reproduce the volume of the OMZ as a function of the oxygen threshold chosen to define its boundary (i.e.,

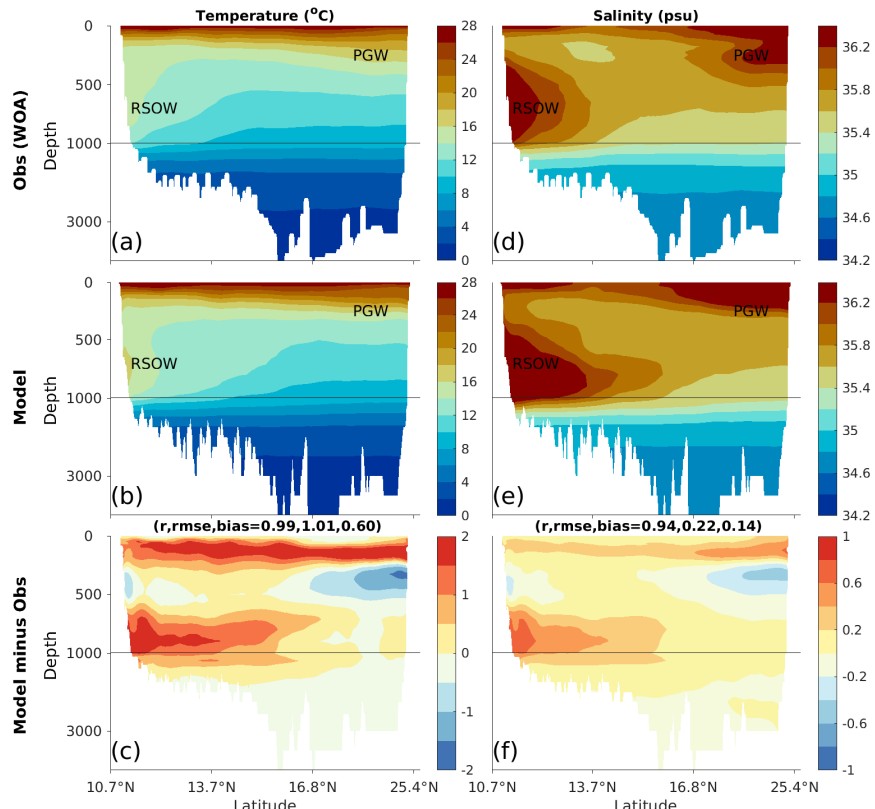

**Figure 14.** Depth section of annual mean temperature and salinity in the Gulf of Oman (southwest to northeast) and the Arabian Sea (south-north at 61°E) in (a,d) observations, (b,e) MOM6-COBALT-IND12 model and (c,f) differences between model results and observations. See location of section on Figure 12. Correlation coefficients r, RMSE and bias between the observed and model annual means are indicated. Temperature and salinity observations are from the WOA18 (Table 2). Model results are averaged over the 1980-2020 period. Persian Gulf Waters (PGW) and Red Sea Overflow Waters (RSOW) are indicated.

volume bounded by oxygen concentrations from 5 to 150 $\mu$mol kg$^{-1}$, Figure 16b). At the basin scale, MOM6-COBALT-IND12 reproduces relatively well the observed OMZ volumes defined by thresholds above 30 $\mu$mol kg$^{-1}$, in particular the volume of hypoxic waters delimited by 60 $\mu$mol kg$^{-1}$ (approximately $1 \times 10^{16}$ m$^3$) and the volume of low oxygenated waters delimited 515 by 100 $\mu$mol kg$^{-1}$ (approximately $2 \times 10^{16}$ m$^3$; Figure 16b). In contrast, the model overestimates the volume of suboxic waters delimited by 5 $\mu$mol kg$^{-1}$ ($0.17 \times 10^{16}$ m$^3$ vs. $0.06 \times 10^{16}$ m$^3$ in Bianchi et al. (2012) observations), mostly because of the large suboxic volume simulated in the Bay of Bengal ($0.10 \times 10^{16}$ m$^3$ vs. $0.00 \times 10^{16}$ m$^3$ in observations; Figure 16b). Meanwhile, the volume of suboxic waters in the Arabian Sea is well represented ($0.07 \times 10^{16}$ m$^3$ vs. $0.06 \times 10^{16}$ m$^3$). Finally, we note that the good match between observed and modeled hypoxic volumes is favored by the partial compensation of small biases in the

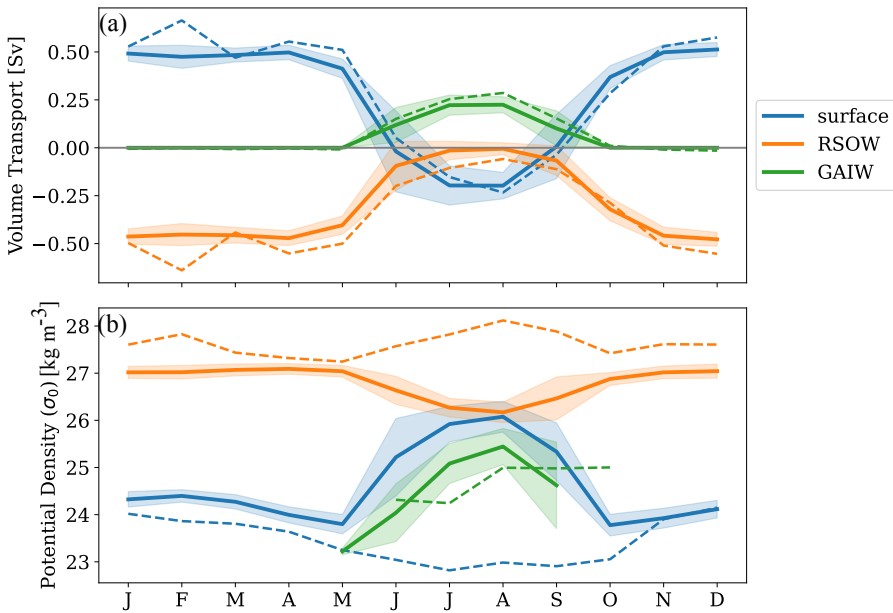

**Figure 15.** Seasonal evolution of the (a) volume transport and (b) potential density at the Bab-El-Mandeb Strait (between the Red Sea and the Gulf of Aden) in observations (dashed) and the model (solid). The three water masses are surface waters, Gulf of Aden Intermediate Waters (GAIW) and Red Sea Outflow Waters (RSOW). Observations and water mass detection method using flow direction are from Sofianos et al. (2002, see Table 2). Positive transport is into the Red Sea, negative transport into the Gulf of Aden. Model is averaged over 1980-2020. See details on water masses in Section 3.1.

Arabian Sea (model volume about $0.14 \times 10^{16}$ m$^3$ lower than in observations) and the Bay of Bengal (model volume about $0.06 \times 10^{16}$ m$^3$ higher than in observations; Figure 16b).

## 6 Intraseasonal variability

We quantify the intraseasonal variability (ISV) in the surface ocean circulation using the intraseasonal standard deviation of the sea level anomaly (see Section 3.2). This diagnostic captures variability linked to all dynamical processes varying on intraseasonal time-scales, which includes mesoscale eddies and filaments, as well as meandering jets and planetary waves (Rossby and Kelvin waves). These intraseasonally varying features are key to the transport and mixing of physical and biogeochemical tracers, such as nutrients and oxygen, and to the onset and spatial extent of the seasonal phytoplankton blooms in the Indian Ocean (e.g., Resplandy et al., 2011, 2012; Lachkar et al., 2016; Rixen et al., 2020; Pearson et al., 2022; Vinayachandran et al., 2021)

Satellite observations show two hotspots where the intraseasonal variability in SLA exceeds 5 cm and can reach values higher than 10 cm (Figure 17a). The first hotspot is in the western Arabian Sea offshore Somalia and the Arabian Peninsula, where the high energy dynamics of the western boundary current and the presence of upwelling systems and complex coastal

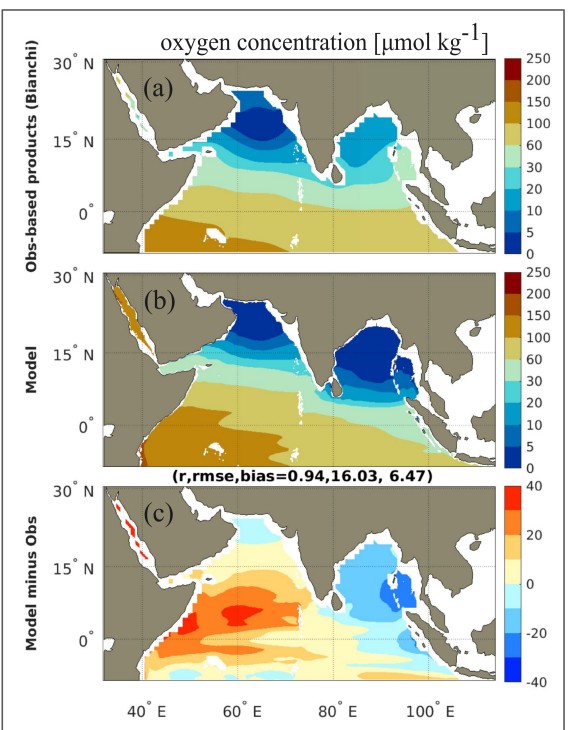
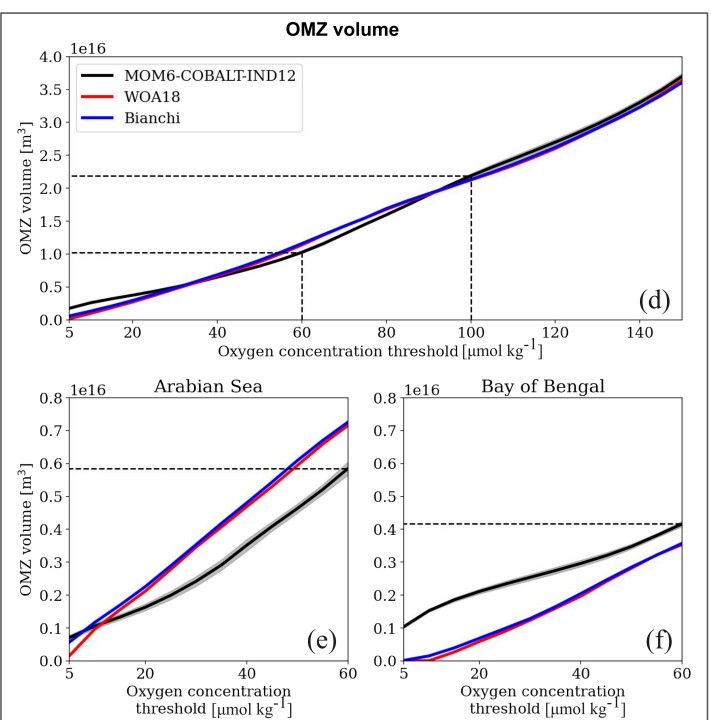

**Figure 16.** (a) Annual mean subsurface (300-700 m depth) oxygen concentrations in observations (top row, Bianchi et al., 2012), MOM6-COBALT-IND12 (middle row) and differences between model results and observations (bottom row). Correlation coefficients r, RMSE and bias between the observed and model annual means are indicated. (b) Observed and simulated ocean volume within a certain oxygen concentration threshold in the model domain (top row), the Arabian Sea and the Bay of Bengal (bottom row). Observations from Bianchi et al. (2012, in blue) and WOA18 (in red) differ mostly on the volume at low oxygen values. Grey shading indicates the 1-sigma model interannual variability. Model results are for the 1980-2020 period.

topography (capes/headland) promote the formation of large mesoscale eddies such as the Great Whirl and Socotra Eddy (see section 4.2), and filaments extending from the Arabian Peninsula into the central Arabian Sea (e.g., Beal and Donohue, 2013; Resplandy et al., 2011; Brandt et al., 2003; Wang et al., 2018). The second hotspot covers the central and western Bay of Bengal and extends south of Sri Lanka, and has been attributed to mesoscale eddies and Rossby waves generated in the coastal eastern Bay of Bengal that propagate westward into the central and western Bay of Bengal (Sengupta et al., 2001, 2007; Cheng et al., 2013). MOM6-COBALT-IND12 simulates the locations of the two hotspots of highest ISV in the western Arabian Sea and western Bay of Bengal, but the amplitude tends to be weaker than observed, with typical values of 3-8 cm in the model versus 4-12 cm in the observations (Figure 17).

Other regions of relatively high observed ISV (>3 cm) include the mouths of major rivers, such as the Ganges-Brahmaputra, Irrawaddy-Sittang and Narmada-Tapti river systems (see Figure 1 for rivers location), coastal ocean waters along the eastern

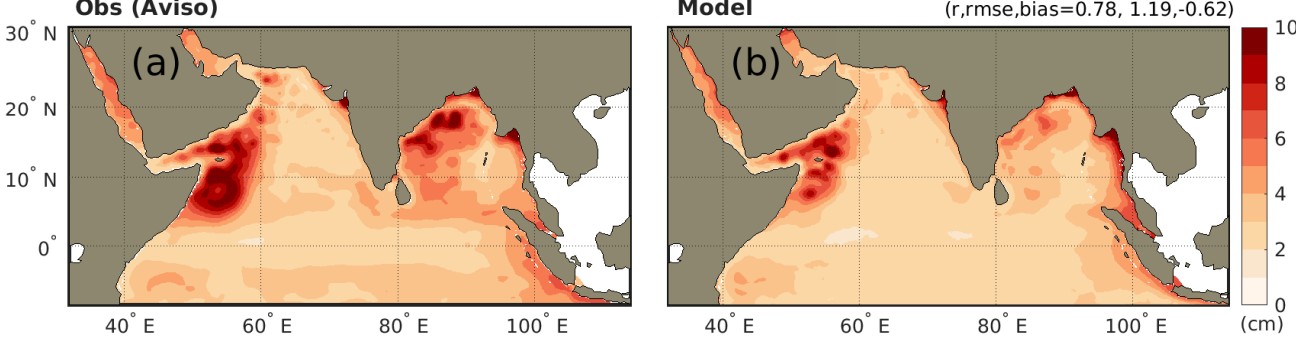

**Figure 17.** Intraseasonal variability (includes mesoscale eddy activity and wave-driven variability) quantified by the intraseasonal standard deviation of the sea level anomaly (SLA) in (a) AVISO satellite observations and (b) MOM6-COBALT-IND12. SLA over the 1994-2017 period was detrended using a linear regression and filtered using a 14-120 days band-pass filter.

Bay of Bengal and eastern Arabian Sea, and the 5°N-10°N band in both the Arabian Sea and Bay of Bengal (Figure 17). ISV at the river mouths and the coastal ocean can largely be attributed to the ISV in river freshwater discharge (up to 50% of seasonal variability amplitude for the Ganges, for instance; Jian et al., 2009), tidal forcing and the propagation of coastal Kelvin waves (e.g., Nienhaus et al., 2012). MOM6-COBALT-IND12 reproduces relatively well the observed ISV in the coastal ocean and part of the ISV at river mouths.

Finally, the intraseasonal variability in the 5°N-10°N band, which reaches 3 to 5 cm in the satellite-based estimate in response to the westward propagation of Rossby waves (Bruce et al., 1994; Shankar and Shetye, 1997; Vialard et al., 2009; Cheng et al., 2017), is also underestimated in the model (1-3 cm; Figure 17). These underestimations might indicate the current spatial resolution of the model (1/12°) may still be insufficient for resolving these processes.

Figure 18 illustrates the influence of eddies and filaments on surface chlorophyll and phytoplankton production and their seasonality in the Arabian Sea, specifically in the first hotspot of ISV described above (western Arabian Sea and central Arabian Sea). The model reproduces the fine-scale features structuring the winter and summer blooms. During the winter monsoon, fine-scale eddies (∼ 20-50 km in diameter) shape the bloom occurring in the northern and central Arabian Sea (Figure 18a,b,e,f). This is consistent with the results of Resplandy et al. (2011), which showed that these fine-scale eddies sustain the bloom by vertically transporting nutrients to the euphotic zone during early winter and by locally re-stratifying and alleviating light limitation during late winter when convection occurs (see section 4.2 and Figure 4 for mixed layer seasonality). During the summer monsoon, surface chlorophyll is highest in the coastal upwelling regions of Oman and Somalia in the early phase of the bloom (Figure 18c,g) and then extends offshore in long filaments wrapped around mesoscale eddies in the central Arabian Sea and around the Great Whirl in the late phase of the bloom (Figure 18d,h). The structure of the bloom here is also consistent with the findings of Resplandy et al. (2011), which showed that eddy-induced vertical transport supplied most of the nutrients in coastal waters during the early stage of the summer upwelling, while horizontal transport by filaments supplied nutrients to

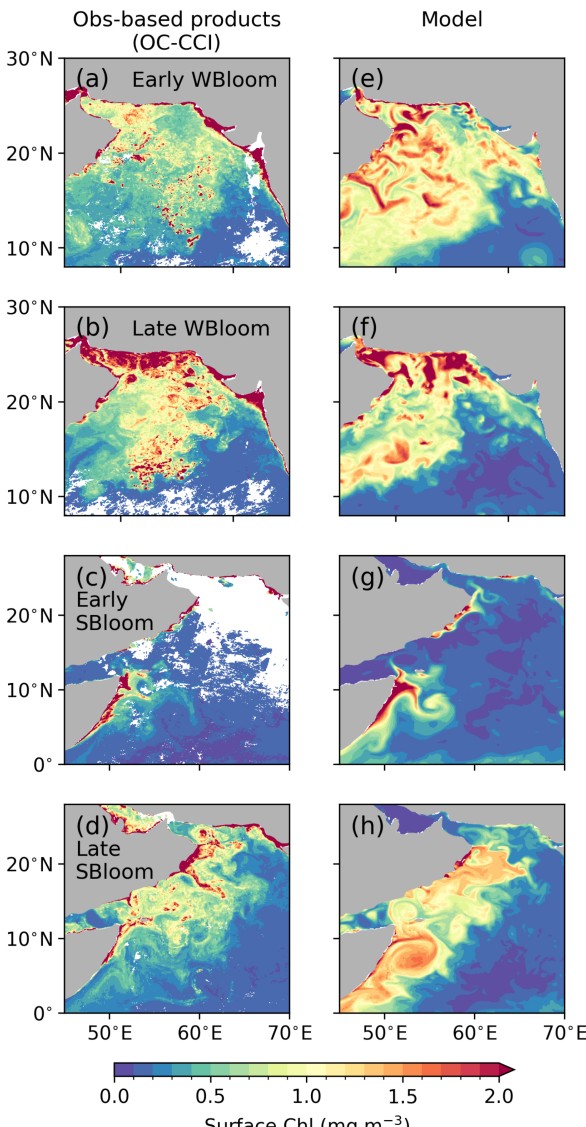

**Figure 18.** Fine-scale structures (eddies, filaments) revealed by surface chlorophyll in the OC-CCI satellite product (a-d) and the MOM6-COBALT-IND12 model (e-h). Snapshots are for (a,e) early winter bloom (WBloom, Jan. 29, 2014), (b,f) late WBloom (Mar. 10, 2014), (c,g) early summer bloom (SBloom, Jun. 6, 2011) and (d,h) late SBloom (Oct. 4, 2011). OC-CCI images are 8-day composites and model images are 7-day averages (e-h).

the central Arabian Sea. We note that the shape of the winter and summer eddies and filaments is well captured by the model, although their exact location might not be the same. Indeed, we expect the model to reproduce the statistics of mesoscale structures (e.g., eddies and filaments) for a given season and region, but not necessarily their exact position. We also note

that simulated surface chlorophyll concentrations are biased high in the model, in particular during the late summer monsoon (Figure 18d,h). This is in line with the finding that chlorophyll is overestimated in the model, although primary productivity appears to be well simulated, likely due to the high contribution of large phytoplankton with high chlorophyll-to-carbon ratio (see section 4.5 and Figures 9,10).

## 7 Interannual Indian Ocean Dipole

The model reproduces the amplitude and zonal pattern of SST changes expected in response to the Interannual IOD ($r > 0.9$; Figure 19b-e; see Figure 19a for timing of positive and negative IOD phases). This includes the strong SST response in the eastern equatorial Indian Ocean, offshore Java and Sumatra, where the surface cools by -0.5 to -1°C during positive IODs and warms by +0.5 to +1°C during negative IODs, as well as the weaker response in the eastern and central equatorial Indian Ocean, where the ocean surface warms by +0.2 to +0.5°C during positive IODs and cools by -0.2 to -0.5°C during negative IODs. This SST signature of IODs is associated with anomalous winds and changes in thermocline depth along the equator (Saji et al., 1999; Webster et al., 1999; Currie et al., 2013). During positive IODs, easterly wind anomalies in the central Indian Ocean shallow the thermocline in the east and generate anomalously cold eastern SSTs. In the west these wind anomalies, in conjunction with Rossby waves, deepen the thermocline and produce anomalously warm western SSTs. During negative IODs, anomalous westerly winds lead to the opposite east/west pattern in SST and thermocline depth. These SST signatures develop in boreal summer, peak in fall, and decay through winter.

Figure 20 focuses on this zonal contrast introduced by the IOD, through a comparison of observed and modeled interannual anomalies in SST and thermocline depth at two equatorial Indian Ocean sites: one eastern mooring offshore Sumatra (95°E, 5°S) and one western mooring in the Seychelles-Chagos thermocline ridge (57°E, 4°S). We use observations from the *in-situ* RAMA that we complement with OISST data and Argo float-based thermocline depth (see Table 2). MOM6-COBALT-IND12 reproduces particularly well the timing and amplitude of interannual variations in SST (r of 0.78-0.90 with RAMA and 0.75-0.79 with OISST) and in thermocline depth (r of 0.75-0.82 with RAMA and 0.73-0.84 with Argo) at both RAMA stations (Figure 20), including the asymmetry in the response between IOD phases (Hong et al., 2008b, a; Cai et al., 2013; Nakazato et al., 2021). At the eastern station, the thermocline deepens by 20-30 m and SSTs increase by +0.5 to +1°C during negative IODs. In contrast, the thermocline only shallows by 10-20 m and SSTs generally decrease by less than -0.5°C during positive IODs, except during the strong positive IOD of 2019 during which SSTs cooled by more than 1.5°C in both observations and models. The model also captures interannual variations observed at the western station Figure 20. While IOD-driven variability is present at the western mooring, its influence is likely weaker compared to other sources of variability. Nevertheless, the model reproduces the associated variabilities, including a deeper thermocline and cooler SSTs during negative IODs, and shallower thermocline and warmer SSTs during positive IODs (Figure 20).

The wind anomalies associated with the IOD also produce equatorially trapped Kelvin waves that travel east towards Sumatra and Java, impinge on their coasts and continue traveling counterclockwise around the rim of the northern Indian Ocean, thereby modulating the seasonal upwelling/downwelling motions described in section 4.3 above (see details on coastal Kelvin wave

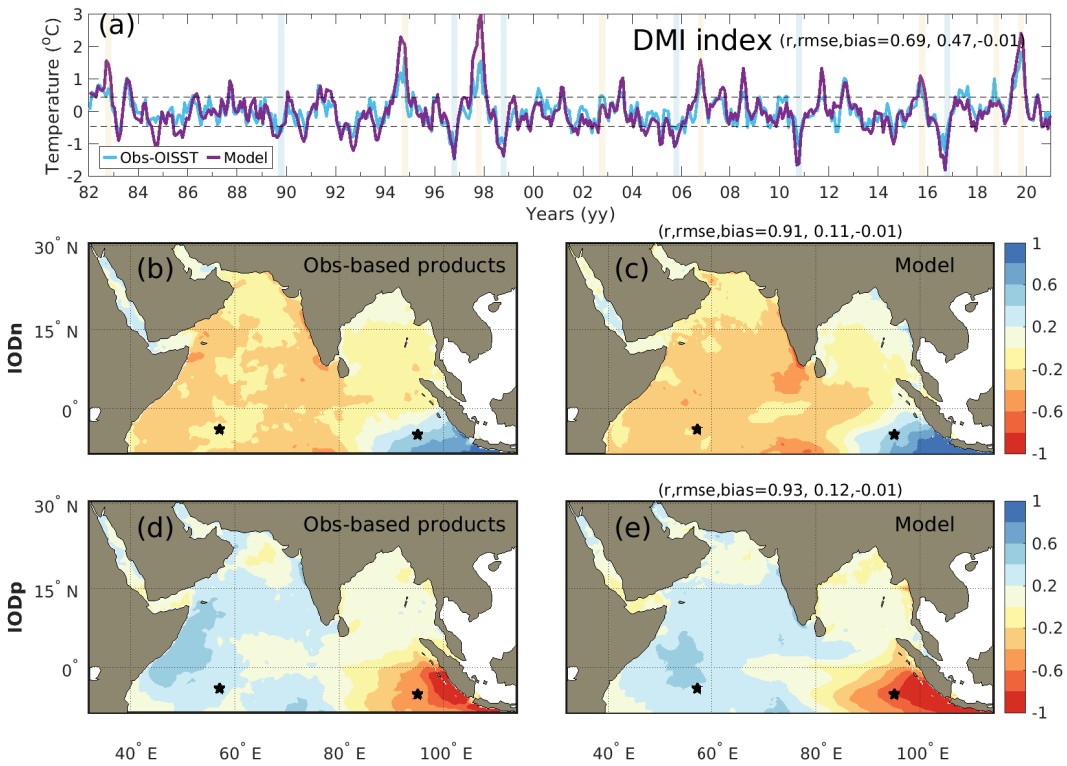

**Figure 19.** Interannual variability associated with the Indian Ocean Dipole (IOD). (a) Dipole mode index (DMI) which quantifies the intensity of the IOD phases; (b,c) SST composites during IOD negative phases (IODn) in observations and MOM6-COBALT-IND12 model; (d,e) SST composite during IOD positive phases (IODp) in observations and MOM6-COBALT-IND12 model. Composites are for September-to-November months of positive (1982, 1994, 1997, 2002, 2006, 2015, 2018, 2019) and negative (1989, 1996, 1998, 2005, 2010, 2016) IODs. SST observations are from OISSTv2.1 (see Table 2). Black stars indicate the positions of two Research Moored Array for African-Asian-Australian Monsoon Analysis and Prediction (RAMA) moorings used in Figure 20.

modulation by IOD in Aparna et al., 2012; Suresh et al., 2018; Pearson et al., 2022). As shown in Figure 7, the model reproduces the coastal SLA interannual anomalies associated with IOD phases. In particular, it simulates the upwelling anomaly observed between September and January during positive IOD phases along the coasts of the Bay of Bengal (arrows for waves IX and X; SLA internanual anomalies of -12 to -5 cm), and the downwelling anomaly observed during negative IOD phases (arrows for waves XI and XII, same months; SLA internanual anomalies of +5 to +12 cm; Figure 7). The model also simulates the

weaker SLA anomalies of opposite sign (compared to the Bay of Bengal) observed along the coasts of the Arabian Sea (SLA interannual downwelling anomaly of +2 to +5 cm during positive IODs, and upwelling anomaly of -2 to 0 cm during negative IODs; Figure 7).

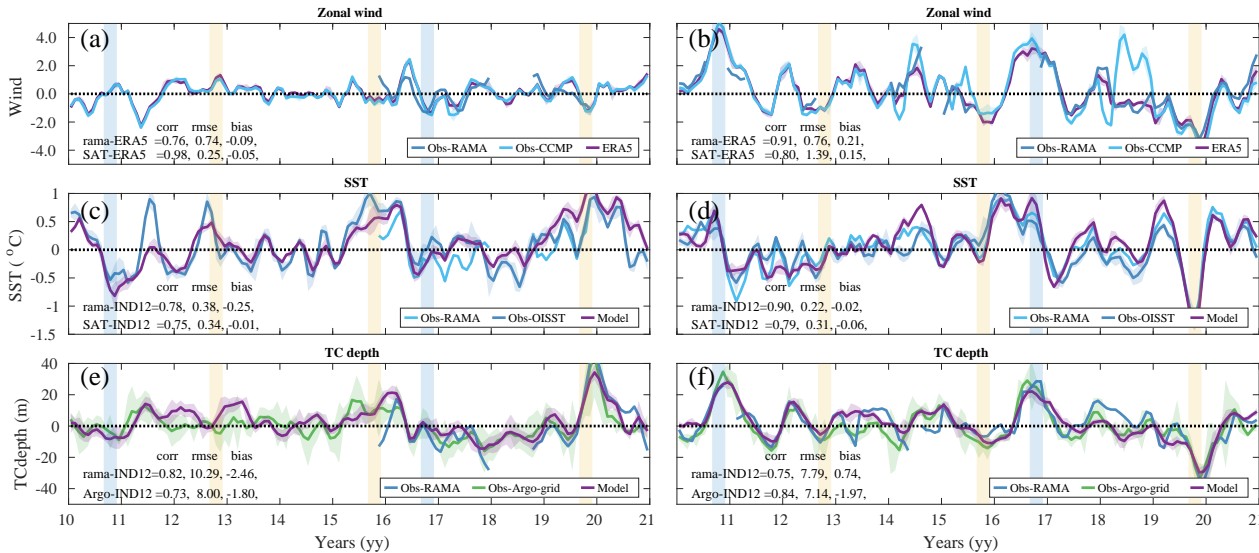

**Figure 20.** Interannual variability associated with the Indian Ocean Dipole (IOD) at two RAMA moorings in western (a,c,e: 57°E, 4°S) and eastern (b,d,f: 90°E, 5°S) equatorial Indian Ocean. panels (a,b): zonal wind (m s$^{-1}$) from observations (RAMA mooring and CCMP satellite) and the ERA5 reanalysis used to force MOM6-COBALT-IND12; (c,d): SST in observations (RAMA moorings and OISSTv2.1) and in MOM6-COBALT-IND12; (e,f): thermocline depth (TCdepth) is calculated as the depth of the 20 °C isotherm. Positive and negative IODs are indicated by orange and blue shading. Correlation coefficients r, RMSE and bias between the observed and reanalysis or model time-series are indicated in each panel. Positions of the two RAMA moorings are shown by black stars in Figure 19.

IOD phases are associated with biogeochemical signatures visible at the basin scale in satellite ocean color observations (Murtugudde and Busalacchi, 1999; Wiggert et al., 2009; Currie et al., 2013). Figure 21 compares composites of integrated
primary productivity anomalies from the CbPM satellite and MOM6-COBALT-IND12 in boreal fall (September-November). Negative IOD phases are characterized by negative primary productivity anomalies in the eastern equatorial Indian Ocean (-150 to -300 mg C m$^{-2}$ d$^{-1}$ offshore Sumatra; Figure 21a) due to the depressed thermocline and associated nutricline and weaker upwelling-favorable wind (Figure 20), and positive primary productivity anomalies in the western equatorial Indian Ocean (+50 to +150 mg C m$^{-2}$ d$^{-1}$ offshore Somalia) due to the shallower thermocline and nutricline (Figure 20). Negative IODs are
also associated with strong positive primary productivity anomalies around the tip of India (> +200 mg C m$^{-2}$ d$^{-1}$) associated with the wave-driven shoaling of the thermocline and nutricline (Figure 7) and positive anomalies in most of the Arabian Sea (Figure 21a). The response to positive IODs mirrors the response of negative IODs in the equatorial Indian Ocean, with positive anomalies observed in the eastern equatorial Indian Ocean and negative anomalies in the western equatorial Indian Ocean and around the tip of India (Figure 21b). We note, however, that the primary productivity anomalies in the northern and central
Arabian Sea are positive during both negative and positive IOD phases. The model captures remarkably well the pattern and

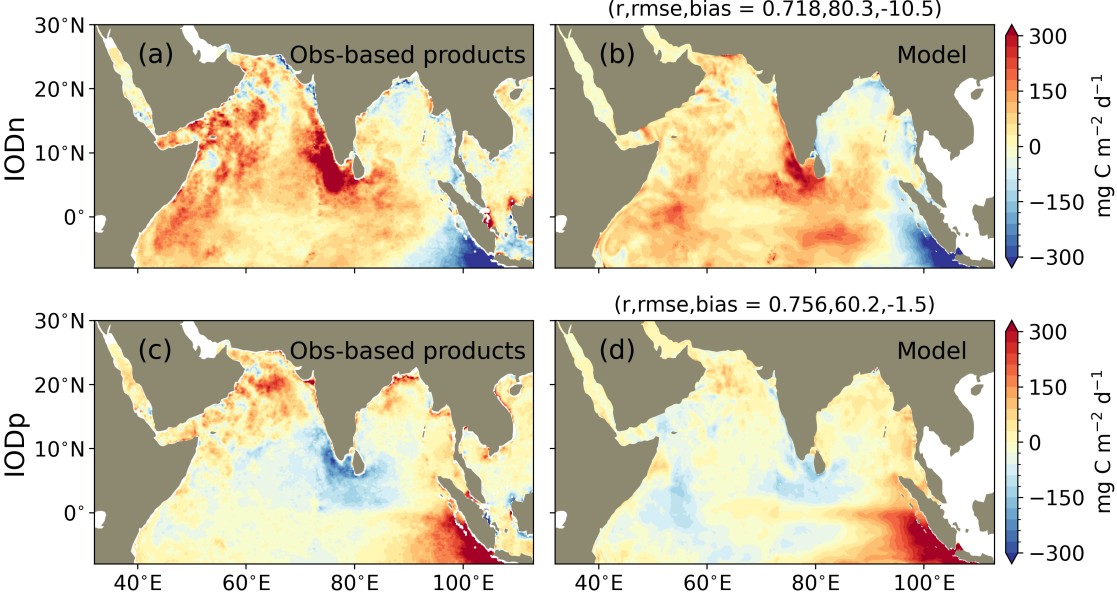

**Figure 21.** Integrated net primary productivity (PP) anomaly associated with the Indian Ocean Dipole (IOD). September-to-November PP composites during (a-b) IOD negative phases (IODn) and (c-d) IOD positive phases (IODp) in observation-based product and MOM6-COBALT-IND12. Composites are for September to November months available in CbPM satellite product for positive (2002, 2006, 2015, 2018, 2019) and negative (2005, 2010, 2016) IODs (see Table 2 for details on data).

sign of the observed primary productivity anomalies during both negative and positive IODs (correlation coefficient r > 0.7), although the amplitude of the anomaly is slightly lower in the model than in the CbPM satellite product (RMSE of 60 to 80 mg C m$^{-2}$ d$^{-1}$, bias of -1.5 to -10.5 mgC m$^{-2}$ d$^{-1}$; Figure 21c-d). On the one hand, the model–observation discrepancy in primary productivity anomaly amplitude may stem from weak nutrient variability associated with the shallow MLD. On the other hand,

the amplitude of the primary productivity anomalies obtained from satellite products remains uncertain. For instance, the primary productivity anomaly composites obtained from another satellite product (CAFE) show similar patterns but with an amplitude that is about half of the CbPM satellite product (Figure 21a,c and Appendix Figure A7a,c). The amplitude of the anomaly in the model sits between the two satellite products (RMSE of 60-80 mg C m$^{-2}$ d$^{-1}$ in both cases and absolute bias between -10.5 and +24 mg C m$^{-2}$ d$^{-1}$; see Figure 21 and Appendix Figure A7). Given the discrepancies among observational

datasets, the model likely provides a reasonable estimate of primary productivity variability. Nevertheless, additional in situ observations are needed to more accurately constrain the true primary productivity response to the IOD.

## 8 Discussion and conclusions

In this study, we configured, customized, and validated a high-resolution (1/12°) regional ocean biogeochemical model (MOM6-COBALT-IND12 v1.0) for the northern Indian Ocean. Specifically, we adjusted river discharge rates and nutrient loadings in

the Bay of Bengal according to observational constraints, significantly improving simulations of river plume dynamics and surface salinity. Additionally, we enhanced lithogenic particle fluxes from rivers, adjusted detritus sinking rate, and refined the parameterization of the nitrogen cycle, resulting in better representations of subsurface oxygen distributions and suboxic conditions. These improvements collectively allow the model to capture most key aspects of biogeochemical and physical processes in the northern Indian Ocean

At the basin scale, the MOM6-COBALT-IND12 model simulates the contrast between the Arabian Sea – characterized by high evaporation, inflow from the saline marginal seas (Red Sea and Persian Gulf) and high upper ocean salinity – and the Bay of Bengal – characterized by high precipitation, high river runoffs and low upper ocean salinity. On seasonal time-scales, the model captures the monsoonal reversal in ocean circulation, including the development of the Great Whirl and wind-driven summer coastal upwelling systems along the western boundary, the winter convective mixing in the northern Arabian Sea, as

well as the propagation of upwelling and downwelling coastal Kelvin waves along the equatorial waveguide and the rim of the northern Indian Ocean. On intraseasonal time-scales, the model also reproduces the hotspots of variability associated with eddies, filaments and planetary waves, and on interannual time-scales the east-west variability in the thermocline introduced by the IOD. This strong physical performance likely stems from the effective parameterizations of surface heat and momentum fluxes, combined with well-constrained surface forcing fields derived from ERA5 reanalysis.

The good agreement between observed and modeled physical features provides a foundation for accurately simulating the ocean biogeochemical and biological response. This includes the intensity and timing of the seasonal blooms triggered by monsoonal circulation changes and modulated by intraseasonal features such as eddies and filaments, and interannual IOD phases. Specifically, the model reproduces the summer bloom associated with coastal upwelling systems and their extension offshore in mesoscale filaments, as well as the winter bloom associated with convective mixing and modulated by fine-scale

eddies (Lévy et al., 2007; Resplandy et al., 2011, 2012; Mahadevan, 2016; Lachkar et al., 2016; Rixen et al., 2019a; Vinay-achandran et al., 2021; Anjaneyan et al., 2023). These biogeochemical improvements reflect the targeted model development efforts outlined above. The model also captures the patterns and amplitude of the phytoplankton changes expected in response to the IOD positive and negative phases. This includes the modulation of the production in the equatorial region, the Arabian Sea and around the tip of India, although we note these patterns are difficult to generalize to all IOD events. As illustrated

by Wiggert et al. (2009), the chlorophyll responses vary with IOD intensity-for instance, concentrations in the Arabian Sea decreased during the 1997 event but increased during the 2006 event.

The comparison (Appendix Figure A8) between MOM6-COBALT-IND12 and the global model of Liao et al. (2020) demonstrates that the regional model more realistically captures high-frequency variability in SSH, mesoscale dynamics, meandering jets, and planetary waves (Rossby and Kelvin waves). These features significantly influence nutrient and oxygen transport and

mixing, as well as the timing and spatial patterns of seasonal phytoplankton blooms across the Indian Ocean. Our regional configuration also notably improves the representation of marginal sea outflows, particularly from the Red Sea, where global models typically overestimate overflow strength. Furthermore, targeted parameter adjustments–including river discharge, nutrient load, detritus sinking rate, and nitrogen cycle parameterization–improve dissolved oxygen simulations in both the Arabian

Sea and the Bay of Bengal. Collectively, these refinements, substantially improve the accuracy and reliability of physical and
biogeochemical processes simulated by our regional model.

During the setup and customization of the MOM6-COBALT-IND12 v1.0 model, we identified a series of physical and biogeochemical parameters and forcings that influenced the model simulation and led to a significant improvement of the results (see details in section 2). One of the factors that influenced our results were river discharge and nutrient loadings, especially in the Bay of Bengal which hosts major river systems such as the Ganges, Brahmaputra, Irrawaddy and Sittang rivers. A first version of the model used the river inputs from the Japanese 55-year Reanalysis (JRA55-do, Kobayashi et al., 2015) instead of the modified GloFAS product presented in this study. However, we found systematic biases in the timing, amplitude and variability of the riverine discharge in JRA55-do. These biases include a systematic delay of 1-2 months in the annual maximum discharge and a lower intraseasonal variability (Appendix Figure A1), which led to biases in river plume dynamics and SSS in the northern Bay of Bengal and the eastern Arabian Sea, in line with observations showing that riverine discharge timing and variability are critical to salinity patterns and plume dynamics Li et al. (2021). In addition to river discharge, we modified nutrient loadings to match available observational constraints which was important to reproduce productivity patterns in the coastal Bay of Bengal. We note that the influence of riverine inputs could be further improved by accounting for the anthropogenic increase in riverine nutrient supply (MOM6-COBALT-IND12 v1.0 includes nutrient inputs equivalent to year 2000 from Mayorga et al., 2010), which would likely introduce a long-term trend in coastal primary productivity and oxygen concentrations in the vicinity of large river systems.

While the MOM6-COBALT-IND12 v1.0 configuration is remarkably successful at capturing many of the features and observed variability of the northern Indian Ocean, there are still some areas where there is potential for improvement. The main model bias is the larger horizontal extent and volume of suboxia (oxygen concentrations $< 5~\mu$mol kg$^{-1}$) simulated in the Bay of Bengal. This bias is a well known limitation of ocean and Earth system models in this region (e.g., Bopp et al., 2013; Schmidt and Eggert, 2016; Ditkovsky et al., 2023). The advective supply of oxygen to the thermocline in the Bay of Bengal is weak, but we expect low oxygen demand to prevent the formation of suboxic waters. However, the subsurface oxygen biological demand is likely too high in the model. Notably, this bias in oxygen in the Bay of Bengal was larger in a prior version of the model, and was mitigated by adjusting some of the model parameters. A first set of changes focused on riverine lithogenic fluxes. The increased influx of riverine lithogenic material by an order of magnitude for major rivers and about 50% for small rivers protects more particulate organic matter from remineralization due to the ballasting effect, significantly reducing oxygen consumption in the water column. A higher total river input of lithogenic material in the Bay of Bengal resulted in a greater reduction in oxygen consumption compared to the Arabian Sea. A second set of changes focused on detritus sinking velocities and burial. The detritus sinking velocity was increased by 20% to match sediment trap observations in the region (Rixen et al., 2019b) and the fraction of material that reach the ocean floor and is buried was also increased to match the observation-based reconstruction of LaRowe et al. (2020). These modifications reduced remineralization and oxygen consumption in the subsurface and at depth, further reducing the bias in the size and volume of the Bay of Bengal OMZ, while having a relatively small impact on the Arabian Sea OMZ core where oxygen is entirely depleted. The impact of these modifications are consistent with findings from Luo et al. (2024) and Al Azhar et al. (2017), who showed that fast-sinking detritus reduced oxygen consumption

and shrank OMZs, expanding oxygenated regions at the OMZ boundaries. A third set of modifications focused on the representation of nitrogen cycling in low oxygen environments. These changes allowed denitrification at oxygen concentrations up to 4 $\mu$mol kg$^{-1}$ (instead of 0.8 $\mu$mol kg$^{-1}$; Paulmier and Ruiz-Pino, 2009), which would promote the use of nitrate for oxidation instead of oxygen and therefore reduce oxygen consumption in suboxic environments.

The three sets of changes described above did not entirely remove the model low oxygen bias in the Bay of Bengal. One limitation of the COBALTv2 biogeochemical model is that it only includes one sinking detritus, which limits our ability to reproduce spatial contrasts in detritus sinking speed. Rixen et al. (2019b) showed that detritus sinking velocities are indeed higher in the Bay of Bengal due to the ballasting effect of riverine mineral particles. In addition, Al Azhar et al. (2017) showed that simulating this contrast between the Arabian Sea where detritus are sinking relatively slowly and the Bay of Bengal where detritus are sinking faster improved the representation of the OMZs in an ocean model. Looking ahead, adding multiple detritus pools with different sinking velocity might be a way to improve the OMZ in the Bay of Bengal. Despite these model limitations, it is important to note that uncertainties remain regarding the strength of suboxia in the Bay of Bengal. Recent observations from Argo floats and ship-based in situ measurements have reported lower oxygen concentrations in the Bay of Bengal than those presented in the WOA dataset, including nanomolar-level oxygen conditions (Bristow et al., 2017; Udaya Bhaskar et al., 2021). These findings suggest that the true extent and intensity of hypoxia in the Bay of Bengal remain uncertain, making it difficult to definitively assess the magnitude of the model bias in this region.

In addition to the OMZs, another area that we are considering for future work is the high bias in surface chlorophyll concentration simulated in the model compared to satellite products, in particular near and offshore summer upwelling systems. An extensive compilation of *in-situ* primary productivity measurements shows that the model successfully captures the seasonality in productivity. This strongly suggests that the bias is limited to the phytoplankton chlorophyll content without influencing its carbon content. This bias in chlorophyll is likely due to an overestimation of the contribution of large phytoplankton (higher chlorophyll to carbon ratio) compared to small phytoplankton (lower chlorophyll to carbon ratio), and is therefore expected to have a relatively small impact on nutrient uptake by phytoplankton and oxygen consumption associated with the remineralization of the organic matter in the water column. This bias in chlorophyll content might be mitigated in the future when using the COBALT version 3 biogeochemical module which incorporates four phytoplankton groups instead of three, including a medium size class that allows for a smoother transition from small to large, and accounts for photoacclimation and photoadaptation which is critical in simulating chlorophyll (Stock et al., 2025). While this bias complicates comparisons between model and satellite chlorophyll data, it is primarily confined to chlorophyll and has a limited impact on the model's ability to represent regional nutrient, carbon, and oxygen dynamics key to marine ecosystems.

With these results, we are confident that MOM6-COBALT-IND12 is an effective and versatile model to tackle applications in physical and biogeochemical oceanography, as well as applications to marine resources and management on timescales of weeks to decades in northern Indian Ocean. This configuration is particularly well-suited for evaluating the impacts of natural variability and anthropogenic activities on key environmental variables that influence marine resources. One key application is evaluating the risk of coastal hypoxia——an increasingly pressing issue for local populations and the blue economy, including fisheries, in the region (Naqvi et al., 2009; Vallivattathillam et al., 2017; Pearson et al., 2022; Naqvi, 2021, 2022). Despite

its importance, coastal hypoxia is often only marginally addressed in global studies, which focus primarily on hypoxia events
in Europe and North America (Breitburg et al., 2018; Deutsch et al., 2024). MOM6-COBALT-IND12 is ideally suited to investigate the physical and biological drivers of coastal hypoxia in the northern Indian Ocean, as well as their spatio-temporal variability. This capability is essential for predicting hypoxic events and informing effective management strategies to safeguard marine ecosystems and coastal economies.

# Appendix A: Supplementary Figures

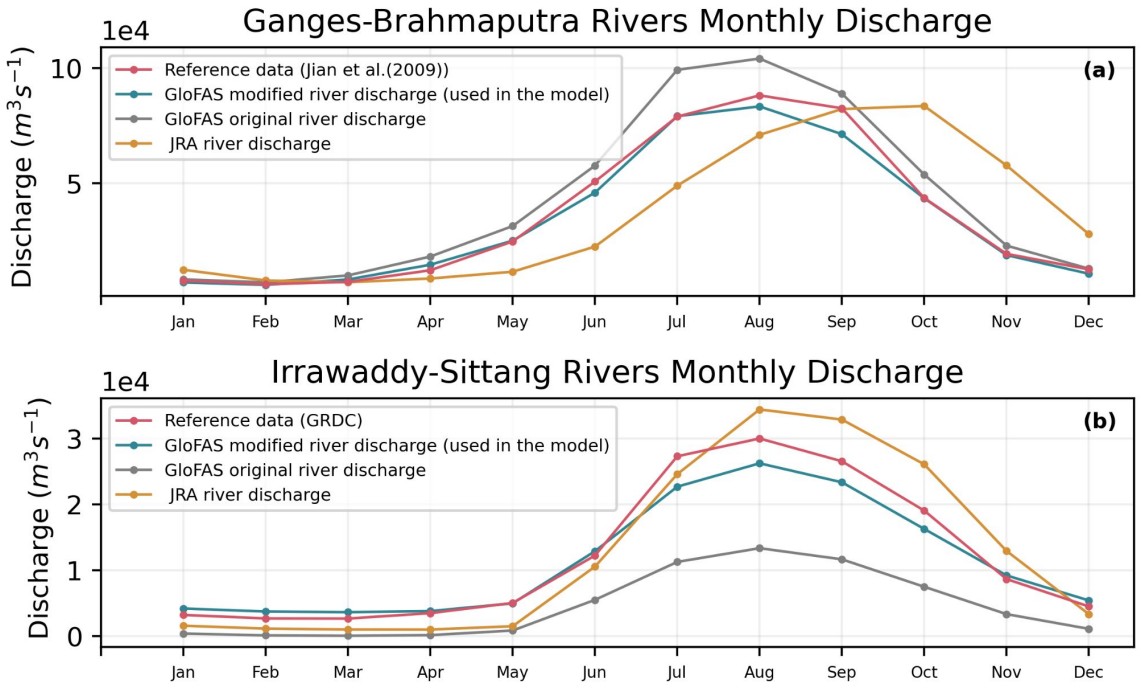

**Figure A1.** Water discharge in the (a) Ganges-Brahmaputra and (b) Irrawaddy-Sittang river systems from observations (red), the raw GloFAS-ERA5 runoff product (grey, Harrigan et al., 2023, 2020), the modified GloFAS-ERA5 runoff product used to force MOM6-COBALT-IND12 (teal, $0.75 \times$ GloFAS-ERA5 m$^3$/s for Ganges-Brahmaputra and $1.7 \times$ GloFAS-ERA5 $+ 3564$ m$^3$/s for Irrawaddy-Sittang), and in the JRA55-do reanalysis (orange, Tsujino et al., 2018). Observations are from Jian et al. (2009) for Ganges-Brahmaputra and Recknagel et al. (GRDC, 2023) for Irrawaddy-Sittang. We note that the raw GloFAS-ERA5 can overestimates or underestimate the discharge compared to observations, while JRA55-do presents a systematic 1-2 months delay in the timing of the seasonal peak runoff.

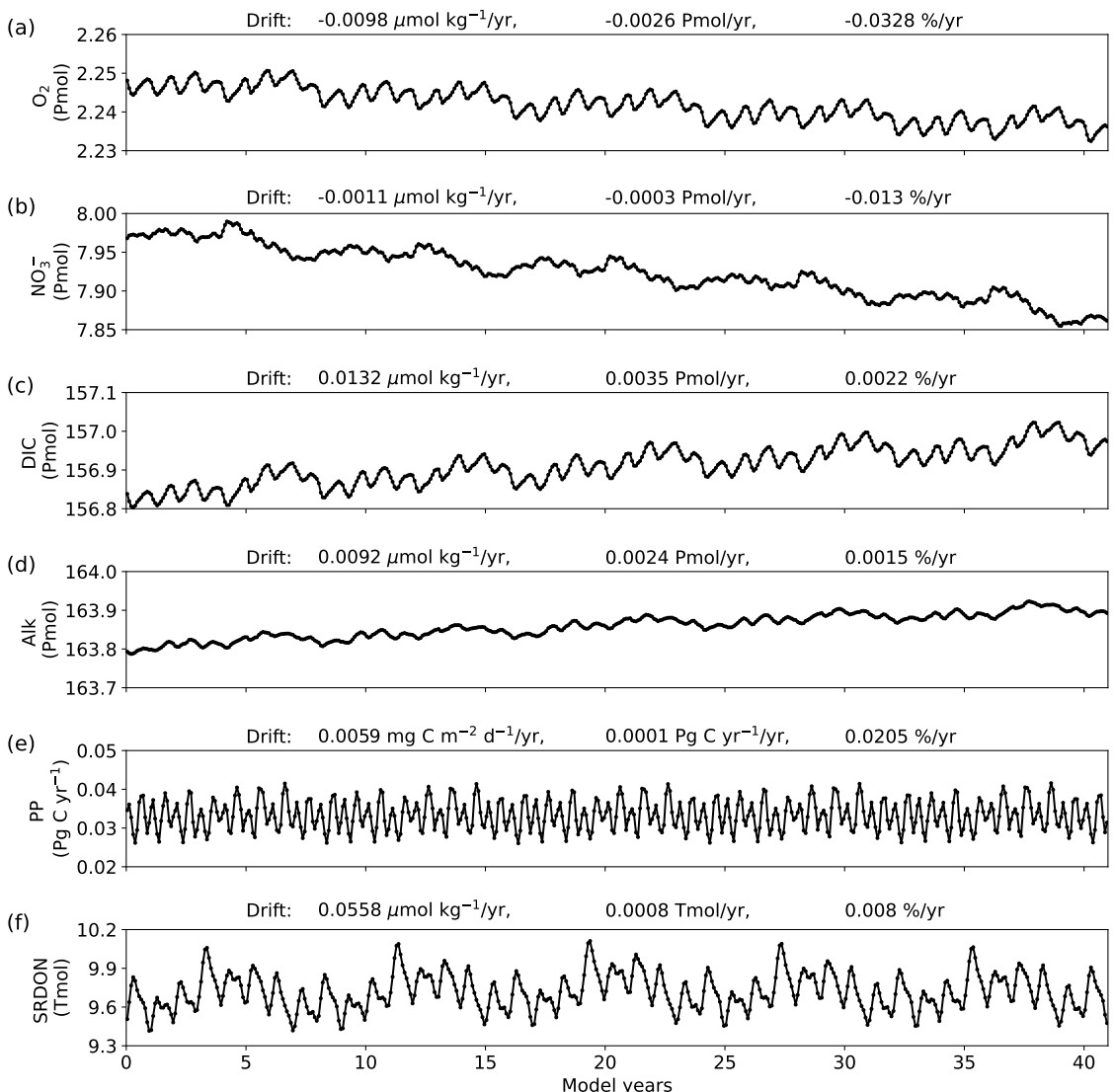

**Figure A2.** Model time series and drift evaluated as the linear trend after the 32-year spin-up in the control simulation with constant forcing: (a) total oxygen ($O_2$), (b) total nitrate ($NO_3^-$), (c) total dissolved inorganic carbon (DIC), (d) total alkalinity (Alk), (e) total vertically integrated primary productivity (PP) and (f) total semi-refractory dissolved organic nitrogen (SRDON). Drifts are indicated above each panel and are all $< 0.05\%$.

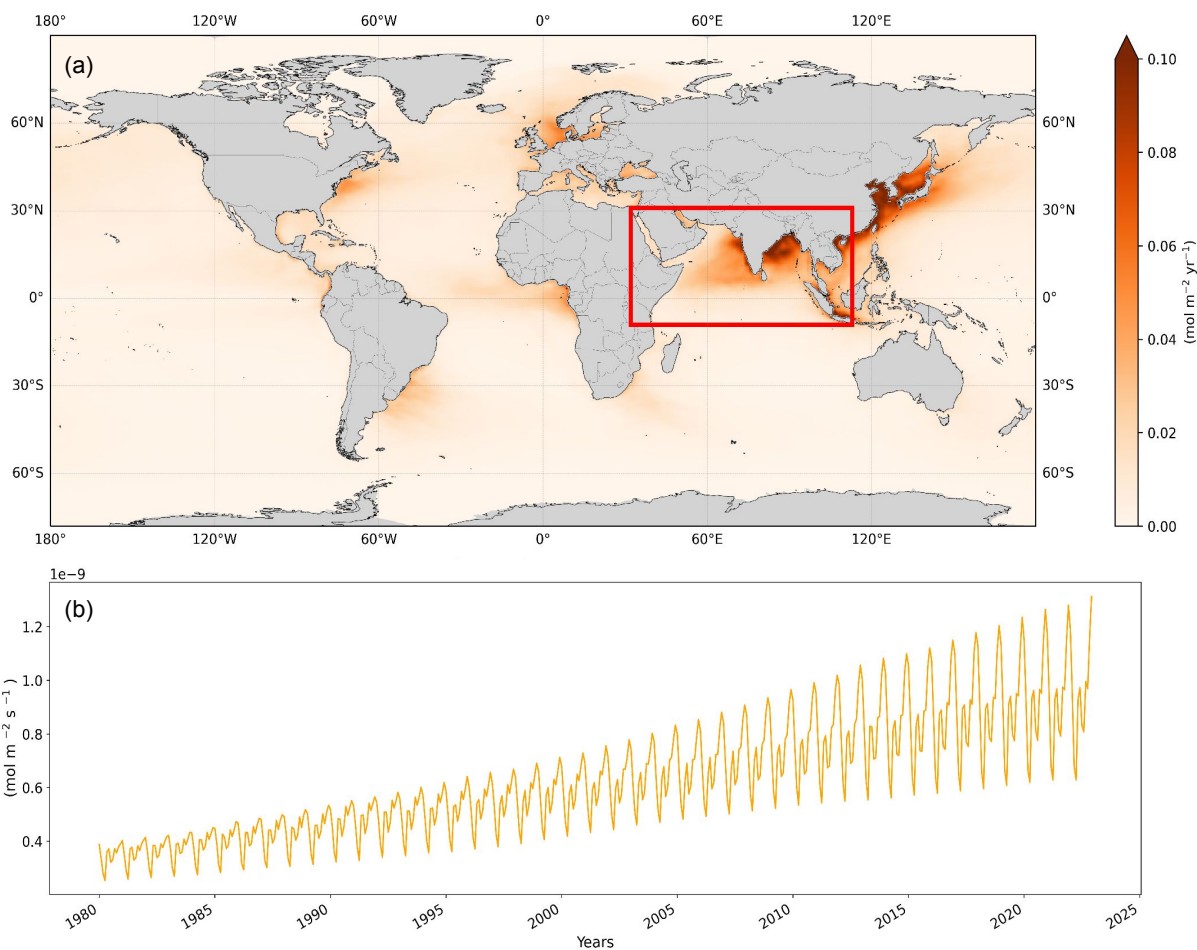

**Figure A3.** Atmospheric deposition of nitrogen from the earth system model ESM4.1 used to force MOM6-COBALT-IND12: (a) spatial distribution in year 2020, and (b) temporal evolution averaged over the model domain calculated using a 15-year monthly moving average (see Section 2.4.2).

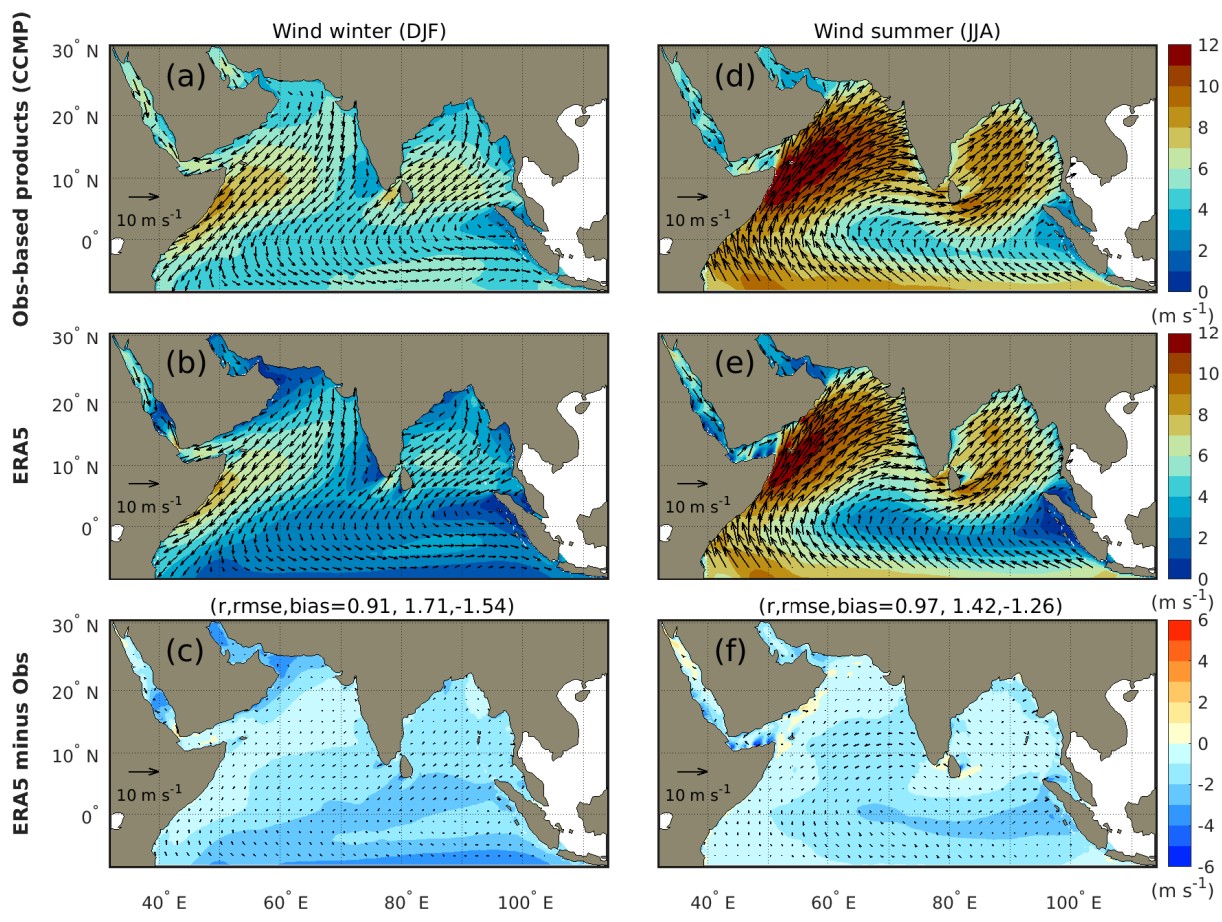

**Figure A4.** Surface wind (10m) during (a-c) winter (December-February) and (d-f) summer (June-August) monsoons. Panels (a,d) show Cross-Calibrated Multi-Platform (CCMP) satellite observation-based product, (b,e) show ERA5 data product and (c,f) show differences between ERA5 and CCMP. Correlation coefficients r, RMSD and bias between the data and model seasonal means are indicated. Wind observational data is from CCMP satellite (see details in Table 2). CCMP and ERA5 results are averaged over the 1993-2020 period.

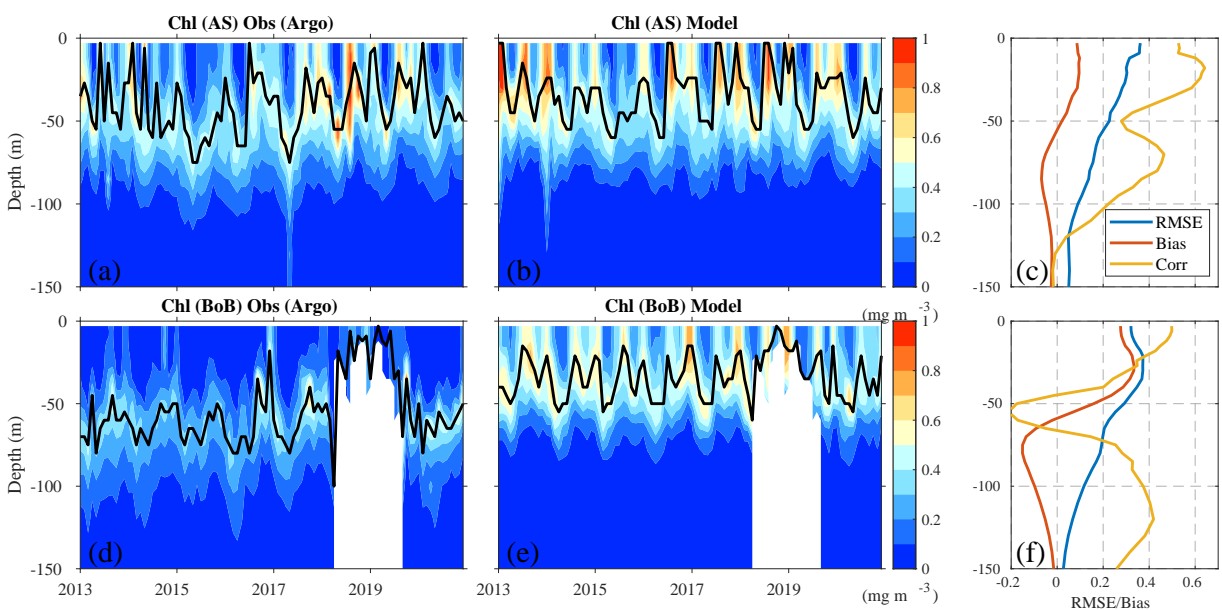

**Figure A5.** Comparison of observed and modeled vertical chlorophyll profiles in the Arabian Sea (a-c) and Bay of Bengal (d-f) using Argo float observations and model output. Panels (a,d) show Argo-derived chlorophyll concentrations; (b,e) show model-simulated chlorophyll concentrations; (c,f) show depth profiles of root mean square error (RMSE), bias, and correlation coefficient between model and Argo observations. In panels a-b and d-e, the black contour line indicates the depth of the subsurface chlorophyll maximum (SCM). Chlorophyll concentrations are shown in mg m$^{-3}$.

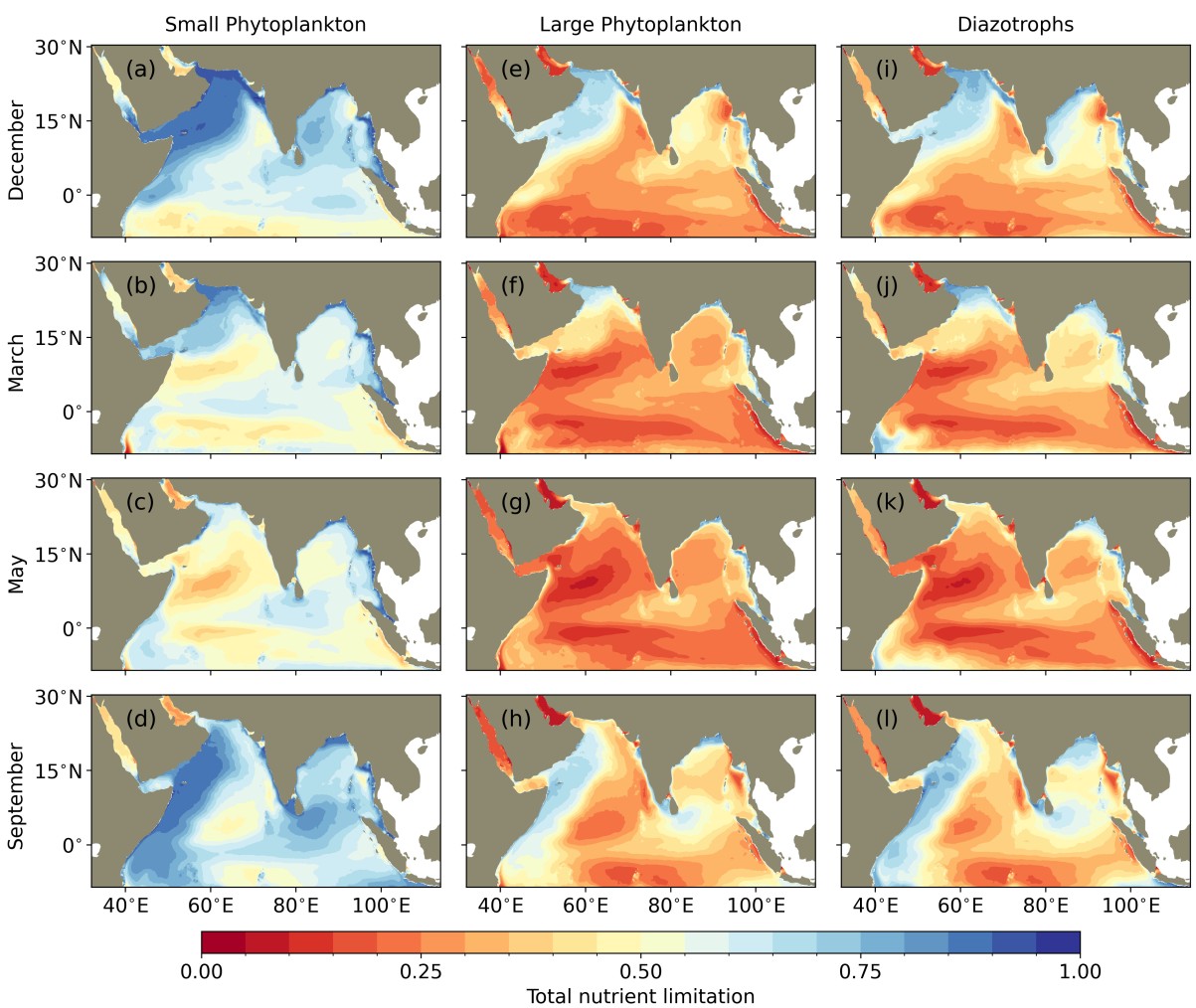

**Figure A6.** Climatological surface total nutrient limitation (nitrogen N, phosphorus P and iron Fe) following Liebig's Law of the Minimum in MOM6-COBALT-IND12 for small phytoplankton, large phytoplankton and diazotrophs in December, March, May and September. Model climatology is for 1980-2020. A value of 1 indicates no growth limitation by nutrients, whereas a value of 0 indicates complete growth limitation by nutrients.

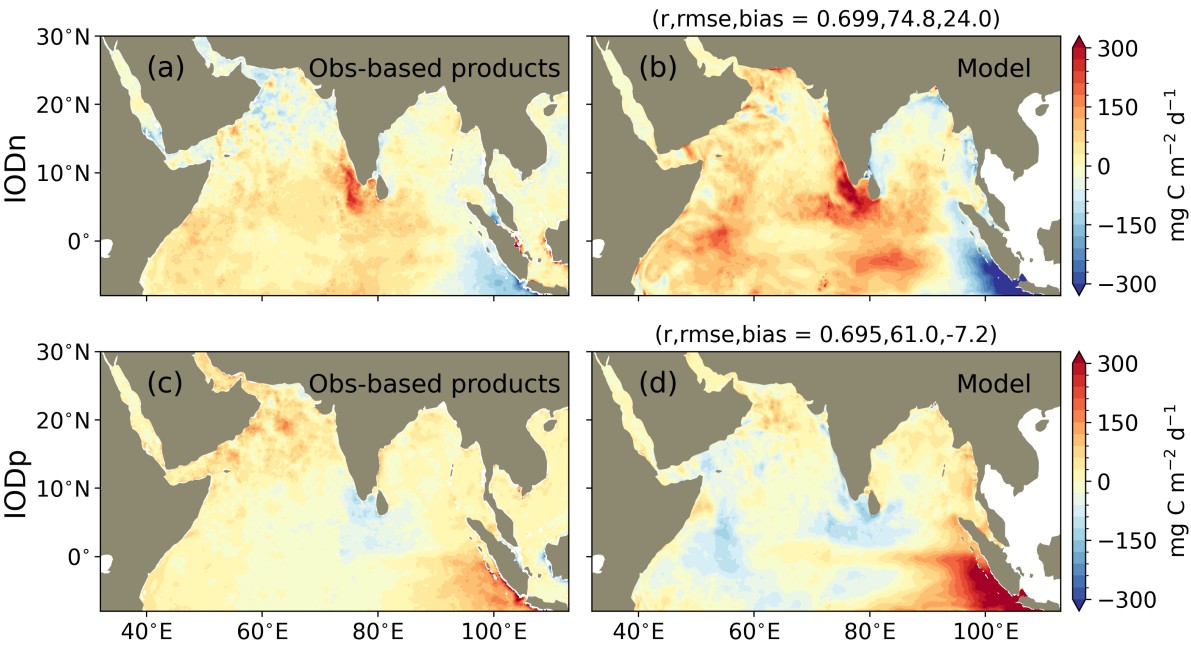

**Figure A7.** Integrated net primary productivity (PP) anomaly associated with the Indian Ocean Dipole (IOD). September-to-November PP composites during a-b) IOD negative phases and c-d) IOD positive phases in observation-based product and MOM6-COBALT-IND12. Composites are for September-to-November months available in the CAFE satellite product for positive (2002, 2006, 2015, 2018, 2019) and negative (2005, 2010, 2016) IODs (see Table 2 for details on data). Panels a and c of this figure showing the CAFE satellite product can be compared to Figure 21a and 21c showing the same composites but for the CbPM satellite product.

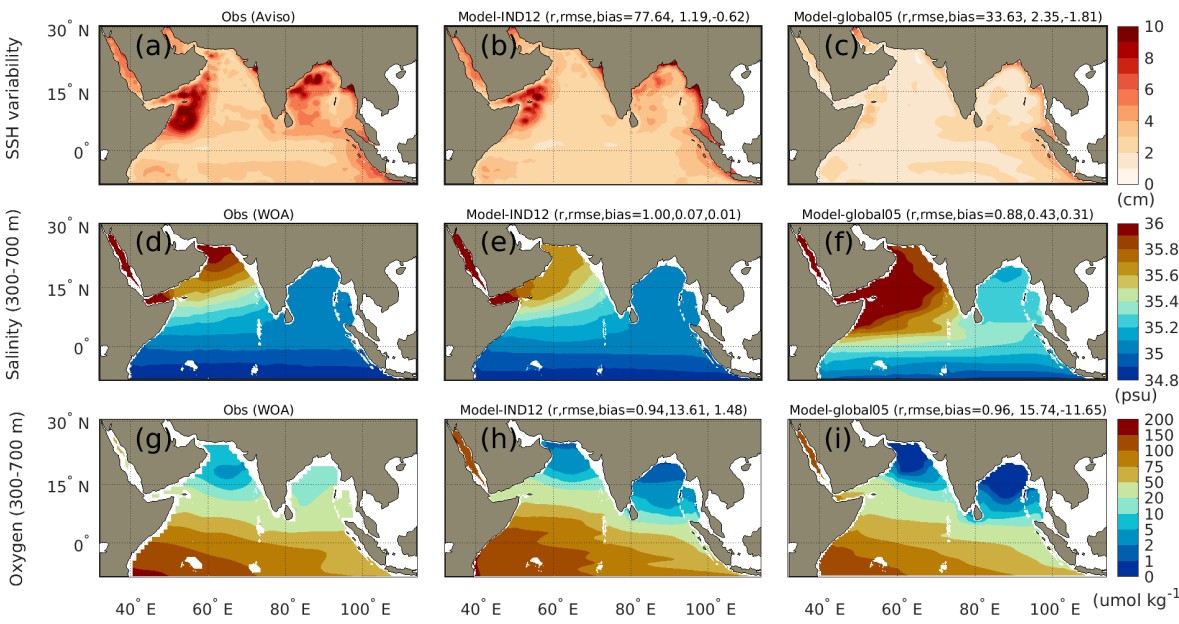

**Figure A8.** Comparison of sea level anomaly intraseasonal variability, subsurface salinity, and subsurface dissolved oxygen between observational products, the regional MOM6-COBALT-IND12 model (labeled as Model-IND12), and the global MOM6-COBALT configuration at 0.5° resolution (labeled as Model-global05). Panels (a–c) show the standard deviation of SLA representing intraseasonal variability (cm); panels (d–f) show the mean salinity averaged over 300–700 m depth (psu); and panels (g–i) show the mean dissolved oxygen averaged over 300–700 m depth. Comparison statistics—correlation coefficient (r), root mean square error (RMSE), and bias—are shown in parentheses. SLA intraseasonal variability is computed as the standard deviation of linearly detrended SLA, filtered using a 14–120 day band-pass filter.

*Code availability.* The source code for the model components is available at https://doi.org/10.5281/zenodo.14184011 (Liao et al., 2024a). The model parameter files and preprocessed forcing data used for the Indian Ocean configuration have been archived at https://doi.org/10.5281/zenodo.14171404 (Liao et al., 2024b). MOM6 is developed openly, with its Git repositories hosted at https://github.com/mom-ocean/MOM6 and https://github.com/NOAA-GFDL/MOM6. These platforms enable users to obtain the latest and experimental versions of the source code, report issues, and contribute new features.

*Data availability.* The model output analyzed in this study is available at https://doi.org/10.5281/zenodo.14183131 (Yang et al., 2024). The datasets used for model validation and comparison are listed as follows: OISSTv2.1 (https://www.ncei.noaa.gov/products/optimum-interpolation-sst, Reynolds et al. 2007), mixed-layer depth (https://mld.ifremer.fr/Surface_Mixed_Layer_Depth.php, De Boyer Montégut et al. 2004), surface currents (https://podaac.jpl.nasa.gov/dataset/OSCAR_L4_OC_INTERIM_V2.0, ESR 2009), sea level anomaly (https://doi.org/10.24381/CDS.4C328C78, Lopez 2018), CCMP wind speed (https://podaac.jpl.nasa.gov/MEaSUREs-CCMP, Mears et al. 2022), OC-CCI v5.0 (https://climate.esa.int/en/projects/ocean-colour/data/, Sathyendranath et al. 2019), RAMA temperature and salinity (https://www.pmel.noaa.gov/gtmba/pmel-theme/indian-ocean-rama, McPhaden et al. 2009), net primary productivity (https://orca.science.oregonstate.edu/index.php, Westberry et al. 2008), and World Ocean Atlas 2018 (https://www.ncei.noaa.gov/archive/accession/NCEI-WOA18, Garcia H.E. et al. 2019).

The datasets used to create the model forcing are listed as follows: ORAS5 reanalysis (https://cds.climate.copernicus.eu/datasets, Zuo et al. 2019), TPXO9 (https://www.tpxo.net/home, Egbert and Erofeeva 2002), GloFAS (https://doi.org/10.24381/cds.a4fdd6b9, Zsoter 2019), and ERA5 (https://cds.climate.copernicus.eu/datasets, Herbert et al. 2018).

*Author contributions.* Conceptualization: LR. Data curation: EL, FY, YZ. Formal analysis: EL, FY, YZ, SD, MM, LR. Funding acquisition: LR, CS. Investigation: EL, FY, YZ, SD, MM, LR. Methodology: EL, JP, SD, FY, YZ, ACR, RH, CS, LR. Software: EL, JP, SD, FY, YZ, ACR, MM, RH, CS. Supervision: LR, RH, CS. Visualization: EL, FY, YZ, SD, MM. Writing – original draft: LR. Writing – review and editing: LR, SD, FY, YZ, CS, RH.

*Competing interests.* No competing interests

*Acknowledgements.* This work was supported by the NOAA Climate Program Office Award No. NA21OAR4310119, and the High Meadows Environmental Institute Climate, Chemistry and Life in the Indian Ocean Initiative and the NSF CAREER award no. 2042672. The bio-Argo data were collected and made freely available by the International Argo Program and the national programs that contribute to it. (https://argo.ucsd.edu, https://www.ocean-ops.org). The Argo Program is part of the Global Ocean Observing System.

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
