# Peer review of "A high-resolution physical-biogeochemical model for marine resource applications in the Northern Indian Ocean (MOM6-COBALT-IND12 v1.0)"

_EGUsphere, 2024_

## Community Comment (CC1)

**General comments:**

This article is discussing the development and validation of an ocean-bio-geochemistry model customized for the north Indian Ocean. The authors make a good effort to get the simulations done and for the validation, and is publishable. Modelling ocean bio-geochemistry is very difficult and many models still struggle to get the bio-geochemistry right in those simulations. However, I have some concerns, which need to be addressed before it can be accepted.

We sincerely thank the reviewer for the constructive and thoughtful comments, which helped us improve the quality and clarity of the manuscript. We have prepared revisions to address all of these comments. The main reviewer's concern is the bias in surface chlorophyll compared to satellite observations. In our response, we show that our chlorophyll biases are within the range of previously published models, and argue that our model biases in primary production – which is a more robust indicator of biological nutrient and carbon uptake than chlorophyll – are much lower than the chlorophyll biases. The other comments are requests for additional points of evaluation or clarification of the model configuration. We have prepared all additional analysis required to address these comments, and the model performs well on these evaluations. Specifically, we propose to:

- Include an assessment of wind forcing using ERA5 and validate it against CCMP winds and RAMA moorings, along with an expanded discussion and an additional figure (major comment #1).

- Expand the discussion of surface and subsurface chlorophyll biases, comparing model performance with other regional studies (e.g., Sunanda et al., Chakraborty et al.), and presenting depth-dependent validation using bio-Argo data (major comment #2, 3, 8, 9).

- Clarify the oxygen minimum zones evaluation, supported by volumetric and concentration-based diagnostics (major comment #4).

- Clarify the simulation setup, including spin-up duration and equilibration, boundary conditions, atmospheric forcing, as well as references and criteria for rescaling river discharge and assigning riverine lithogenic concentrations (major comment #6; minor comment #9, 12, 14, 16).

- Clarify what we developed vs. what we customized in the model configuration, and add model performance metrics (major comment #6).

- Implement minor edits and clarifications are proposed to enhance precision and consistency throughout the text, as detailed in the point-by-point response (major comment #5, 7; minor comment #1-28).

Please see the detailed response below.

**Major comments:**

1. I do not see any wind simulation and its validation in the model. Since the monsoonal currents dictate the dynamics and associated processes in NIO, the wind simulations and their assessment are very important and must be presented in the main text.

   Our model is a physical-biogeochemical coupled, ocean-forced model. The atmospheric forcing — including wind, precipitation, and solar radiation — is derived from the 1/4° horizontal resolution European Centre for Medium-Range Weather Forecasts (ECMWF) Reanalysis 5th Generation (ERA5) at 1-hour frequency (Hersbach et al., 2020). ERA5 is widely recognized as one of the highest-quality atmospheric reanalysis products. Numerous studies have employed ERA5 to drive ocean models in the Indian Ocean, capturing key features such as the monsoonal seasonality of ocean waves (Sreelakshmi and Bhaskaran, 2020) and ocean circulation (Vogt-Vincent and Johnson, 2023). See details in section 4.1 in Screelakshmi and Bhaskaran (2020) and sections 4.2-4.3 in Vogt-Vincent and Johnson (2023).

   We evaluated ERA5 winds in the Indian Ocean by comparing them with the Cross-Calibrated Multi-Platform (CCMP) wind product, an observationally constrained dataset derived from satellite measurements, to ensure the robustness of the atmospheric forcing used in our model (new Figure R1). The comparison indicates that the ERA5 wind forcing reproduces the seasonal cycle and spatial distribution of summer and winter monsoons. To assess interannual variability, we further propose to compare ERA5 wind data with in situ observations from two RAMA mooring stations. The root-mean-square errors (RMSE) at these stations are 0.25 m/s and 1.39 m/s, respectively. These relatively small errors indicate that ERA5 forcing captures not only the seasonal dynamics but also the interannual variability of wind fields over the Indian Ocean.

   In addition, Section 4 of the current manuscript assesses the seasonal dynamics of key oceanic and biogeochemical features, including sea surface temperature, upper ocean circulation, coastal upwelling and downwelling, sea surface salinity, river plumes, and plankton bloom. These features are well reproduced in comparison with available data products, providing further evidence that the ERA5 forcing captures the seasonal dynamics of the Indian monsoon system. The details of atmospheric forcing setup was described in Lines 109-110 in the manuscript. We propose to add the wind evaluation using the new Figure R1 in the revised manuscript.

   The proposed clarified text reads: "**In addition, a comparison between ERA5 and CCMP wind products demonstrates that ERA5 wind forcing effectively captures the seasonal cycle and spatial distribution of the summer and winter monsoons (Appendix Figure A6).**"

[Figure]

**Figure R1.** Surface wind (10m) during (a-c) winter (December-February) and (d-f) summer (June-August) monsoons. (a,d) CCMP satellite data, (b,e) ERA5 data product and (c,f) differences between ERA5 and CCMP. Correlation coefficients r, root-mean squared error (rmse) and bias between ERA5 and CCMP seasonal means are indicated. Wind data is from CCMP satellite (see details in Table 2). ERA5 results are averaged over the 1980-2020 period.

2.  I thought, generally the models have relatively good simulations for surface Chl-a and compare well with the satellite and Argo data, which is not the case here in your model simulations. It is good to have a discussion based on other model simulations for NIO and other oceanic regions for Chl-a comparisons. Please see this and mention such model simulations and validation: https://doi.org/10.1016/j.ocemod.2024.102419

We emphasize in our manuscript that primary productivity is a better metric of biological production, nutrient and carbon uptake. In this regard, our model performs very well in comparison with observations (Section 4.5 and Figure 10). The seasonal patterns of PP are consistent with satellite-derived estimates and in-situ observations (351 stations). The model captures the magnitude of the double bloom productivity in the central and western Arabian Sea (about 1000-1500 mg C m$^{-2}$ d$^{-1}$ in CAS and WAS), as well as the lower productivity observed in the Bay of Bengal (<1000 mg C m$^{-2}$ d$^{-1}$, Figure 10 a,b,e,f).

Regarding the chlorophyll bias in our and other models. We carefully reviewed other biogeochemical modeling studies in the northern Indian Ocean (e.g., Sunanda et al., 2024; Chakraborty et al. 2023; Gutknecht et al. 2016). The biases in our model fall within the range of these other studies. Chakraborty et al. (2023) reported a RMSE of ~0.7 mg m⁻³ in the western and eastern Indian Ocean coast, similar to our model which yields a range from 0.62 (winter) to 0.93 (summer) mg m⁻³. Gutknecht et al. (2016), using the NEMO–PISCES model, reported a domain-mean chlorophyll concentration of 0.53 mg m⁻³ for the Indonesian Throughflow (ITF) region, compared with a MODIS-derived mean of 0.30 mg m⁻³. Although our model covers a different domain (north Indian Ocean), its mean chlorophyll concentration is 0.44 mg m⁻³, versus an OC-CCI observational mean of 0.34 mg m⁻³.

As suggested, we also made a comparison to Argo Chlorophyll showing depth-dependent statistical metrics, including RMSE, bias, and correlation coefficient (see new Figure R2 below). The RMSE between our model and Argo ranges from 0.03 to 0.37 mg m⁻³ ni upper ocean in the Arabian Sea and the Bay of Bengal, which is lower than the RMSE between our model and satellite products listed above, but higher than the RMSE reported for the model of Sunanda et al. 2024 (0–0.1 mg m⁻³).

The potential sources of the chlorophyll bias were discussed in section 4.5 of our manuscript. Notably, large phytoplankton are characterized by a higher chlorophyll-to-carbon ratio, whereas small phytoplankton have a lower ratio. Given that the model's carbon-based estimates (primary production) align well with the observational data, we propose that the overestimation of chlorophyll arises from an overrepresentation of large phytoplankton, which contribute disproportionately to the chlorophyll signal. However, the strong agreement of our model with observations of PP gives us confidence in the model's representation of biogeochemical dynamics despite moderate biases in chlorophyll.

We propose to expand the chlorophyll assessment and discussion by including Figure R2 in the revised manuscript.

- The proposed revised text in Line 362 reads: "The model also simulates the subsurface chlorophyll maximum captured by Argo floats in both the Arabian Sea and Bay of Bengal, with RMSE values over the vertical ranging from 0.03 to 0.37 mg m−3 (Appendix Figure A5). This suggests that the model effectively represents the vertical distribution of plankton and associated subsurface biological dynamics. Overall, comparison of our model's mean bias and RMSE with values reported in previous studies suggests that our chlorophyll simulation performance falls within the median range relative to other regional biogeochemical models (Chakraborty et al., 2023; Gutknecht et al., 2016; Sunanda et al., 2024)".
- The proposed revised text in Lines 370 reads (new in bolded): "The fact that the model simulates the magnitude of observed PP (in carbon units) but overestimates the surface chlorophyll content suggests that it might overestimate the contribution of large phytoplankton, **which is characterized by a high chlorophyll-to-carbon ratio,**

compared to small phytoplankton, **characterized by a lower chlorophyll-to-carbon ratio**. **This overestimation of the contribution of large phytoplankton to the assemblage would indeed explain the good match in primary productivity and high bias in chlorophyll**".

[Figure]

**Figure R2**. Comparison of observed and modeled vertical chlorophyll profiles in the Arabian Sea (AS, top row) and the Bay of Bengal (BoB, bottom row) using Argo float observations and model output. (a, d) Argo-derived chlorophyll concentrations; (b, e) model-simulated chlorophyll concentrations; (c, f) depth profiles of root mean square error (RMSE), bias, and correlation coefficient between model and Argo observations. In panels a-b and d-e, the black contour line indicates the depth of the subsurface chlorophyll maximum (SCM). Chlorophyll concentrations are shown in mg m$^{-3}$.

3. There is a subsurface maximum for Chl-a in the NIO. Please show the model simulations and comparison with Argo measurements.

   Thank you for the suggestion to compare the model to Argo. As detailed in comment #2, we propose to evaluate the modeled subsurface chlorophyll structure using Argo float observations in the revised manuscript (Figure R2). See the proposed revised text and figure in major comment #2.

4. L459: If the model overestimates, how that would affect the OMZ in AS and BoB? Which oceanic region has large OMZs? L463, you say that in AS, it is well represented, but BoB overestimated. Fig 16 provides no clue that any basin is better for this. Also, why the equatorial region has such a big difference?

   In observation-based products (WOA18 and Bianchi et al., 2012), the AS features an intense OMZ with subsurface oxygen concentrations around 5 µmol kg$^{-1}$, whereas the BoB exhibits a

much less intense OMZ, with subsurface oxygen concentrations typically between 10–20 µmol kg⁻¹. Figure 16 of our manuscript does compare the OMZ in the AS and BoB in products and the model. Panel a shows maps of O2 in model, products and the model bias. Panel b compares the OMZ volume in products and model for the full basin, the AS and the BoB for different oxygen thresholds (e.g., suboxia is defined by 5 umol/kg but other thresholds such as hypoxia are relevant for organisms etc.).

As described in section 5.2, the oxygen bias is higher in the BoB (approximately −10 to −20 µmol kg⁻¹) than in the AS (approximately −10 to +10 µmol kg⁻¹, Figure 16a). Additionally, we find that at the basin scale the model reproduces the volume of hypoxic (<60 µmol kg⁻¹) and low-oxygen (<100 µmol kg⁻¹) waters (Figure 16b). However, it overestimates suboxic waters (<5 µmol kg⁻¹), mainly due to the overprediction in the Bay of Bengal. This is visible in the lower right panel of Figure 16 when comparing the products (blue and red lines) to the model (black lines) at oxygen values of 5 umol/kg. We propose to clarify the description of Figure 16 in section 5.2 as follows: "In contrast, the model overestimates the volume of suboxic waters delimited by 5 umol kg-1 (0.17×1016 m3 vs. 0.06×1016 m3 in Bianchi et al. (2012) observations), mostly because of the large suboxic volume simulated in the Bay of Bengal (0.10×1016 m3 vs. 0.00×1016 m3 in observations, Figure 16b)".

We also propose to expand the results to include the western equatorial bias in section 5.2. It reads as: "In this region, the model shows a high oxygen bias near the base of the thermocline (500–1000 m), coinciding with a low nitrate bias (not shown). This pattern points to either a misrepresentation of biological remineralization at depth or an inaccurate representation of the relative contribution of the water masses forming the Central Waters supplying oxygen to this region. These waters originate from a blend of the Indonesian Throughflow (ITF) and southern-sourced Mode Waters. Previous studies have demonstrated that oxygen distribution in this region is highly sensitive to the relative contribution between these two sources (DItkovsky et al 2023). However, the scarcity of direct observations in this region limits our ability to conclusively attribute the model bias to either mechanism."

However, recent studies indicate an uncertainty regarding the true extent and severity of hypoxia in this region  (Bhaskar et al., 2021, Bristow et al., 2017). We therefore also propose to expand the discussion of the model oxygen bias in section 8 (Discussion and Conclusions) by adding the following text: "Despite these model limitations, it is important to note that uncertainties remain regarding the strength of suboxia in the Bay of Bengal. Recent observations from Argo floats and ship-based in situ measurements have reported lower oxygen concentrations in the BoB than those presented in the WOA dataset, including nanomolar-level oxygen conditions (Bristow et al., 2017; Bhaskar et al., 2021). These findings suggest that the true extent and intensity of hypoxia in the BoB remain uncertain, making it difficult to definitively assess the magnitude of the model bias in this region."

5. Write the validation information with bias values in the abstract

We propose to revise the abstract (Lines 12-13) as follows: "**Quantitatively, the model exhibits relatively small biases, as reflected by root mean square error (RMSE) values in key variables: surface temperature (0.25–0.30 °C), mixed layer depth (7–8.09 m), sea level anomaly (0.02 m), sea surface salinity (0.53–0.71 psu), vertical chlorophyll (0.03–0.3 mg m⁻³), subsurface temperature (0.33 °C), and subsurface salinity (0.07 psu)**".

6. Developed or customized the model, please make sure that you use a correct word for this

The global MOM6-COBALTv2 model was originally developed by GFDL scientists including coauthors in this paper. However, regional capabilities are new. We have developed the regional Indian Ocean configuration and customized the MOM6-COBALTv2 model by modifying several parameters (e.g., detritus sinking velocity, burial fraction, oxygen half-saturation for nitrification, oxygen constraint on water column denitrification), as well as rewriting portions of the input and output Fortran routines within regional MOM6-COBALTv2 to better suit the specific needs of our regional configuration and biogeochemical applications. This is not merely a simple customization; rather, we are deeply engaged in developing and refining the model to address bugs and enhance its functionality as a regional biogeochemical modeling tool.

- The proposed revised text in Lines 63-66 reads: "It is with these applications in mind that we **configured, customised and validated** the regional Indian Ocean simulation presented here based on the Modular Ocean Model 6 (MOM6, Adcroft et al., 2019) coupled with the Carbon, Ocean, Biogeochemistry, and Lower Trophics module version 2.0 (COBALTv2, Stock et al., 2014, 2020)."
- The proposed revised text in Line 569 reads: "We **configured, customised and validated** a regional ocean biogeochemical model at 1/12∘ horizontal resolution (MOM6-COBALT-IND12 v1.0) that captures most key features of the northern Indian Ocean dynamics."
- The proposed revised text in Line 589 reads: "During the **setup and customization** of the MOM6-COBALT-IND12 v1.0 model, we identified a series of physical and biogeochemical parameters and forcings that influenced the model simulation and led to a significant improvement of the results (see details in section 2)."

7. L336: reproduced is a "lighter" word; how good is the comparison? Please write some numbers.

We propose to revise as suggested. The comparison is already quantified in the text, with a regional correlation coefficient (r > 0.95) and RMSE < 0.7, as also shown in Figure 8. We propose to revise the text to present these metrics more clearly: "Performance metrics indicate that the simulation achieves **a strong spatial correlation (r > 0.95) and a small regional RMSE (0.53-0.7)**".

8. LK353: Why the Chl-a simulation is not good in the Somali coast or western Indian Ocean? Summer Chl-a is even worse?

9. L371-372: How did you arrive into this conclusion; is this about the size of the plankton?

These two comments are related to Comments #2 and #3 and we answer them together here. The potential sources of chlorophyll bias are discussed in detail in our responses to comments #2 and #3 and were also discussed in Section 4.5 of the manuscript. Large phytoplankton has a higher chlorophyll-to-carbon ratio and the model likely overestimates the contribution of this group to the assemblage. Again we want to emphasize that the model simulates primary production comparable with observations (Figure 10), which is a more important metric than chlorophyll for constraining nutrient, carbon and oxygen cycling.

The larger bias in the Somali Current, and summer specifically, is likely related to enhanced upwelling during this season, which brings nutrients to the surface. In addition, the strengthened coastal current driven by the southwest monsoon may further transport nutrients into the western Indian Ocean. This elevated nutrient availability facilitates larger phytoplankton growth and higher Chl content in the model. See the proposed revised text and figure in major comment #2.

**Minor comments:**

1. L4: north of 8S? It can be anywhere north of that latitude. Please be specific

   We propose to revise as suggested: "The model covers the northern Indian Ocean (**from 8°S to the northern continental boundaries**), central to the livelihoods and economies of countries that comprise about one-third of the world's population."

2. L22: and is missing

   We propose to revise as suggested: "The northern Indian Ocean is central to the livelihood and economy of about one third of the Earth's population which live in its littoral countries (e.g., India, Indonesia, Pakistan, Bangladesh, Tanzania, Myanmar, Malaysia, Kenya**, and** Yemen) ".

3. L23; separate the Roy citation from the bracket

   We propose to revise as suggested: "... provides valuable resources via the "blue economy", such as fishery, aquaculture, **marine tourism (Roy, 2019)**".

4. L40: about the NIO stressors: https://doi.org/10.1016/j.pocean.2023.103164

   We propose to add further discussion on stressors affecting the northern Indian Ocean under climate change, and cite two relevant studies: Sunanda and Chakraborty (2023) and Sunanda et al. (2021).

   The revised text in Line 45 will read: "Projections from Coupled Model Intercomparison Project (CMIP) models suggest substantial shifts in net primary production (NPP) and sharp declines in pH in the coming decades, **highlighting the North Indian Ocean's particular vulnerability to climate change (Sunanda et al., 2021; Sunanda and Chakraborty, 2023)**."

5. L53: models are "tools" for studying

   We propose to revise as suggested: "Models are powerful **tools** for exploring the Indian Ocean".

6. L55: this is another model validation for this region: https://doi.org/10.1016/j.ocemod.2024.102419

We propose to revise as suggested and cite one more model work in the revised manuscript. The proposed revised text will read: "... assess the impacts on biogeochemistry and ecosystems (e.g., Sengupta et al., 2001; Rahaman et al.,2014; Lachkar et al., 2018, 2019; Resplandy et 55 al., 2011, 2012; Schmidt et al., 2021; Ditkovsky et al., 2023; **Sunanda et al., 2024**)."

7. L85: coordinates of the region

We propose to revise as suggested: "MOM6-COBALT-IND12 is therefore considered an 'eddy resolving' model for the region **with a rectilinear and orthogonal grid ($32°E$ to $114°E$ and $8.6°S$ to $30.3°N$)**".

8. L113: salinity from 1998 data, any updated version?

The salinity restoring data used in this study is PHC2.1, released in 2002, which remains the version currently used by the Geophysical Fluid Dynamics Laboratory (GFDL), NOAA. A newer version, PHC3.0, became available in 2005. We plan to update the salinity restoring data to PHC3.0 or a similar newer product, such as WOA 2023, in future simulations.

9. L115: How long was the spin up and when did the model stabilize? Which year onward you analyse the model results for science?

We propose to clarify the spin-up stabilization and hindcast simulation by rewriting section 2.2.1 of the revised manuscript (modified text in bold):

"**The ocean model was initialized using temperature and salinity from annual mean fields from the World Ocean Atlas version 2013 (WOA13, Locarnini et al., 2014; Zweng et al., 2014). Our simulations were run using the atmospheric forcing from the $1/4°$ horizontal resolution European Center for Medium-range Weather Forecasts reanalysis 5th generation (ERA5) at 1-hour frequency (Hersbach et al., 2020).** The sea surface salinity (SSS) was restored to the polar science center hydrographic climatology (PHC2.1), which is based on the World Ocean Atlas 98 with data replenishment in the Arctic Ocean (Steele et al., 2001), with a piston velocity of $0.1667$ m d$-1$. **We conducted a 32-year spin-up, which was achieved by looping four consecutive 8-year loops of the 1980 to 1987 forcing field and reached a well-equilibrated state with minimal linear trends of physical and biogeochemical variables (e.g., drift in sea surface temperature, SSS, oxygen, nitrate, primary production and ocean surface partial pressure of carbon dioxide $p$CO$_2$ < ~0.1% for model years 17-32). Using outputs from the end of the spinup simulation as initial conditions, the hindcast simulation was started on January 1 1980 and was run from 1980 to 2020 for our analysis in this study**."

The minimal model drift in sea surface temperature, SSS, oxygen, nitrate, primary production and $p$CO$_2$ for model years 17-32 of the spin-up is shown in Figure R3 below for your reference.

[Figure]

**Figure R3**. Time series of domain-mean model variables during the 32-year spin-up forced by repeated atmospheric and open boundary conditions looping over 1980–1987. Panels show: (a) sea surface temperature (SST), (b) sea surface salinity (SSS), (c) oxygen ($O_2$), (d) total nitrate ($NO_3^-$), (e) primary productivity (PP), and (f) ocean $pCO_2$. Drifts over model years 17-32 are indicated above each panel and are all < ~0.1%

10. L119: citation format is not correct

We will correct the format. The proposed revised text reads: "Open boundary conditions (OBC) are set using the Flather formulation for the tidal and sub-tidal sea level and barotropic velocity and the Orlanski formulation for the baroclinic velocity (**Flather, 1976**; Orlanski, 1976)".

11. Figure 1: rivers can be in red color, to differentiate from the bathymetry blue color

We propose to revise as suggested. The updated figure is shown below.

[Figure]

**Figure R4**. Domain and bathymetry of the regional Indian Ocean MOM6-COBALT-IND12. Pink shading indicates the extent of sponge layers (see methods). Major rivers are indicated in red.

12. L147: any reference for this? Overestimation and scaling have got any criterion? Why 25%?

The overestimation and subsequent scaling of river discharge are based on our evaluation of the GloFAS dataset, as shown in Appendix Figure A4 in the current manuscript and discussed in section 2.2.3. We compared the river discharge from GloFAS with observational estimates for the Ganges–Brahmaputra river system reported by Jian et al. (2009). To address this bias and improve the realism of the freshwater input, we applied a 25% reduction to the GloFAS-derived discharge for the Ganges–Brahmaputra river system (see Figure A4). We also corrected the Irrawaddy-Sittang river system using a similar approach and data from the Global Runoff Data Center (Recknagel et al., 2023). These references are already cited in the original manuscript (Line 145 and Table 2) but we will also add them in Line 147 to improve clarity in Section 2.2.3.

The proposed revised text reads: "By comparing GloFAS to published discharge observations (Jian et al., 2009; Siswanto et al., 2023), we found that GloFAS overestimated discharge in the Ganges-Brahmaputra river system, and therefore scaled down the freshwater discharge by 25% to match observations in these two rivers (see Appendix Figure A4, Jian et al., 2009, Siswanto et al., 2023)".

13. L175: not from WOA 2023?

The nutrient and oxygen fields were initialized using data from the World Ocean Atlas (WOA) 2018, as the project began before the release of WOA 2023. We plan to incorporate the updated WOA 2023 fields in future model simulations.

14. L176: CO2 is increasing, so the old climatology values are good?

The initial DIC field used for the model spin-up is derived from the GLODAP climatology, which reflects approximately the year 2002 conditions, higher indeed than the 1980 conditions of our initial state. However, the 32-year spin-up with repeated forcing fields from 1980–1987 (air $CO_2$ levels and DIC concentration at open boundary conditions are also from 1980-1987) yielded a stabilized surface ocean $p\mathrm{CO_2}$ lower than its initial values, toward the end of the spin-up period with a linear trend of ~0.05% (see Figure R3 above), indicating minimal drift. We propose to clarify this point in section 2.4.1 of the revised manuscript when describing the initialization of the model spin-up and model drift (added text in bold):

"**For the model spin-up**, nutrients (nitrate, phosphate, and silicate) and oxygen were initialized using annual means from the World Ocean Atlas 2018 (WOA18, Garcia et al., 2019). DIC and alkalinity were initialized using annual means from GLODAPv2 **which are representative of year 2002** (Olsen et al., 2016). … Model drift after the 32-year spin-up and over the 41 years of a control simulation with constant forcing is small, with linear trends < 0.05% for oxygen, nitrate, DIC, alkalinity, semi-refractory dissolved organic nitrogen pools and integrated primary productivity (see Appendix Figure A1). **The slight drift indicates that the hindcast simulation starts from a well-equilibrated initial state provided by the spin-up simulation**."

15. L192: SSP 5-8.5 is an extreme case. So how much that would influence your simulations?

The atmospheric deposition data were taken from the GFDL ESM4.1 historical simulation for the period 1980–2014 and from the SSP5-8.5 scenario of the same model for the period 2015–2020. We selected the SSP5-8.5 pathway because it is regarded as a "business-as-usual" scenario, and our current society did not largely reduce carbon emission in the past few years (i.e., 2015-2020). Therefore, it is reasonable to still consider the past few years as the "business-as-usual" scenario. This is the reason we select the SSP5-8.5 data for the period 2015-2020.

In addition, we assess the consistency between the historical and scenario-based datasets. As shown in Figure A2 in the current manuscript, there is no noticeable discontinuity between the two periods in either magnitude or trend of nitrate deposition. Therefore, the use of SSP5-8.5 for the 2015–2020 period does not introduce artificial discontinuities and is not expected to significantly influence the model simulation results.

16. L223: How the adjustments are made? Just random or any criterion followed?

The adjustments to riverine lithogenic concentrations are described in the manuscript (Lines 226-228) and the supporting observational dataset (Milliman and Farnsworth, 2011) is listed in Table 2 under the "Riverine lithogenic flux" entry. We will clarify this point in the revised

manuscript at the beginning of the lithogenic input description by adding the following sentence (**added text in bold**): "**To reflect spatial differences in sediment supply, we specify riverine lithogenic concentrations based on observational data from Milliman and Farnsworth (2011):** the lithogenic input from rivers was adjusted to 200 g m$^{-3}$ for major rivers (i.e., rivers with sediment loads exceeding 10 Mt y$^{-1}$, e.g., Godavari, Krishna, Ganges, Brahmaputra, Irrawaddy, Sittang, Salween, Indus, Tapti and Narmada rivers, see Figure 1 for rivers location) and 20 g m$^{-3}$ for all other rivers, rather than applying a global constant of 13 g m$^{-3}$ used for all rivers in Stock et al. (2020). These adjustments account for the significantly higher total suspended sediment loads in these rivers (Milliman and Farnsworth, 2011; Rixen et al., 2019b), and are supported by river observations from Milliman and Farnsworth (2011) showing a broad range from 10 g m$^{-3}$ (Muvattupuzha River) to 1,061 g m$^{-3}$ (Ganges River)".

17. L243: citation format is not correct

We propose to edit as suggested: "temperature and salinity from the World Ocean Atlas 2018 (WOA18, **Garcia et al., 2019**)."

18. L274: SST has been already defined

We propose to edit as suggested: "Patterns of **SST** in the northern Indian Ocean follow the well described basin-scale features."

19. L276: particularly and especially, Please rephrase

We propose to edit as suggested: "MOM6-COBALT-IND12 captures the seasonal SST patterns **well, notably** the contrast between the vast warm pool (SST >28°C) covering most of the basin and the colder SST regions that emerge in response to seasonal variations in atmospheric and oceanic circulation (Figure 3)."

20. Figure 4: Why summer MLD is bad in the model?

We propose to expand the evaluation of winter and summer MLD in section 4.1 by adding the following paragraph: "The model captures the seasonal contrast in mixed layer depth (MLD) between the Arabian Sea and the Bay of Bengal, with deeper mixed layers in the Arabian Sea and shallower layers in the Bay of Bengal during both winter and summer (Figure 4). The MLD is generally deeper in summer than in winter. The spatial patterns, including the locations of local MLD maxima, are broadly consistent with observational data. Quantitatively, the basin-wide correlation values are similar between the two seasons, although the RMSE is larger in summer (8.09 m) than in winter (7.00 m). One possible contributor to the larger summer bias is the enhanced wind forcing during the monsoon season (see Figure A6), which intensifies turbulent mixing and deepens the mixed layer. At the same time, the MOM6 model includes the mixed layer eddy (MLE) parameterization of Fox-Kemper et al. (2011), which represents restratification driven by baroclinic eddies within the mixed layer. This restratification process may also be more active in summer, potentially leading to an overcorrection that offsets vertical mixing too strongly. The interaction between intensified wind-driven mixing and enhanced

restratification may thus contribute to the larger MLD bias observed in summer compared to winter."

21. L337: SSS has been defined already

We will remove and edit as suggested: "The model reproduces the main observed patterns of SSS, including the high **SSS** (>34) in the Arabian Sea".

22. L350: Narmada-Tapti

We will correct as: "The GloFAS runoff product captures the discharge of one of the main river systems for which we have direct observations, i.e., the **Narmada-Tapti** rivers."

23. Fig 17: How that affects simulations of SLA?

As shown in Figure 17, the model underestimates the amplitude of SSH variability in the 14–120 day/cycle range compared to AVISO observations. This frequency band corresponds to intraseasonal, including Kelvin and Rossby waves related to high-frequency atmospheric forcing responses and other ocean processes. As a result, the magnitude of sea level anomalies associated with these waves may be underestimated, and their influence on the biogeochemical system could be underrepresented in the simulations.

Nevertheless, the model still performs well in reproducing most of the key variabilities in both physical and biogeochemical processes. As demonstrated in Section 4.3 and Figure 7, the model captures the seasonal cycle of sea level anomalies with high fidelity, yielding a strong correlation ($r = 0.93$) and low RMSE ($0.02$ m) relative to observations. Therefore, while the model underestimates SLA variability in the high-frequency band, this has only a minor impact on the overall accuracy of SLA simulation, which is largely governed by lower-frequency components.

24. L495-500: remarkably well? Not sure, if you look at the SLA figure.

We propose to rephrase the text to ensure consistency between the descriptions of the SLA and chlorophyll simulations. Specifically, we propose to rephrase the sentence as the model "reproduces the fine-scale features structuring the winter and summer blooms," rather than describing the performance as "remarkably well". The revised text will read: "The model **reproduces the fine-scale features** structuring the winter and summer blooms".

25. L514: IOD has been defined, as for L528: RAMA, OISST

We propose to remove and edit as suggested.

- L514, the revised text will read: "The model reproduces the amplitude and zonal pattern of SST changes expected in response to the Interannual **IOD**". Line 530 in the revised manuscript.

- L528, the revised text will read: "We use observations from the in-situ **RAMA** that we complement with **OISST** data and Argo float-based thermocline depth." Line 543 in the revised manuscript.

26. 579-580: model is good because of its good bio-geochemistry simulations? What about the model physics?

The text here intends to emphasize that the model's strong physical performance provides the foundation for accurately simulating ocean biogeochemical and biological responses. The physical component of the model has been thoroughly evaluated, as detailed in Sections 2–7, which assess circulation variability, meridional temperature and salinity profiles, seasonal and intraseasonal sea level variability, and interannual SST variability. The well physical model performance supports its capacity to accurately represent the biogeochemical and biological variabilities.

We propose to rephrase the text for clarity: "The good agreement between observed and modeled physical features **provides a foundation for accurately simulating** the ocean biogeochemical and biological response".

27. L584: a detailed account of winter blooms are here: https://doi.org/10.1016/j.jenvman.2023.117435

We propose to revise as suggested. We plan to cite the suggested winter bloom work in the revised manuscript. The proposed revised text will read: "Specifically, the model reproduces the summer bloom associated with coastal upwelling systems and their extension offshore in mesoscale filaments, as well as the winter bloom associated with convective mixing and modulated by fine-scale eddies (Lévy et al., 2007; Resplandy et al., 2011, 2012; Mahadevan, 2016; Lachkar et al., 2016; Rixen et al., 2019a; Vinayachandran et al., 2021; **Anjaneyan et al., 2023**)".

28. L588: different response? please be specific

We propose to specify the detailed responses reported in Wiggert et al. (2009). The proposed revised text will read: "This includes the modulation of the production in the equatorial region, the Arabian Sea and around the tip of India, although we note these patterns are difficult to generalize to all IOD events. **As illustrated by Wiggert et al. (2009) who showed the chlorophyll responses vary with IOD intensity in most regions, such as the eastern and western Indian Ocean and the Bay of Bengal. Notably, chlorophyll concentration in the Arabian Sea decreased during the 1997 event but increased during the 2006 event**".

**References not in manuscript:**

Anjaneyan, P., J. Kuttippurath, P. V. Hareesh Kumar, S. M. Ali, and M. Raman (2023), Spatio-temporal changes of winter and spring phytoplankton blooms in Arabian sea during

the period 1997–2020, *Journal of Environmental Management*, *332*, 117435, doi:https://doi.org/10.1016/j.jenvman.2023.117435.

Bristow, L. A., et al. (2017), N2 production rates limited by nitrite availability in the Bay of Bengal oxygen minimum zone, *Nature Geoscience*, *10*(1), 24-29, doi:10.1038/ngeo2847.

Chakraborty, K., L. Rose, T. Bhattacharya, J. Ghosh, P. K. Ghoshal, and A. Akhand (2023), Primary Productivity Dynamics in the Northern Indian Ocean: An Ecosystem Modeling Perspective, in *Dynamics of Planktonic Primary Productivity in the Indian Ocean*, edited by S. C. Tripathy and A. Singh, pp. 169-190, Springer International Publishing, Cham, doi:10.1007/978-3-031-34467-1_8.

Ditkovsky, S., L. Resplandy, and J. Busecke (2023), Unique ocean circulation pathways reshape the Indian Ocean oxygen minimum zone with warming, Biogeosciences, 20(23), 4711-4736, doi:10.5194/bg-20-4711-2023.

Fox-Kemper, B., G. Danabasoglu, R. Ferrari, S. M. Griffies, R. W. Hallberg, M. M. Holland, M. E. Maltrud, S. Peacock, and B. L. Samuels (2011), Parameterization of mixed layer eddies. III: Implementation and impact in global ocean climate simulations, *Ocean Modelling*, *39*(1), 61-78, doi:https://doi.org/10.1016/j.ocemod.2010.09.002.

Gutknecht, E., G. Reffray, M. Gehlen, I. Triyulianti, D. Berlianty, and P. Gaspar (2016), Evaluation of an operational ocean model configuration at 1/12° spatial resolution for the Indonesian seas (NEMO2.3/INDO12) – Part 2: Biogeochemistry, *Geosci. Model Dev.*, *9*(4), 1523-1543, doi:10.5194/gmd-9-1523-2016.

Sreelakshmi, S., and P. K. Bhaskaran (2020), Wind-generated wave climate variability in the Indian Ocean using ERA-5 dataset, *Ocean engineering*, *209*, 107486.

Sunanda, N., J. Kuttippurath, R. Peter, K. Chakraborty, and A. Chakraborty (2021), Long-Term Trends and Impact of SARS-CoV-2 COVID-19 Lockdown on the Primary Productivity of the North Indian Ocean, *Front. Mar. Sci.*, *Volume 8 - 2021*, doi:10.3389/fmars.2021.669415.

Sunanda, N., J. Kuttippurath, A. Chakraborty, and R. Peter (2023), Stressors of primary productivity in the north Indian ocean revealed by satellite, reanalysis and CMIP6 data, *Progress in Oceanography*, *219*, 103164, doi:https://doi.org/10.1016/j.pocean.2023.103164.

Sunanda, N., J. Kuttippurath, R. Peter, and A. Chakraborty (2024), An atmosphere–ocean coupled model for simulating physical and biogeochemical state of north Indian Ocean: Customisation and validation, *Ocean Modelling*, *191*, 102419, doi:https://doi.org/10.1016/j.ocemod.2024.102419.

Sreelakshmi, S., and P. K. Bhaskaran (2020), Wind-generated wave climate variability in the Indian Ocean using ERA-5 dataset, *Ocean engineering*, *209*, 107486.

Udaya Bhaskar, T. V. S., V. V. S. S. Sarma, and J. Pavan Kumar (2021), Potential Mechanisms Responsible for Spatial Variability in Intensity and Thickness of Oxygen Minimum Zone in

the Bay of Bengal, *Journal of Geophysical Research: Biogeosciences*, *126*(6), e2021JG006341, doi:https://doi.org/10.1029/2021JG006341.

Vogt-Vincent, N. S., and H. L. Johnson (2023), Multidecadal and climatological surface current simulations for the southwestern Indian Ocean at 1⁄50° resolution, Geosci. Model Dev., 16(3), 1163-1178, doi:10.5194/gmd-16-1163-2023.

---

## Author Comment (AC1)

**Summary:**

We sincerely thank both anonymous reviewers for their thorough and constructive comments on our manuscript entitled "A high-resolution physical–biogeochemical model for marine resource applications in the Northern Indian Ocean (MOM6-COBALT-IND12 v1.0)". Their suggestions have significantly improved the clarity, completeness, and overall quality of the manuscript. We have included copies of their comments and our replies below.

**Referee #1**

**General comments:**

This article is discussing the development and validation of an ocean-bio-geochemistry model customized for the north Indian Ocean. The authors make a good effort to get the simulations done and for the validation, and is publishable. Modelling ocean bio-geochemistry is very difficult and many models still struggle to get the bio-geochemistry right in those simulations. However, I have some concerns, which need to be addressed before it can be accepted.

We sincerely thank the reviewer for the constructive and thoughtful comments, which helped us improve the quality and clarity of the manuscript. We have prepared revisions to address all of these comments. The main reviewer's concern is the bias in surface chlorophyll compared to satellite observations. In our response, we show that our chlorophyll biases are within the range of previously published models, and argue that our model biases in primary production – which is a more robust indicator of biological nutrient and carbon uptake than chlorophyll – are much lower than the chlorophyll biases. The other comments are requests for additional points of evaluation or clarification of the model configuration. We have prepared all additional analysis required to address these comments, and the model performs well on these evaluations. Specifically, we propose to:

- Include an assessment of wind forcing using ERA5 and validate it against CCMP winds and RAMA moorings, along with an expanded discussion and an additional figure (major comment #1).

- Expand the discussion of surface and subsurface chlorophyll biases, comparing model performance with other regional studies (e.g., Sunanda et al., Chakraborty et al.), and presenting depth-dependent validation using bio-Argo data (major comment #2, 3, 8, 9).

- Clarify the oxygen minimum zones evaluation, supported by volumetric and concentration-based diagnostics (major comment #4).

- Clarify the simulation setup, including spin-up duration and equilibration, boundary conditions, atmospheric forcing, as well as references and criteria for rescaling river

discharge and assigning riverine lithogenic concentrations (major comment #6; minor comment #9, 12, 14, 16).

- Clarify what we developed vs. what we customized in the model configuration, and add model performance metrics (major comment #6).

- Implement minor edits and clarifications as proposed to enhance precision and consistency throughout the text, as detailed in the point-by-point response (major comment #5, 7; minor comment #1-28).

Please see the detailed response below.

**Major comments:**

1. I do not see any wind simulation and its validation in the model. Since the monsoonal currents dictate the dynamics and associated processes in NIO, the wind simulations and their assessment are very important and must be presented in the main text.

Our model is a physical-biogeochemical coupled, ocean-forced model. The atmospheric forcing — including wind, precipitation, and solar radiation — is derived from the 1/4° horizontal resolution European Centre for Medium-Range Weather Forecasts (ECMWF) Reanalysis 5th Generation (ERA5) at 1-hour frequency (Hersbach et al., 2020). ERA5 is widely recognized as one of the highest-quality atmospheric reanalysis products. Numerous studies have employed ERA5 to drive ocean models in the Indian Ocean, capturing key features such as the monsoonal seasonality of ocean waves (Sreelakshmi and Bhaskaran, 2020) and ocean circulation (Vogt-Vincent and Johnson, 2023). See details in section 4.1 in Screelakshmi and Bhaskaran (2020) and sections 4.2-4.3 in Vogt-Vincent and Johnson (2023).

We evaluated ERA5 winds in the Indian Ocean by comparing them with the Cross-Calibrated Multi-Platform (CCMP) wind product, an observationally constrained dataset derived from satellite measurements, to ensure the robustness of the atmospheric forcing used in our model (new Figure R1). The comparison indicates that the ERA5 wind forcing reproduces the seasonal cycle and spatial distribution of summer and winter monsoons. To assess interannual variability, we further propose to compare ERA5 wind data with in situ observations from two RAMA mooring stations. The root-mean-square errors (RMSE) at these stations are 0.25 m/s and 1.39 m/s, respectively. These relatively small errors indicate that ERA5 forcing captures not only the seasonal dynamics but also the interannual variability of wind fields over the Indian Ocean.

In addition, Section 4 of the current manuscript assesses the seasonal dynamics of key oceanic and biogeochemical features, including sea surface temperature, upper ocean circulation, coastal upwelling and downwelling, sea surface salinity, river plumes, and plankton bloom. These features are well reproduced in comparison with available data products, providing further evidence that the ERA5 forcing captures the seasonal dynamics of the Indian monsoon system.

The details of atmospheric forcing setup was described in Lines 109-110 in the manuscript. We propose to add the wind evaluation using the new Figure R1 in the revised manuscript.

The proposed clarified text reads: "**In addition, a comparison between ERA5 and CCMP wind products demonstrates that ERA5 wind forcing effectively captures the seasonal cycle and spatial distribution of the summer and winter monsoons (Appendix Figure A6).**"

[Figure]

**Figure R1.** Surface wind (10m) during (a-c) winter (December-February) and (d-f) summer (June-August) monsoons. (a,d) CCMP satellite data, (b,e) ERA5 data product and (c,f) differences between ERA5 and CCMP. Correlation coefficients r, root-mean squared error (rmse) and bias between ERA5 and CCMP seasonal means are indicated. Wind data is from CCMP satellite (see details in Table 2). ERA5 results are averaged over the 1980-2020 period.

2. I thought, generally the models have relatively good simulations for surface Chl-a and compare well with the satellite and Argo data, which is not the case here in your model simulations. It is good to have a discussion based on other model simulations for NIO and other oceanic regions for Chl-a comparisons. Please see this and mention such model simulations and validation: https://doi.org/10.1016/j.ocemod.2024.102419

We emphasize in our manuscript that primary productivity is a better metric of biological production, nutrient and carbon uptake. In this regard, our model performs very well in

comparison with observations (Section 4.5 and Figure 10). The seasonal patterns of PP are consistent with satellite-derived estimates and in-situ observations (351 stations). The model captures the magnitude of the double bloom productivity in the central and western Arabian Sea (about 1000-1500 mg C m$^{-2}$ d$^{-1}$ in CAS and WAS), as well as the lower productivity observed in the Bay of Bengal (<1000 mg C m$^{-2}$ d$^{-1}$, Figure 10 a,b,e,f).

Regarding the chlorophyll bias in our and other models. We carefully reviewed other biogeochemical modeling studies in the northern Indian Ocean (e.g., Sunanda et al., 2024; Chakraborty et al. 2023; Gutknecht et al. 2016). The biases in our model fall within the range of these other studies. Chakraborty et al. (2023) reported a RMSE of ~0.7 mg m$^{-3}$ in the western and eastern Indian Ocean coast, similar to our model which yields a range from 0.62 (winter) to 0.93 (summer) mg m$^{-3}$. Gutknecht et al. (2016), using the NEMO–PISCES model, reported a domain-mean chlorophyll concentration of 0.53 mg m$^{-3}$ for the Indonesian Throughflow (ITF) region, compared with a MODIS-derived mean of 0.30 mg m$^{-3}$. Although our model covers a different domain (north Indian Ocean), its mean chlorophyll concentration is 0.44 mg m$^{-3}$, versus an OC-CCI observational mean of 0.34 mg m$^{-3}$.

As suggested, we also made a comparison to Argo Chlorophyll showing  depth-dependent statistical metrics, including RMSE, bias, and correlation coefficient  (see new Figure R2 below). The RMSE between our model and Argo ranges from 0.03 to 0.37 mg m$^{-3}$ ni upper ocean in the Arabian Sea and the Bay of Bengal, which is lower than the RMSE between our model and satellite products listed above, but higher than the RMSE reported for the model of Sunanda et al. 2024 (0–0.1 mg m$^{-3}$).

The potential sources of the chlorophyll bias were discussed in section 4.5 of our manuscript. Notably, large phytoplankton are characterized by a higher chlorophyll-to-carbon ratio, whereas small phytoplankton have a lower ratio. Given that the model's carbon-based estimates (primary production) align well with the observational data, we propose that the overestimation of chlorophyll arises from an overrepresentation of large phytoplankton, which contribute disproportionately to the chlorophyll signal. However, the strong agreement of our model with observations of PP gives us confidence in the model's representation of biogeochemical dynamics despite moderate biases in chlorophyll.

We propose to expand the chlorophyll assessment and discussion by including Figure R2 in the revised manuscript.

- The proposed revised text in Line 362 reads: "The model also simulates  the subsurface chlorophyll maximum captured by Argo floats in both the Arabian Sea and Bay of Bengal, with RMSE values over the vertical ranging from 0.03 to 0.37 mg m−3 (Appendix Figure A5). This suggests that the model effectively represents the vertical distribution of plankton and associated subsurface biological dynamics. Overall, comparison of our model's mean bias and RMSE with values reported in previous studies suggests that our chlorophyll simulation performance falls within the median range

relative to other regional biogeochemical models (Chakraborty et al., 2023; Gutknecht et al., 2016; Sunanda et al., 2024)".

- The proposed revised text in Lines 370 reads (new in bolded): "The fact that the model simulates the magnitude of observed PP (in carbon units) but overestimates the surface chlorophyll content suggests that it might overestimate the contribution of large phytoplankton, **which is characterized by a high chlorophyll-to-carbon ratio,** compared to small phytoplankton, **characterized by a lower chlorophyll-to-carbon ratio. This overestimation of the contribution of large phytoplankton to the assemblage would indeed explain the good match in primary productivity and high bias in chlorophyll**".

[Figure]

**Figure R2**. Comparison of observed and modeled vertical chlorophyll profiles in the Arabian Sea (AS, top row) and the Bay of Bengal (BoB, bottom row) using Argo float observations and model output. (a, d) Argo-derived chlorophyll concentrations; (b, e) model-simulated chlorophyll concentrations; (c, f) depth profiles of root mean square error (RMSE), bias, and correlation coefficient between model and Argo observations. In panels a-b and d-e, the black contour line indicates the depth of the subsurface chlorophyll maximum (SCM). Chlorophyll concentrations are shown in mg m⁻³.

3. There is a subsurface maximum for Chl-a in the NIO. Please show the model simulations and comparison with Argo measurements.

Thank you for the suggestion to compare the model to Argo. As detailed in comment #2, we propose to evaluate the modeled subsurface chlorophyll structure using Argo float observations in the revised manuscript (Figure R2). See the proposed revised text and figure in major comment #2.

4. L459: If the model overestimates, how that would affect the OMZ in AS and BoB? Which oceanic region has large OMZs? L463, you say that in AS, it is well represented, but BoB overestimated. Fig 16 provides no clue that any basin is better for this. Also, why the equatorial region has such a big difference?

In observation-based products (WOA18 and Bianchi et al., 2012), the AS features an intense OMZ with subsurface oxygen concentrations around 5 µmol kg⁻¹, whereas the BoB exhibits a much less intense OMZ, with subsurface oxygen concentrations typically between 10–20 µmol kg⁻¹. Figure 16 of our manuscript does compare the OMZ in the AS and BoB in products and the model. Panel a shows maps of O2 in model, products and the model bias. Panel b compares the OMZ volume in products and model for the full basin, the AS and the BoB for different oxygen thresholds (e.g., suboxia is defined by 5 umol/kg but other thresholds such as hypoxia are relevant for organisms etc.).

As described in section 5.2, the oxygen bias is higher in the BoB (approximately –10 to –20 µmol kg⁻¹) than in the AS (approximately –10 to +10 µmol kg⁻¹, Figure 16a). Additionally, we find that at the basin scale the model reproduces the volume of hypoxic (<60 µmol kg⁻¹) and low-oxygen (<100 µmol kg⁻¹) waters (Figure 16b). However, it overestimates suboxic waters (<5 µmol kg⁻¹), mainly due to the overprediction in the Bay of Bengal. This is visible in the lower right panel of Figure 16 when comparing the products (blue and red lines) to the model (black lines) at oxygen values of 5 umol/kg. We propose to clarify the description of Figure 16 in section 5.2 as follows: "In contrast, the model overestimates the volume of suboxic waters delimited by 5 umol kg-1 (0.17×1016 m3 vs. 0.06×1016 m3 in Bianchi et al. (2012) observations), mostly because of the large suboxic volume simulated in the Bay of Bengal (0.10×1016 m3 vs. 0.00×1016 m3 in observations, Figure 16b)".

We also propose to expand the results to include the western equatorial bias in section 5.2. It reads as: "In this region, the model shows a high oxygen bias near the base of the thermocline (500–1000 m), coinciding with a low nitrate bias (not shown). This pattern points to either a misrepresentation of biological remineralization at depth or an inaccurate representation of the relative contribution of the water masses forming the Central Waters supplying oxygen to this region. These waters originate from a blend of the Indonesian Throughflow (ITF) and southern-sourced Mode Waters. Previous studies have demonstrated that oxygen distribution in this region is highly sensitive to the relative contribution between these two sources (DItkovsky et al 2023). However, the scarcity of direct observations in this region limits our ability to conclusively attribute the model bias to either mechanism."

However, recent studies indicate an uncertainty regarding the true extent and severity of hypoxia in this region (Bhaskar et al., 2021, Bristow et al., 2017). We therefore also propose to expand the discussion of the model oxygen bias in section 8 (Discussion and Conclusions) by adding the following text: "Despite these model limitations, it is important to note that uncertainties remain regarding the strength of suboxia in the Bay of Bengal. Recent observations from Argo floats and ship-based in situ measurements have reported lower oxygen concentrations in the BoB than

those presented in the WOA dataset, including nanomolar-level oxygen conditions (Bristow et al., 2017; Bhaskar et al., 2021). These findings suggest that the true extent and intensity of hypoxia in the BoB remain uncertain, making it difficult to definitively assess the magnitude of the model bias in this region."

5. Write the validation information with bias values in the abstract

We propose to revise the abstract (Lines 12-13) as follows: "**Quantitatively, the model exhibits relatively small biases, as reflected by root mean square error (RMSE) values in key variables: surface temperature (0.25–0.30 °C), mixed layer depth (7–8.09 m), sea level anomaly (0.02 m), sea surface salinity (0.53–0.71 psu), vertical chlorophyll (0.03–0.3 mg m⁻³), subsurface temperature (0.33 °C), and subsurface salinity (0.07 psu)**".

6. Developed or customized the model, please make sure that you use a correct word for this

The global MOM6-COBALTv2 model was originally developed by GFDL scientists including coauthors in this paper. However, regional capabilities are new. We have developed the regional Indian Ocean configuration and customized the MOM6-COBALTv2 model by modifying several parameters (e.g., detritus sinking velocity, burial fraction, oxygen half-saturation for nitrification, oxygen constraint on water column denitrification), as well as rewriting portions of the input and output Fortran routines within regional MOM6-COBALTv2 to better suit the specific needs of our regional configuration and biogeochemical applications. This is not merely a simple customization; rather, we are deeply engaged in developing and refining the model to address bugs and enhance its functionality as a regional biogeochemical modeling tool.

- The proposed revised text in Lines 63-66 reads: "It is with these applications in mind that we **configured, customised and validated** the regional Indian Ocean simulation presented here based on the Modular Ocean Model 6 (MOM6, Adcroft et al., 2019) coupled with the Carbon, Ocean, Biogeochemistry, and Lower Trophics module version 2.0 (COBALTv2, Stock et al., 2014, 2020)."
- The proposed revised text in Line 569 reads: "We **configured, customised and validated** a regional ocean biogeochemical model at 1/12° horizontal resolution (MOM6-COBALT-IND12 v1.0) that captures most key features of the northern Indian Ocean dynamics."
- The proposed revised text in Line 589 reads: "During the **setup and customization** of the MOM6-COBALT-IND12 v1.0 model, we identified a series of physical and biogeochemical parameters and forcings that influenced the model simulation and led to a significant improvement of the results (see details in section 2)."

7. L336: reproduced is a "lighter" word; how good is the comparison? Please write some numbers.

We propose to revise as suggested. The comparison is already quantified in the text, with a regional correlation coefficient (r > 0.95) and RMSE < 0.7, as also shown in Figure 8. We propose to revise the text to present these metrics more clearly: "Performance metrics indicate

that the simulation achieves **a strong spatial correlation (r > 0.95) and a small regional RMSE (0.53-0.7)**”.

8. LK353: Why the Chl-a simulation is not good in the Somali coast or western Indian Ocean? Summer Chl-a is even worse?

9. L371-372: How did you arrive into this conclusion; is this about the size of the plankton?

These two comments are related to Comments #2 and #3 and we answer them together here. The potential sources of chlorophyll bias are discussed in detail in our responses to comments #2 and #3 and were also discussed in Section 4.5 of the manuscript. Large phytoplankton has a higher chlorophyll-to-carbon ratio and the model likely overestimates the contribution of this group to the assemblage. Again we want to emphasize that the model simulates primary production comparable with observations (Figure 10), which is a more important metric than chlorophyll for constraining nutrient, carbon and oxygen cycling.

The larger bias in the Somali Current, and summer specifically, is likely related to enhanced upwelling during this season, which brings nutrients to the surface. In addition, the strengthened coastal current driven by the southwest monsoon may further transport nutrients into the western Indian Ocean. This elevated nutrient availability facilitates larger phytoplankton growth and higher Chl content in the model. See the proposed revised text and figure in major comment #2.

**Minor comments:**

1. L4: north of 8S? It can be anywhere north of that latitude. Please be specific

We propose to revise as suggested: “The model covers the northern Indian Ocean (**from 8°S to the northern continental boundaries**), central to the livelihoods and economies of countries that comprise about one-third of the world’s population.”

2. L22: and is missing

We propose to revise as suggested: “The northern Indian Ocean is central to the livelihood and economy of about one third of the Earth’s population which live in its littoral countries (e.g., India, Indonesia, Pakistan, Bangladesh, Tanzania, Myanmar, Malaysia, Kenya**, and** Yemen) ”.

3. L23; separate the Roy citation from the bracket

We propose to revise as suggested: “... provides valuable resources via the “blue economy”, such as fishery, aquaculture, **marine tourism (Roy, 2019)**”.

4. L40: about the NIO stressors: https://doi.org/10.1016/j.pocean.2023.103164

We propose to add further discussion on stressors affecting the northern Indian Ocean under climate change, and cite two relevant studies: Sunanda and Chakraborty (2023) and Sunanda et al. (2021).

The revised text in Line 45 will read: "Projections from Coupled Model Intercomparison Project (CMIP) models suggest substantial shifts in net primary production (NPP) and sharp declines in pH in the coming decades, **highlighting the North Indian Ocean's particular vulnerability to climate change (Sunanda et al., 2021; Sunanda and Chakraborty, 2023)**."

5. L53: models are "tools" for studying

We propose to revise as suggested: "Models are powerful **tools** for exploring the Indian Ocean".

6. L55: this is another model validation for this region: https://doi.org/10.1016/j.ocemod.2024.102419

We propose to revise as suggested and cite one more model work in the revised manuscript. The proposed revised text will read: "... assess the impacts on biogeochemistry and ecosystems (e.g., Sengupta et al., 2001; Rahaman et al.,2014; Lachkar et al., 2018, 2019; Resplandy et 55 al., 2011, 2012; Schmidt et al., 2021; Ditkovsky et al., 2023; **Sunanda et al., 2024**)."

7. L85: coordinates of the region

We propose to revise as suggested: "MOM6-COBALT-IND12 is therefore considered an 'eddy resolving' model for the region **with a rectilinear and orthogonal grid ($32°E$ to $114°E$ and $8.6°S$ to $30.3°N$)**".

8. L113: salinity from 1998 data, any updated version?

The salinity restoring data used in this study is PHC2.1, released in 2002, which remains the version currently used by the Geophysical Fluid Dynamics Laboratory (GFDL), NOAA. A newer version, PHC3.0, became available in 2005. We plan to update the salinity restoring data to PHC3.0 or a similar newer product, such as WOA 2023, in future simulations.

9. L115: How long was the spin up and when did the model stabilize? Which year onward you analyse the model results for science?

We propose to clarify the spin-up stabilization and hindcast simulation by rewriting section 2.2.1 of the revised manuscript (modified text in bold):

"**The ocean model was initialized using temperature and salinity from annual mean fields from the World Ocean Atlas version 2013 (WOA13, Locarnini et al., 2014; Zweng et al., 2014). Our simulations were run using the atmospheric forcing from the $1/4°$ horizontal resolution European Center for Medium-range Weather Forecasts reanalysis 5th generation (ERA5) at 1-hour frequency (Hersbach et al., 2020).** The sea surface salinity (SSS) was restored to the polar science center hydrographic climatology (PHC2.1), which is based on the World Ocean Atlas 98 with data replenishment in the Arctic Ocean (Steele et al., 2001), with a piston velocity of 0.1667 m d−1. **We conducted a 32-year spin-up, which was achieved by looping four consecutive 8-year loops of the 1980 to 1987 forcing field and reached a well-equilibrated state with minimal linear trends of physical and biogeochemical variables (e.g., drift in sea surface temperature, SSS, oxygen, nitrate,**

**primary production and ocean surface partial pressure of carbon dioxide $pCO_2$ < ~0.1% for model years 17-32). Using outputs from the end of the spinup simulation as initial conditions, the hindcast simulation was started on January 1 1980 and was run from 1980 to 2020 for our analysis in this study.**"

The minimal model drift in sea surface temperature, SSS, oxygen, nitrate, primary production and $pCO_2$ for model years 17-32 of the spin-up is shown in Figure R3 below for your reference.

[Figure]

**Figure R3**. Time series of domain-mean model variables during the 32-year spin-up forced by repeated atmospheric and open boundary conditions looping over 1980–1987. Panels show: (a)

sea surface temperature (SST), (b) sea surface salinity (SSS), (c) oxygen ($O_2$), (d) total nitrate ($NO_3^-$), (e) primary productivity (PP), and (f) ocean $pCO_2$. Drifts over model years 17-32 are indicated above each panel and are all $< \sim0.1\%$

10. L119: citation format is not correct

We will correct the format. The proposed revised text reads: "Open boundary conditions (OBC) are set using the Flather formulation for the tidal and sub-tidal sea level and barotropic velocity and the Orlanski formulation for the baroclinic velocity (**Flather, 1976**; Orlanski, 1976)".

11. Figure 1: rivers can be in red color, to differentiate from the bathymetry blue color

We propose to revise as suggested. The updated figure is shown below.

[Figure]

**Figure R4**. Domain and bathymetry of the regional Indian Ocean MOM6-COBALT-IND12. Pink shading indicates the extent of sponge layers (see methods). Major rivers are indicated in red.

12. L147: any reference for this? Overestimation and scaling have got any criterion? Why 25%?

The overestimation and subsequent scaling of river discharge are based on our evaluation of the GloFAS dataset, as shown in Appendix Figure A4 in the current manuscript and discussed in section 2.2.3. We compared the river discharge from GloFAS with observational estimates for the Ganges–Brahmaputra river system reported by Jian et al. (2009). To address this bias and improve the realism of the freshwater input, we applied a 25% reduction to the GloFAS-derived discharge for the Ganges–Brahmaputra river system (see Figure A4). We also corrected the Irrawaddy-Sittang river system using a similar approach and data from the Global Runoff Data Center (Recknagel et al., 2023). These references are already cited in the original manuscript (Line 145 and Table 2)  but we will also add them in Line 147  to improve clarity in Section 2.2.3.

The proposed revised text reads: "By comparing GloFAS to published discharge observations (Jian et al., 2009; Siswanto et al., 2023), we found that GloFAS overestimated discharge in the Ganges-Brahmaputra river system, and therefore scaled down the freshwater discharge by 25% to match observations in these two rivers (see Appendix Figure A4, Jian et al., 2009, Siswanto et al., 2023)".

13. L175: not from WOA 2023?

The nutrient and oxygen fields were initialized using data from the World Ocean Atlas (WOA) 2018, as the project began before the release of WOA 2023. We plan to incorporate the updated WOA 2023 fields in future model simulations.

14. L176: CO2 is increasing, so the old climatology values are good?

The initial DIC field used for the model spin-up is derived from the GLODAP climatology, which reflects approximately the year 2002 conditions, higher indeed than the 1980 conditions of our initial state. However, the 32-year spin-up with repeated forcing fields from 1980–1987 (air $CO_2$ levels and DIC concentration at open boundary conditions are also from 1980-1987) yielded a stabilized surface ocean $pCO_2$ lower than its initial values, toward the end of the spin-up period with a linear trend of ~0.05% (see Figure R3 above), indicating minimal drift. We propose to clarify this point in section 2.4.1 of the revised manuscript when describing the initialization of the model spin-up and model drift (added text in bold):

"**For the model spin-up**, nutrients (nitrate, phosphate, and silicate) and oxygen were initialized using annual means from the World Ocean Atlas 2018 (WOA18, Garcia et al., 2019). DIC and alkalinity were initialized using annual means from GLODAPv2 **which are representative of year 2002** (Olsen et al., 2016). … Model drift after the 32-year spin-up and over the 41 years of a control simulation with constant forcing is small, with linear trends < 0.05% for oxygen, nitrate, DIC, alkalinity, semi-refractory dissolved organic nitrogen pools and integrated primary productivity (see Appendix Figure A1). **The slight drift indicates that the hindcast simulation starts from a well-equilibrated initial state provided by the spin-up simulation**."

15. L192: SSP 5-8.5 is an extreme case. So how much that would influence your simulations?

The atmospheric deposition data were taken from the GFDL ESM4.1 historical simulation for the period 1980–2014 and from the SSP5-8.5 scenario of the same model for the period 2015–2020. We selected the SSP5-8.5 pathway because it is regarded as a "business-as-usual" scenario, and our current society did not largely reduce carbon emission in the past few years (i.e., 2015-2020). Therefore, it is reasonable to still consider the past few years as the "business-as-usual" scenario. This is the reason we select the SSP5-8.5 data for the period 2015-2020.

In addition, we assess the consistency between the historical and scenario-based datasets. As shown in Figure A2 in the current manuscript, there is no noticeable discontinuity between the two periods in either magnitude or trend of nitrate deposition. Therefore, the use of SSP5-8.5 for

the 2015–2020 period does not introduce artificial discontinuities and is not expected to significantly influence the model simulation results.

16. L223: How the adjustments are made? Just random or any criterion followed?

The adjustments to riverine lithogenic concentrations are described in the manuscript (Lines 226-228) and the supporting observational dataset (Milliman and Farnsworth, 2011) is listed in Table 2 under the "Riverine lithogenic flux" entry. We will clarify this point in the revised manuscript at the beginning of the lithogenic input description by adding the following sentence (**added text in bold**): "**To reflect spatial differences in sediment supply, we specify riverine lithogenic concentrations based on observational data from Milliman and Farnsworth (2011):** the lithogenic input from rivers was adjusted to 200 g m$^{-3}$ for major rivers (i.e., rivers with sediment loads exceeding 10 Mt y$^{-1}$, e.g., Godavari, Krishna, Ganges, Brahmaputra, Irrawaddy, Sittang, Salween, Indus, Tapti and Narmada rivers, see Figure 1 for rivers location) and 20 g m$^{-3}$ for all other rivers, rather than applying a global constant of 13 g m$^{-3}$ used for all rivers in Stock et al. (2020). These adjustments account for the significantly higher total suspended sediment loads in these rivers (Milliman and Farnsworth, 2011; Rixen et al., 2019b), and are supported by river observations from Milliman and Farnsworth (2011) showing a broad range from 10 g m$^{-3}$ (Muvattupuzha River) to 1,061 g m$^{-3}$ (Ganges River)".

17. L243: citation format is not correct

We propose to edit as suggested: "temperature and salinity from the World Ocean Atlas 2018 (WOA18, **Garcia et al., 2019**)."

18. L274: SST has been already defined

We propose to edit as suggested: "Patterns of **SST** in the northern Indian Ocean follow the well described basin-scale features."

19. L276: particularly and especially, Please rephrase

We propose to edit as suggested: "MOM6-COBALT-IND12 captures the seasonal SST patterns **well, notably** the contrast between the vast warm pool (SST >28°C) covering most of the basin and the colder SST regions that emerge in response to seasonal variations in atmospheric and oceanic circulation (Figure 3)."

20. Figure 4: Why summer MLD is bad in the model?

We propose to expand the evaluation of winter and summer MLD in section 4.1 by adding the following paragraph: "The model captures the seasonal contrast in mixed layer depth (MLD) between the Arabian Sea and the Bay of Bengal, with deeper mixed layers in the Arabian Sea and shallower layers in the Bay of Bengal during both winter and summer(Figure 4). The MLD is generally deeper in summer than in winter. The spatial patterns, including the locations of local MLD maxima, are broadly consistent with observational data. Quantitatively, the basin-wide correlation values are similar between the two seasons, although the RMSE is larger in summer (8.09 m) than in winter (7.00 m). One possible contributor to the larger summer bias

is the enhanced wind forcing during the monsoon season (see Figure A6), which intensifies turbulent mixing and deepens the mixed layer. At the same time, the MOM6 model includes the mixed layer eddy (MLE) parameterization of Fox-Kemper et al. (2011), which represents restratification driven by baroclinic eddies within the mixed layer. This restratification process may also be more active in summer, potentially leading to an overcorrection that offsets vertical mixing too strongly. The interaction between intensified wind-driven mixing and enhanced restratification may thus contribute to the larger MLD bias observed in summer compared to winter."

21. L337: SSS has been defined already

We will remove and edit as suggested: "The model reproduces the main observed patterns of SSS, including the high **SSS** (>34) in the Arabian Sea".

22. L350: Narmada-Tapti

We will correct as: "The GloFAS runoff product captures the discharge of one of the main river systems for which we have direct observations, i.e., the **Narmada-Tapti** rivers."

23. Fig 17: How that affects simulations of SLA?

As shown in Figure 17, the model underestimates the amplitude of SSH variability in the 14–120 day/cycle range compared to AVISO observations. This frequency band corresponds to intraseasonal, including Kelvin and Rossby waves related to high-frequency atmospheric forcing responses and other ocean processes. As a result, the magnitude of sea level anomalies associated with these waves may be underestimated, and their influence on the biogeochemical system could be underrepresented in the simulations.

Nevertheless, the model still performs well in reproducing most of the key variabilities in both physical and biogeochemical processes. As demonstrated in Section 4.3 and Figure 7, the model captures the seasonal cycle of sea level anomalies with high fidelity, yielding a strong correlation (r = 0.93) and low RMSE (0.02 m) relative to observations. Therefore, while the model underestimates SLA variability in the high-frequency band, this has only a minor impact on the overall accuracy of SLA simulation, which is largely governed by lower-frequency components.

24. L495-500: remarkably well? Not sure, if you look at the SLA figure.

We propose to rephrase the text to ensure consistency between the descriptions of the SLA and chlorophyll simulations. Specifically, we propose to rephrase the sentence as the model "reproduces the fine-scale features structuring the winter and summer blooms," rather than describing the performance as "remarkably well". The revised text will read: "The model **reproduces the fine-scale features** structuring the winter and summer blooms".

25. L514: IOD has been defined, as for L528: RAMA, OISST

We propose to remove and edit as suggested.

- L514, the revised text will read: "The model reproduces the amplitude and zonal pattern of SST changes expected in response to the Interannual **IOD**". Line 530 in the revised manuscript.

- L528, the revised text will read: "We use observations from the in-situ **RAMA** that we complement with **OISST** data and Argo float-based thermocline depth." Line 543 in the revised manuscript.

26. 579-580: model is good because of its good bio-geochemistry simulations? What about the model physics?

The text here intends to emphasize that the model's strong physical performance provides the foundation for accurately simulating ocean biogeochemical and biological responses. The physical component of the model has been thoroughly evaluated, as detailed in Sections 2–7, which assess circulation variability, meridional temperature and salinity profiles, seasonal and intraseasonal sea level variability, and interannual SST variability. The well physical model performance supports its capacity to accurately represent the biogeochemical and biological variabilities.

We propose to rephrase the text for clarity: "The good agreement between observed and modeled physical features **provides a foundation for accurately simulating** the ocean biogeochemical and biological response".

27. L584: a detailed account of winter blooms are here: https://doi.org/10.1016/j.jenvman.2023.117435

We propose to revise as suggested. We plan to cite the suggested winter bloom work in the revised manuscript. The proposed revised text will read: "Specifically, the model reproduces the summer bloom associated with coastal upwelling systems and their extension offshore in mesoscale filaments, as well as the winter bloom associated with convective mixing and modulated by fine-scale eddies (Lévy et al., 2007; Resplandy et al., 2011, 2012; Mahadevan, 2016; Lachkar et al., 2016; Rixen et al., 2019a; Vinayachandran et al., 2021; **Anjaneyan et al., 2023**)".

28. L588: different response? please be specific

We propose to specify the detailed responses reported in Wiggert et al. (2009). The proposed revised text will read: "This includes the modulation of the production in the equatorial region, the Arabian Sea and around the tip of India, although we note these patterns are difficult to generalize to all IOD events. **As illustrated by Wiggert et al. (2009) who showed the chlorophyll responses vary with IOD intensity in most regions, such as the eastern and western Indian Ocean and the Bay of Bengal. Notably, chlorophyll concentration in the Arabian Sea decreased during the 1997 event but increased during the 2006 event**".

**References not in manuscript:**

Anjaneyan, P., J. Kuttippurath, P. V. Hareesh Kumar, S. M. Ali, and M. Raman (2023), Spatio-temporal changes of winter and spring phytoplankton blooms in Arabian sea during the period 1997–2020, *Journal of Environmental Management*, *332*, 117435, doi:https://doi.org/10.1016/j.jenvman.2023.117435.

Bristow, L. A., et al. (2017), N2 production rates limited by nitrite availability in the Bay of Bengal oxygen minimum zone, *Nature Geoscience*, *10*(1), 24-29, doi:10.1038/ngeo2847.

Chakraborty, K., L. Rose, T. Bhattacharya, J. Ghosh, P. K. Ghoshal, and A. Akhand (2023), Primary Productivity Dynamics in the Northern Indian Ocean: An Ecosystem Modeling Perspective, in *Dynamics of Planktonic Primary Productivity in the Indian Ocean*, edited by S. C. Tripathy and A. Singh, pp. 169-190, Springer International Publishing, Cham, doi:10.1007/978-3-031-34467-1_8.

Ditkovsky, S., L. Resplandy, and J. Busecke (2023), Unique ocean circulation pathways reshape the Indian Ocean oxygen minimum zone with warming, Biogeosciences, 20(23), 4711-4736, doi:10.5194/bg-20-4711-2023.

Fox-Kemper, B., G. Danabasoglu, R. Ferrari, S. M. Griffies, R. W. Hallberg, M. M. Holland, M. E. Maltrud, S. Peacock, and B. L. Samuels (2011), Parameterization of mixed layer eddies. III: Implementation and impact in global ocean climate simulations, *Ocean Modelling*, *39*(1), 61-78, doi:https://doi.org/10.1016/j.ocemod.2010.09.002.

Gutknecht, E., G. Reffray, M. Gehlen, I. Triyulianti, D. Berlianty, and P. Gaspar (2016), Evaluation of an operational ocean model configuration at 1/12° spatial resolution for the Indonesian seas (NEMO2.3/INDO12) – Part 2: Biogeochemistry, *Geosci. Model Dev.*, *9*(4), 1523-1543, doi:10.5194/gmd-9-1523-2016.

Sreelakshmi, S., and P. K. Bhaskaran (2020), Wind-generated wave climate variability in the Indian Ocean using ERA-5 dataset, *Ocean engineering*, *209*, 107486.

Sunanda, N., J. Kuttippurath, R. Peter, K. Chakraborty, and A. Chakraborty (2021), Long-Term Trends and Impact of SARS-CoV-2 COVID-19 Lockdown on the Primary Productivity of the North Indian Ocean, *Front. Mar. Sci.*, *Volume 8 - 2021*, doi:10.3389/fmars.2021.669415.

Sunanda, N., J. Kuttippurath, A. Chakraborty, and R. Peter (2023), Stressors of primary productivity in the north Indian ocean revealed by satellite, reanalysis and CMIP6 data, *Progress in Oceanography*, *219*, 103164, doi:https://doi.org/10.1016/j.pocean.2023.103164.

Sunanda, N., J. Kuttippurath, R. Peter, and A. Chakraborty (2024), An atmosphere–ocean coupled model for simulating physical and biogeochemical state of north Indian Ocean: Customisation and validation, *Ocean Modelling*, *191*, 102419, doi:https://doi.org/10.1016/j.ocemod.2024.102419.

Sreelakshmi, S., and P. K. Bhaskaran (2020), Wind-generated wave climate variability in the Indian Ocean using ERA-5 dataset, *Ocean engineering*, *209*, 107486.

Udaya Bhaskar, T. V. S., V. V. S. S. Sarma, and J. Pavan Kumar (2021), Potential Mechanisms Responsible for Spatial Variability in Intensity and Thickness of Oxygen Minimum Zone in the Bay of Bengal, *Journal of Geophysical Research: Biogeosciences*, *126*(6), e2021JG006341, doi:https://doi.org/10.1029/2021JG006341.

Vogt-Vincent, N. S., and H. L. Johnson (2023), Multidecadal and climatological surface current simulations for the southwestern Indian Ocean at 1⁄50° resolution, Geosci. Model Dev., 16(3), 1163-1178, doi:10.5194/gmd-16-1163-2023.

**Referee #2**

**General comments:**

This manuscript presents a coupled ocean dynamics-biogeochemistry model whose domain covers the Indian Ocean north of 8°S with a grid resolution of 1/12°. The model's performance is evaluated through comparisons between model results for the years 1980–2020 and observations.

We thank the reviewer for the feedback. We have prepared revisions to address all of the comments. The reviewer's primary concerns were related to the scope of the manuscript, the model's performance and comparisons of our model with previous regional and global models. In response to these concerns and other detailed comments, we have strengthened the manuscript by making the following revisions:

- Expand the discussion section to more clearly articulate the features and advancements of our model development (major comment #1).

- Provide enhanced explanations for both limitations and strengths of the model simulation compared to observations. We focus particularly on mixed-layer depth (MLD), chlorophyll, sea surface height (SSH), sea surface temperature (SST), and oxygen concentrations (major comment #2 and #5).

- Include a comparison of our regional model with previous regional and global modeling efforts, supplemented by an additional figure to clearly illustrate the improved performance of our model (major comment #3 and #4).

- Revise minor edits and clarifications (major comment #6; minor comment #1-24).

Please see our detailed responses to each comment below.

**Major comments:**

We have separated the general comment into six specific comments and addressed them individually.

1. This manuscript presents a coupled ocean dynamics-biogeochemistry model whose domain covers the Indian Ocean north of 8°S with a grid resolution of 1/12°. The model's performance is evaluated through comparisons between model results for the years 1980–2020 and observations. While it is clear that a considerable amount of work has gone into the development and validation of the model, I feel the manuscript reads more like a report than a journal article and is thus not yet ready for publication. I believe the discussion needs to be more developed on: what is new about this work.

We would like to clarify that our manuscript is a model description paper, intended to introduce and evaluate a newly configured and regionally adapted ocean biogeochemistry model for the northern Indian Ocean. Our manuscript fully aligns with the scope of Geoscientific Model Development (GMD), which includes model descriptions and comprehensive evaluation. See for instance recent publications in the same GMD special issue: Ross et al. (2023, GMD) and Drenkard et al. (2024, GMD). This model will serve as a baseline model for future studies to investigate regional biogeochemical processes, their variability, and long-term trends in the northern Indian Ocean.

Nevertheless, we agree with the reviewer's comment that the manuscript could be improved by placing greater emphasis on these novel aspects. We discuss the novelty of our work in the Discussion section (Lines 589–632) of original manuscript and plan to expand this discussion by replacing Lines 569–570 as follows: "**In this study, we developed, customized, and validated a high-resolution (1/12°) regional ocean biogeochemical model (MOM6-COBALT-IND12 v1.0) for the northern Indian Ocean. Specifically, we adjusted river discharge rates and nutrient loadings in the Bay of Bengal according to observational constraints, significantly improving simulations of river plume dynamics and surface salinity. Additionally, we enhanced lithogenic particle fluxes from rivers, adjusted detritus remineralization rate, and refined the parameterization of the nitrogen cycle, resulting in better representations of subsurface oxygen distributions and suboxic conditions. These improvements collectively allow the model to capture most key aspects of biogeochemical and physical processes in the northern Indian Ocean**".

2. What are the possible causes for the model not performing as well in some aspects as in others.

We have carefully considered the potential reasons underlying the model's varying performance across different aspects, and these have been discussed in detail in the discussion section (Lines 579–645) of the original manuscript. In the revised manuscript, we propose to add more discussion on the factors controlling model performance across different aspects, including mixed-layer depth (MLD), chlorophyll, sea surface height (SSH), sea surface temperature (SST), and oxygen. Notably, the expanded discussion on MLD, chlorophyll, and oxygen are in line with comments from the other reviewer. The revised discussion is provided below:

- Mixed layer depth (we plan to add it in Line 288): "**One possible contributor to the larger summer bias is the enhanced wind forcing during the monsoon season (see Figure A6), which intensifies turbulent mixing and deepens the mixed layer. At the**

**same time, the MOM6 model includes the mixed layer eddy (MLE) parameterization of Fox-Kemper et al. (2011), which represents restratification driven by baroclinic eddies within the mixed layer. This restratification process may also be more active in summer, potentially leading to an overcorrection that offsets vertical mixing too strongly. The interaction between intensified wind-driven mixing and enhanced restratification may thus contribute to the larger MLD bias observed in summer compared to winter**"

- Chlorophyll (we plan to add it in Line 370): "The fact that the model simulates the magnitude of observed PP (in carbon units) but overestimates the surface chlorophyll content suggests that it might overestimate the contribution of large phytoplankton, **which is characterized by a high chlorophyll-to-carbon ratio**, compared to small phytoplankton, **characterized by a lower chlorophyll-to-carbon ratio. This overestimation of the contribution of large phytoplankton to the assemblage would indeed explain the good match in primary productivity and high bias in chlorophyll**".

- SSH (we plan to add it in Line 484): "**These underestimations might indicate the current spatial resolution of the model (1/12°) may still be insufficient for resolving these processes.**"

- SST (we plan to insert it in Lines 578): "**This strong physical performance likely stems from the effective parameterizations of surface heat and momentum fluxes, combined with well-constrained surface forcing fields derived from ERA5 reanalysis**".

- Biogeochemical variables (we plan to add it in Line 585). "**These biogeochemical improvements reflect the targeted model development efforts outlined above.**"

- Oxygen (we plan to add it in Line 632): "**Despite these model limitations, it is important to note that uncertainties remain regarding the strength of suboxia in the Bay of Bengal. Recent observations from Argo floats and ship-based in situ measurements have reported lower oxygen concentrations in the BoB than those presented in the WOA dataset, including nanomolar-level oxygen conditions (Bristow et al., 2017; Udaya Bhaskar et al., 2021). These findings suggest that the true extent and intensity of hypoxia in the BoB remain uncertain, making it difficult to definitively assess the magnitude of the model bias in this region.**"

3. The authors do discuss the ways in which the current version of their model performs better than an earlier version, but I am curious about how this model is different, in terms of setup and/or performance, to other physical-biogeochemical regional models of the northern Indian Ocean, such as that of Sunanda et al. (2024).

We agree with the reviewer's comment on the importance of inter-model comparisons. We have added additional model comparisons in the revised manuscript. This point was also raised in

general comment #2 by the other reviewer. We propose to add the following comparison to prior work in Line 362: "**The model also simulates the subsurface chlorophyll maximum captured by Argo floats in both the Arabian Sea and Bay of Bengal, with RMSE values over the vertical ranging from 0.03 to 0.37 mg m$^{-3}$ (Figure A5). This suggests that the model effectively represents the vertical distribution of plankton and associated subsurface biological dynamics. Overall, comparison of our model's mean bias and RMSE with values reported in previous studies suggests that our chlorophyll simulation performance falls within the median range relative to other regional biogeochemical models (Chakraborty et al., 2023; Gutknecht et al., 2016; Sunanda et al., 2024)**".

Model setup differences were described in Section 2 of our manuscript and Section 2.2 of Sunanda et al. (2024), including physical components (MOM6 vs. ROMS), biogeochemical component (COBALTv2 vs. NPZD), resolution (1/12° with 75 layers vs. 1/4° with 40 layers), and a focus on ocean-driven dynamics versus coupled atmosphere–ocean processes. Among these differences, the discrepancy in chlorophyll performance may stem from differences in the phytoplankton functional types and parameterizations within the biogeochemical components. We propose to expand the discussion on this around Line 370 as follows: "The fact that the model simulates the magnitude of observed PP (in carbon units) but overestimates the surface chlorophyll content suggests that it might overestimate the contribution of large phytoplankton, **which is characterized by a high chlorophyll-to-carbon ratio**, compared to small phytoplankton, **characterized by a lower chlorophyll-to-carbon ratio. This overestimation of the contribution of large phytoplankton to the assemblage would indeed explain the good match in primary productivity and high bias in chlorophyll**".

4. The authors state in Section 1 that the state of the Indian Ocean tends to not be reproduced well in global models, but how do performances of past regional models and MOM6-COBALT-IND12 v1.0 compare to that of global models?

Our high-resolution regional model (MOM6-COBALT-IND12 v1.0) with 1/12° resolution demonstrates significant improvements over global models with 0.5° resolution (MOM6-COBALT-global05; Liao et al., 2020), particularly in simulating key mesoscale processes such as eddies, filaments, marginal-sea outflows, and interior oxygen, which critically influence regional ocean circulation, nutrient supply, biological productivity.

To illustrate the enhanced performance of the regional model, we propose to add an additional figure (see Figure R1 below) in the appendix of the revised manuscript, comparing it with MOM6-COBALT-global05. We propose to insert the following comparison discussion around Line 589 of the manuscript: "**The comparison (Figure A8) between MOM6-COBALT-IND12 and the global model of Liao et al. (2020) demonstrates that the regional model more realistically captures high-frequency variability in SSH, mesoscale dynamics, meandering jets, and planetary waves (Rossby and Kelvin waves). These features significantly influence nutrient and oxygen transport and mixing, as well as the timing and spatial patterns of seasonal phytoplankton blooms across the Indian Ocean. Our regional configuration also**

**notably improves the representation of marginal sea outflows, particularly from the Red Sea, where global models typically overestimate overflow. Furthermore, targeted parameter adjustments—including river discharge, nutrient load, detritus remineralization rate, and nitrogen cycle parameterization, enhance salinity and dissolved oxygen simulations in both the Arabian Sea and the Bay of Bengal. Collectively, these refinements substantially improve the realism and reliability of physical and biogeochemical processes simulated by our regional model**".

[Figure]

**Figure R1**. Comparison of sea level anomaly (SLA) intraseasonal variability, subsurface salinity, and subsurface dissolved oxygen between observational products, the regional MOM6-COBALT-IND12 model (labeled as Model-IND12), and the global MOM6-COBALT configuration at 0.5° resolution (labeled as Model-global05). Panels (a–c) show the standard deviation of SLA representing intraseasonal variability (cm); panels (d–f) show the mean salinity averaged over 300–700 m depth (psu); and panels (g–i) show the mean dissolved oxygen averaged over 300–700 m depth. Model minus data comparison statistics—correlation coefficient (r), root mean square error (RMSE), and bias—are shown in parentheses. SLA intraseasonal variability is computed as the standard deviation of linearly detrended SLA, filtered using a 14–120 day band-pass filter.

5. What are the possible causes for discrepancies between observations and simulations for fields other than chlorophyll?

This question overlaps with a portion of general comment #2 regarding the model's varying performance across different aspects. In our response to this comment, we have provided a detailed discussion of the possible reasons for model performance in mixed-layer depth (MLD),

chlorophyll, sea surface height (SSH), sea surface temperature (SST), and oxygen. For details, please refer to our response to general comment #2.

6. The following are more detailed comments. I do not have time to list all the grammar and punctuation errors because they are too numerous.

We propose to revise the manuscript to correct all grammatical and punctuation errors in the revised version.

**Minor Comments:**

1. Line 21: I suggest adding your definition of "northern Indian Ocean" here, because the inclusion of Tanzania among nations along its coast is confusing for readers who assume the term refers to the portion of the Indian Ocean north of the Equator.

We propose to revise as suggested: "The northern Indian Ocean **(from 8.6°S to the northern continental boundaries and 32°E to 114°E)** is central to the livelihood and economy of about one third of the Earth's population".

2. Line 23 and elsewhere: When a set of parentheses include both explanatory text and citations, I believe the correct format is to end the text with a semicolon.

This sentence is restructured in the revised manuscript, and now reads:"provides valuable resources via the "blue economy"**, such as fishery, aquaculture, and marine tourism (Roy, 2019)**".

We reviewed other places in the manuscript where explanatory text and in-text citations appear within the same parentheses. In those cases, we revised the text and inserted semicolons to clearly separate the explanation from the citation, in accordance with journal formatting guidelines.

3. Line 89: There seems to be a mistake in the citation.

We propose to revise as follows: "The model bathymetry was generated using the General Bathymetric Chart of the Oceans version 2020 **(Weatherall et al., 2015)** by averaging the GEBCO bathymetry at a resolution of 15 arcsec in each grid cell".

4. Line 91: How does this manipulated depth differ from the real depth? Is the cross-sectional area of the channel conserved?

We note that only a minor adjustment to the bathymetry was made and this modification only slightly alters the cross-sectional area of the channel. The primary motivation for this adjustment is that, although GEBCO bathymetry is based on observational data, it does not fully represent the true seafloor topography. When interpolated onto the model grid, GEBCO depths can introduce unrealistic artifacts or inconsistencies, particularly in narrow and complex straits or regions where gridded products may not adequately resolve critical flow pathways.

In this context, we manually slightly adjusted the bathymetry near the Red Sea outflow, despite the lack of direct observational constraints for the modified depths. This adjustment enables a more realistic exchange of water between the Red Sea and the adjacent basin. Importantly, the revised bathymetry improves the simulated temperature and salinity fields downstream of the Red Sea outflow, bringing them into closer agreement with observations.

5. Line 112: How well does the model reproduce the observed sea surface salinity without the restoring? Are there spatial variations in the differences between the restored and unrestored salinity values?

The salinity restoring in this model is deliberately kept weak, using a piston velocity of $0.1667 \, \text{m d}^{-1}$, which corresponds to a timescale of approximately 300 days to restore a mixed layer of 50 m depth. Salinity restoring is a standard technique in ocean modeling used to constrain (in this case weakly) surface salinity towards observations, and limit the influence of biases in the reanalysis used to force the model (evaporation, precipitation, river runoff etc.) that would yield unrealistic drifts and influence the circulation.

6. Line 120: I suggest reserving the term "boundary condition" for the scheme by which the boundary condition is specified, which is different from the values that are specified.

We agree with the reviewer's distinction between the boundary condition scheme and the boundary values. We propose to revise this sentence and other similar wording throughout the manuscript according to the reviewer's suggestion.

- Line 120: "In addition, we nudge the **boundary values** towards external forcing with a strong 3-day time-scale for baroclinic normal and tangential velocities entering the mode".

- Line 149: "Sub-tidal velocities, temperature and salinity **at the southern open boundary** are from the monthly ORAS5 (Zuo et al., 2019)".

- Lines 181-182: "Biogeochemical **boundary values** are prescribed from WOA18 monthly climatologies for nitrate, phosphate, silicate, and oxygen".

- Line 416: "We note that the model only extends to 8°S, and therefore does not fully resolve the ITF centered at 5-10°S nor the Southern Equatorial Current at 10-20°S, but receives contributions from ITF waters and southern waters **through the open boundary**".

7. Figure 1 caption and elsewhere: There are no sections in this manuscript called "Methods" so references to this non-existent section should be changed.

We propose to revise this sentence and other similar wording throughout the manuscript to reference sections by number.

- Figure 1 caption: "Pink shading indicates the extent of sponge layers **(see Section 2.2.2)**".

- Figure 15 caption: "see details on water masses in **Section 3.1**".

- Line 468: "We quantify the intraseasonal variability (ISV) in the surface ocean circulation using the intraseasonal standard deviation of the sea level anomaly (SLA, **see Section 3.2**)".

- Figure A2 caption: "temporal evolution averaged over the model domain calculated using a 15-year monthly moving average (**see Section 2.4.2**)".

8. Line 126: ORAS5 is not yet defined at this point.

   We propose to revise as suggested: "nudging the model to time-varying **Ocean Reanalysis System 5** (ORAS5) temperature and salinity with a time-scale increasing from 12 days"

9. Line 176: "Initial DIC and alkalinity were initialized" does not make sense.

   We propose to revise as suggested: "**Dissolved inorganic carbon (DIC) and alkalinity were initialized** using annual means from GLODAPv2 which are representative of year 2002".

10. Line 177: Having "a small impact" is different from having "a negligible impact" or "little impact". Which do you mean?

    The impact is negligible in this context. We propose to revise the sentence as follows: "This initial value has **a negligible impact** on the solution as most of these remaining tracers have turnover time-scales much shorter than the 32 years spin-up duration.

11. Line 198: What is meant by "long-term decadal"? Long-term as well as decadal?

    We propose to revise for clarity: "that retain the seasonality and **the decadal anthropogenic increase** in deposition but remove the interannual variability".

12. Figure 3 and elsewhere: I suggest not using the word "data" to mean observations.

    We propose to update Figure 3 and the related figures (Figures 3, 7, 9, 12, 13, 14, 17) accordingly by explicitly labeling the observational datasets as "Obs". For other cases (Figures 4, 5, 8, 16, 18, 19, 21, A5), where the datasets are not strictly observations but are derived from observational sources, we refer to them as observation-based products and label them as "Obs-based products" for clarity.

[Figure]

**Figure R2**. Revised version of Figure 3. Sea surface temperature (SST) during a-c) winter (December-February) and d-f) summer (June-August) monsoons. (a,d) OISST optimum interpolation from observations, (b,e) MOM6-COBALT-IND12 model and (c,f) differences between model and observations. Correlation coefficients r, RMSE and bias between the observed and model seasonal means are indicated. See details on observations in Table 2. Model results are averaged over the 1980-2020 period.

13. Line 246: What is the source of the updated data?

We propose to add the source of the updated data and revised the text as follows: "the mixed layer depth climatology from De Boyer Montégut et al. (2004; updated in November 2008; **https://mld.ifremer.fr/Surface_Mixed_Layer_Depth.php**)".

14. Line 287: How is the surface heat flux calculated in this model? What are the inputs?

MOM6 employs a COARE bulk formula scheme (Large & Yeager, 2004) to compute surface air–sea heat exchange when bulk forcing is applied. The required inputs include 10 m wind components (u and v), air temperature at 10 m, specific humidity at 10 m, net downward shortwave and longwave radiation, precipitation, and sea level pressure. Using these inputs, the COARE algorithm computes the following outputs: sensible heat flux, latent heat flux, net longwave radiation, and net surface heat flux.

We propose to clarify the heat flux computation method in Section 2.2.1 and revise the text as follows: "In the ocean model, air–sea heat fluxes were computed using the bulk algorithm of Large and Yeager (2004**), which requires atmospheric input variables referenced at 10 m. As**

**the ERA5 forcing provides near-surface temperature and humidity at 2 m, these variables were vertically adjusted to 10 m following the procedure recommended by Large and Yeager (2004)**".

15. Lines 292 and 293: Are these directions stated correctly?

We compared the simulated circulation directions with both the OSCAR data product and schematic circulation maps presented in Schott and McCreary (2001; Figures 8–11) and Schott et al. (2009; Figures 3–4). Based on these comparisons, we confirm that the seasonal current directions and their reversals are correctly represented.

We propose to clarify the ocean circulation description and revised the text in the Lines 292-293 as follows: "**MOM6-COBALT-IND12 reproduces the observed seasonal reversal of the main current systems, as confirmed by comparison with the updated OSCAR drifters database (arrows on Figure 4)**. In the Equatorial band, these seasonal changes include the shift from an eastward transport by the Northeast Monsoon Current (Equator-10°N) and westward transport by the South Equatorial Countercurrent (5°S-Equator) in winter, to a mostly westward transport by the the Southwest Monsoon Current in summer (Equator-10°N, Figure 4)".

16. Lines 308 and 422: I suggest labeling these places in Figure 1.

We propose to add labels to Figure 1 clearly identifying Socotra Island and the Bab-El-Mandeb Strait.

[Figure]

**Figure R3**. Revised version of Figure 1. Domain and bathymetry of the regional Indian Ocean MOM6-COBALT-IND12. Pink shading indicates the extent of sponge layers (see Section 2.2.2). Major rivers are indicated in red. Socotra Island and the Bab-el-Mandeb Strait are labeled on the map.

17. Section 4.3: Why are arrows IX through XII included in Figure 7 if they are not discussed?

The context related to arrows IX-XII is discussed in Section 7, where we describe the interannual variability of sea level. We propose to clarify references to Figure 7 in the descriptions of these arrows in Lines 542-546 of Section 7, as follows: "In particular, it simulates the upwelling anomaly observed between September and January during positive IOD phases along the coasts of the Bay of Bengal (**arrows for waves IX and X**, SLA interannual anomalies of -12 to -5 cm), and the downwelling anomaly observed during negative IOD phases (**arrows for waves XI and XII**, same months, SLA interannual anomalies of +5 to +12 cm, Figure 7)".

18. Lines 360–362: I suggest rewriting this sentence to clarify the proposed relationship between 1) low surface chlorophyll and 2) the weak seasonality of surface chlorophyll on one hand and 1) strong stratification, 2) lower nutrient supply, and 3) the presence of the subsurface chlorophyll maxima. Do all three proposed causes affect both aspects of the surface chlorophyll?

We propose to clarify this sentence: "**In the Bay of Bengal, the persistently low surface chlorophyll and its weak seasonality primarily result from strong salinity-driven stratification, which suppresses vertical nutrient supply to the mixed layer year-round (Sarma and Aswanikumar, 1991). Additionally, the presence of a subsurface chlorophyll maximum confines most primary production below the mixed layer, further reducing surface chlorophyll levels and attenuating their seasonal variability (Sarma and Aswanikumar, 1991)**".

19. Line 397: What is meant by "northeastern Arabian"?

\We propose to revise as follows: "The Red Sea and Persian Gulf overflows contribute warm and salty waters (>13°C and >35.6 psu) **to the Gulf of Aden and Gulf of Oman in the Arabian Sea, respectively** (You and Tomczak, 1993)".

20. Line 403: What is the definition of a shadow zone?

We propose to remove the use of 'shadow zone' in this line. The planned revised sentence is as follows: " Finally, intermediate temperature and salinity in the Bay of Bengal (about 10°C and 35 psu) arise from the relatively weak thermocline ventilation, mostly maintained by the eastward transport from the Arabian Sea and Equatorial region".

21. Lines 462–465: What is the definition of hypoxia used here?

Hypoxia criterion has already been explicitly defined in the original manuscript (lines 456–459) as follows: "At the basin scale, MOM6-COBALT-IND12 reproduces relatively well the observed OMZ volumes defined by thresholds above 30 µmol kg−1, in particular the volume of **hypoxic waters delimited by 60 µmol kg$^{-1}$** (approximately $1\times10^{16}$ m$^3$) and the volume of low oxygenated waters delimited by 100 µmol kg$^{-1}$ (approximately $2\times10^{16}$ m$^3$, Figure 16b)".

22. Lines 523 and 547: "opposed" is not the correct word to use here.

We propose to revise as follows.

- Line 523: During negative IODs, anomalous westerly winds lead to **the opposite** east/west pattern in SST and thermocline depth.

- Line 547: the model also simulates the weaker SLA anomalies of **opposite sign.**

23. Lines 559–560: The positive primary production anomaly in the Arabian Sea is much smaller in the model than in observations. What might be the causes for this?

We have discussed possible reasons for this discrepancy in Lines 564–567 of the original manuscript. To improve clarity, we propose to revise this section as follows, with boldface indicating the changes:

"The model captures remarkably well the pattern and sign of the observed PP anomalies during both negative and positive IODs (correlation coefficient r > 0.7), although the amplitude of the anomaly is slightly lower in the model than in the CbPM satellite product (RMSE of 60 to 80 mg C m$^{-2}$ d$^{-1}$, bias of -1.5 to -10.5 mgC m$^{-2}$ d$^{-1}$, Figure 21 c-d). **On the one hand, the model–observation discrepancy in PP anomaly amplitude may stem from weak nutrient variability associated with the shallow MLD. On the other hand,** the amplitude of the PP anomalies obtained from satellite products remains uncertain. For instance, the PP anomaly composites obtained from another satellite product (CAFE) show similar patterns but with an amplitude that is about half of the CbPM satellite product (Figures 21 and A7 panels a and c). The amplitude of the anomaly in the model essentially sits in between the two satellite products (RMSE of 60-80 mg C m$^{-2}$ d$^{-1}$ in both cases and absolute bias between -10.5 and +24 mg C m$^{-2}$ d$^{-1}$, see Figures 21 and A7). **Given the discrepancies among observational datasets, the model likely provides a reasonable estimate of PP variability. Nevertheless, additional in situ observations are needed to further constrain the PP response to the IOD**."

24. Appendix: The title as well as some of the figures are in the wrong places. Citation: https://doi.org/10.5194/egusphere-2024-3646-RC2

This is indeed a typesetting issue in LaTeX (Overleaf). We propose to relocate all supplementary figures (Figures A1–A5) to Section "Appendix A: Supplementary Figures," positioned before the section References.

**References not in manuscript:**

Bristow, L. A., et al. (2017), N2 production rates limited by nitrite availability in the Bay of Bengal oxygen minimum zone, Nature Geoscience, 10(1), 24-29, doi:10.1038/ngeo2847.

Chakraborty, K., L. Rose, T. Bhattacharya, J. Ghosh, P. K. Ghoshal, and A. Akhand (2023), Primary Productivity Dynamics in the Northern Indian Ocean: An Ecosystem Modeling Perspective, in Dynamics of Planktonic Primary Productivity in the Indian Ocean, edited by S. C. Tripathy and A. Singh, pp. 169-190, Springer International Publishing, Cham, doi:10.1007/978-3-031-34467-1_8.

Drenkard, E. J., et al. (2024), A regional physical-biogeochemical ocean model for marine resource applications in the Northeast Pacific (MOM6-COBALT-NEP10k v1.0), *Geosci. Model Dev. Discuss.*, *2024*, 1-67, doi:10.5194/gmd-2024-195.

Fox-Kemper, B., G. Danabasoglu, R. Ferrari, S. M. Griffies, R. W. Hallberg, M. M. Holland, M. E. Maltrud, S. Peacock, and B. L. Samuels (2011), Parameterization of mixed layer eddies. III: Implementation and impact in global ocean climate simulations, Ocean Modelling, 39(1), 61-78, doi:https://doi.org/10.1016/j.ocemod.2010.09.002.

Gutknecht, E., G. Reffray, M. Gehlen, I. Triyulianti, D. Berlianty, and P. Gaspar (2016), Evaluation of an operational ocean model configuration at 1/12° spatial resolution for the Indonesian seas (NEMO2.3/INDO12) – Part 2: Biogeochemistry, Geosci. Model Dev., 9(4), 1523-1543, doi:10.5194/gmd-9-1523-2016.

Large, W. G., and S. G. Yeager (2004), Diurnal to decadal global forcing for ocean and sea-ice models: The data sets and flux climatologies, edited, National Center for Atmospheric Research Boulder.

Liao, E., L. Resplandy, J. Liu, and K. W. Bowman (2020), Amplification of the Ocean Carbon Sink During El Niños: Role of Poleward Ekman Transport and Influence on Atmospheric $CO_2$, *Global Biogeochemical Cycles*, *34*(9), e2020GB006574, doi:https://doi.org/10.1029/2020GB006574.

Ross, A. C., et al. (2023), A high-resolution physical–biogeochemical model for marine resource applications in the northwest Atlantic (MOM6-COBALT-NWA12 v1.0), *Geosci. Model Dev.*, *16*(23), 6943-6985, doi:10.5194/gmd-16-6943-2023.

Schott, F. A., S.-P. Xie, and J. P. McCreary Jr (2009), Indian Ocean circulation and climate variability, *Reviews of Geophysics*, *47*(1), doi:https://doi.org/10.1029/2007RG000245.

Sunanda, N., J. Kuttippurath, R. Peter, and A. Chakraborty (2024), An atmosphere–ocean coupled model for simulating physical and biogeochemical state of north Indian Ocean: Customisation and validation, Ocean Modelling, 191, 102419, doi:https://doi.org/10.1016/j.ocemod.2024.102419.

Udaya Bhaskar, T. V. S., V. V. S. S. Sarma, and J. Pavan Kumar (2021), Potential Mechanisms Responsible for Spatial Variability in Intensity and Thickness of Oxygen Minimum Zone in 16 the Bay of Bengal, Journal of Geophysical Research: Biogeosciences, 126(6), e2021JG006341, doi:https://doi.org/10.1029/2021JG006341.